# Geometry Meets Incentives: Sample-Efficient Incentivized Exploration with Linear Contexts

**Benjamin Schiffer**
Department of Statistics
Harvard University
1 Oxford St
bschiffer1@g.harvard.edu

**Mark Sellke**
Department of Statistics
Harvard University
1 Oxford St
msellke@fas.harvard.edu

## Abstract

In the incentivized exploration model, a principal aims to explore and learn over time by interacting with a sequence of self-interested agents. It has been recently understood that the main challenge in designing incentive-compatible algorithms for this problem is to gather a moderate amount of initial data, after which one can obtain near-optimal regret via posterior sampling. With high-dimensional contexts, however, this *initial exploration* phase requires exponential sample complexity in some cases, which prevents efficient learning unless initial data can be acquired exogenously. We show that these barriers to exploration disappear under mild geometric conditions on the set of available actions, in which case incentive-compatibility does not preclude regret-optimality. Namely, we consider the linear bandit model with actions in the Euclidean unit ball, and give an incentive-compatible exploration algorithm with sample complexity that scales polynomially with the dimension and other parameters.

## 1 Introduction

The exploration/exploitation trade-off is fundamental to online decision making. This trade-off is classically exemplified by the multi-armed bandit problem, where a single agent chooses actions sequentially and learns to improve over time. In this setting, the agent has clear justification for early exploration because they can reap future rewards by exploiting the knowledge gained by exploration. But what if agents are unable to reap these future rewards, and so each decision must be justifiable on its own terms without taking into account future rewards? This may occur when actions are *recommendations* made to different agents by a central platform based on user feedback, as in e-commerce, traffic routing, movies, restaurants, etc. In these settings, the agents may have a prior for the rewards of the actions in addition to the recommendation made by the platform. Therefore, while the platform makes recommendations with the goal of learning over time, individual myopic agents will decline to follow any recommendations which seem suboptimal based on their individual priors.

The *incentivized exploration* problem was introduced in Kremer et al. [2014], Che and Hörner [2018] to understand this fundamental tension, and extends the well-studied problem of *Bayesian Persuasion* in information design Bergemann and Morris [2019], Kamenica [2019]. The model adopted in these early works consists of a finite set of actions, each with a (publicly shared) Bayesian prior distribution for rewards. A sequence of agents arrives one by one, and each agent is recommended an action by a central planner. The central planner's recommendations are made using a (publicly known) randomized algorithm, and the agents are assumed to be selfish and rational, aiming only to maximize their own expected reward. Given the planner's recommendation, each agent computes and chooses the posterior-optimal action (using Bayes' rule). As observed in the original works, thanks to the revelation principle of Myerson [1986], the latter step is equivalent to assuming that the planner's recommendations are *Bayesian incentive-compatible* (BIC) so that rational agents will always follow the recommendations (at least under the assumption of agent homogeneity).

39th Conference on Neural Information Processing Systems (NeurIPS 2025).

Initially, most work on incentivized exploration dealt with small finite sets of actions Mansour et al. [2020, 2022], exploring economic aspects of the problem such as exogenous payments for exploration Frazier et al. [2014], Kannan et al. [2017], Wang and Huang [2018], Agrawal and Tulabandhula [2020], Wang et al. [2023], partial data disclosure Immorlica et al. [2020], and agent heterogeneity Immorlica et al. [2019]. See also the surveys [Slivkins, 2019, Chapter 11] and [Immorlica et al., 2023, Chapter 31]. More recently, extensions to combinatorial action sets, multi-stage reinforcement learning, and linear contexts were also considered in Hu et al. [2022], Simchowitz and Slivkins [2024], Sellke [2023]. In these more complex machine learning settings, a fundamental question is how the regret scales with problem parameters such as the size of the action space.

This quantitative dependence was studied in Sellke and Slivkins [2023], which provided a new two-stage algorithm. The algorithm first obtains a constant number of samples from each arm, and then switches permanently to Thompson sampling. The initial stage enjoys sample-efficiency thanks to a carefully tuned "exponential exploration" strategy, and Thompson sampling is BIC after this constant amount of initial exploration (see Gur et al. [2024] for a more general perspective on the latter property). This yielded polynomial regret dependence on the number of actions and other natural parameters, and ensured the "price of incentivization" in the regret is only additive relative to Thompson sampling, which is known to exhibit near-optimal performance Kaufmann et al. [2012], Bubeck and Liu [2013], Agrawal and Goyal [2017], Russo and Van Roy [2014, 2016], Zimmert and Lattimore [2019], Bubeck and Sellke [2022]. Importantly, however, all of these results require that the rewards for the actions are independent under the prior.

A natural setting to consider dependent actions is the linear bandit model, where Thompson sampling remains a gold-standard algorithm Agrawal and Goyal [2013], Dong and Van Roy [2018]. For this setting, Sellke [2023] showed that Thompson sampling is again BIC after an initial data collection stage under mild conditions, but that the sample complexity of *collecting* initial data can scale exponentially with the dimension. In some applications, using exogenous payments for initial exploration can bypass this exponential barrier, but such workarounds are contingent on problem-specific regulatory and ethical constraints. In our work, we show that the exponential barrier disappears when the action set is the $d$-dimensional unit ball, and we provide an incentive-compatible initial exploration algorithmic with polynomial sample complexity.

## 1.1 Our Results

We consider a linear bandit problem where the set $\mathcal{A}$ of possible actions is the $d$-dimensional unit ball. In this model, the observed reward for agent $t$ using action $\mathbf{A}^{(t)} \in \mathcal{A}$ in round $t$ is

$$r^{(t)} = \langle \mathbf{A}^{(t)}, \boldsymbol{\ell}^* \rangle + w_t,$$

where $w_t \sim N(0, 1)$ and $w_{t_1} \perp\!\!\!\perp w_{t_2}$ for all $t_1 \neq t_2$. We assume that $\boldsymbol{\ell}^*$ is drawn from a known prior distribution $\mu$ on $\mathbb{R}^d$ (known both to agents and to the algorithm). As our model and algorithm will be rotationally invariant, we assume for convenience (without loss of generality) that $\mathbb{E}[\boldsymbol{\ell}^*_i] = 0$ for all $i > 1$ and $\mathbb{E}[\boldsymbol{\ell}^*_1] \geq 0$. Importantly, unlike in Sellke and Slivkins [2023], we do *not* require that $\boldsymbol{\ell}^*_i$ is independent of $\boldsymbol{\ell}^*_j$ for $i \neq j$.

At each time step $t \geq 1$, the algorithm recommends an action $\mathbf{A}^{(t)} \in \mathcal{A}$ to agent $t$. We study Bayesian Incentive Compatible (BIC) algorithms, which means that the algorithm will make recommendations such that rational expectation-maximizing agents will always be incentivized to follow the algorithm's recommendation. More formally, BIC actions can be defined as follows:

**Definition 1.** *An action* $\mathbf{A}^{(t)}$ *at time* $t \geq 1$ *is Bayesian Incentive Compatible (BIC) if for all* $A \in \mathcal{A}$:

$$\mathbb{E}[\langle \boldsymbol{\ell}^*, A \rangle \mid \mathbf{A}^{(t)} = A] = \sup_{A' \in \mathcal{A}} \mathbb{E}[\langle \boldsymbol{\ell}^*, A' \rangle \mid \mathbf{A}^{(t)} = A].$$

*Similarly, a bandit algorithm is BIC if for all* $t \in [1, T]$, *the algorithm recommends a BIC action* $\mathbf{A}^{(t)}$.

For the rest of this paper, we will only consider BIC algorithms and rational expectation-maximizing agents, which means that the recommended action $\mathbf{A}^{(t)}$ will always be the action chosen by agent $t$. Our main goal is to develop BIC algorithms that incentivize the agents to explore the entire action space using poly($d$) samples. More precisely, given the linear reward structure, we ask our algorithm to recommend actions $\mathbf{A}^{(1)}, ..., \mathbf{A}^{(T)}$ such that for some (possibly large) constant $\lambda > 0$ we have

almost surely:

$$\sum_{t=1}^{T} (\mathbf{A}^{(t)})^{\otimes 2} \succeq \lambda \mathbf{I}. \tag{1}$$

(Here we use the standard notation that $M \preceq M'$ if $M' - M$ is positive semi-definite; (1) is equivalent to all eigenvalues of $\sum_{t=1}^{T} (\mathbf{A}^{(t)})^{\otimes 2}$ being at least $\lambda$.) This was termed $\lambda$-spectral exploration in Sellke [2023], and shown to imply Thompson Sampling is BIC after time $T$ under mild conditions.

Therefore, our problem can informally be summarized as the following:

**Problem 2.** *In the linear bandit setting, can we design BIC algorithms that guarantee $\lambda$-spectral exploration of the action space in only a polynomial in $d$ number of steps?*

For some priors, it is impossible to explore the action space with a BIC algorithm (see e.g. [Mansour et al., 2020, Section 4] or [Sellke and Slivkins, 2023, Section 8]). Thus we will require mild non-degeneracy assumptions on the prior. Roughly speaking, $\ell^*$ should not be confined to any half-space and should have neither minuscule nor enormous fluctuations in any direction.

**Assumption 3.** *There exist constants $c_d, \epsilon_d, c_v, K > 0$ such that:*

1. *$\ell^*$ is not confined to any half-space:* $\min_{\|\mathbf{v}\|=1} \Pr(\langle \mathbf{v}, \ell^* \rangle \geq c_d) \geq \epsilon_d$.

2. *$\ell^*$ has non-degenerate covariance:* $\min_{\|\mathbf{v}\|=1} \mathrm{Var}(\langle \mathbf{v}, \ell^* \rangle) \geq c_v$.

3. *$\ell^*$ is sub-gaussian:* $\max_{\|\mathbf{v}\|=1} \mathbb{P}(|\langle \mathbf{v}, \ell^* \rangle| \geq t) \leq 2e^{-t^2/K^2}$ *for all $t > 0$.* .

The first two conditions above are both necessary in some form; see Appendix B. The third is a standard condition on the fluctuations of $\mu$. Our main result is as follows.

**Theorem 4.** *Under Assumption 3, there exists a BIC algorithm (Algorithm 1) which almost surely achieves $\bar{\lambda}$-spectral exploration in sample complexity*

$$\bar{\lambda} \left( \frac{d}{c_v + c_d} \right)^{O(1)} \log(1/\epsilon_d). \tag{2}$$

Note that (2) depends polynomially on $c_d, c_v$ but only logarithmically on $\epsilon_d$. This is important because in typical high-dimensional settings, $\epsilon_d$ will be exponentially small in the dimension while the other parameters will not. The next two propositions give illustrative but not exhaustive examples of such high-dimensional distributions. In the first, we take $\mu$ to be uniform on a convex body $\mathcal{K}$ with $B_r(0) \subseteq \mathcal{K} \subseteq B_1(0)$ for some $0 < r < 1$, and say such $\mu$ is $r$-regular. This is the main setting considered in Sellke [2023], and as explained in Section 5, combining Sellke [2023] with our results yields an end-to-end low-regret algorithm which is $\epsilon$-BIC (i.e. with $\epsilon$ subtracted from the right-hand side in Definition 1) with initial sample complexity poly$(d, 1/r, 1/\epsilon)$. (The combination only satisfies $\epsilon$-BIC because Sellke [2023] only shows Thompson sampling is $\epsilon$-BIC unless actions are well-separated.)

**Proposition 5** (Proof in Appendix L). *Any $r$-regular $\mu$ satisfies Assumption 3 with $c_d = r/3$ and $\epsilon_d = (r/3)^d$ and $c_v = \Omega(r^2/d^2)$ and $K = 1.25$.*

The second example consists of log-concave distributions (e.g. Gaussians). We say a density $d\mu(x) \propto e^{-V(x)}dx$ is $\alpha$-log-concave and $\beta$-log-smooth for $0 < \alpha \leq \beta$ if

$$-\beta I_d \preceq \nabla^2 V(\mathbf{x}) \preceq -\alpha I_d, \quad \forall \mathbf{x} \in \mathbb{R}^d.$$

**Proposition 6** (Proof in Appendix M). *Let $\mu$ be $\alpha d$-log-concave and $\beta d$-log-smooth with mode $\mathbf{x}^*$ satisfying $\|\mathbf{x}^*\| \leq \gamma$. Then Assumption 3 holds with $\epsilon_d \geq \frac{Je^{-J^2\beta d/2}\sqrt{\alpha d}}{(1+J^2\beta d)\sqrt{2\pi}}$ for $J = \gamma + \frac{2}{\sqrt{\alpha d}} + c_d$ and any $c_d > 0$. Further, we may take $c_v = \frac{1}{\beta d}$ and $K = 2(\gamma + \sqrt{\alpha d} + (\alpha d)^{-1/2})$. Conversely, there exists such $\mu$ with $\|\mathbf{x}^*\| = 0$ and $(\alpha, \beta) = (1, 2)$ in which $\min_{\|\mathbf{v}\|=1} \Pr(\langle \mathbf{v}, \ell^* \rangle \geq 0) \leq e^{-\Omega(d)}$.*

We note that if $\mu$ has mean 0, then $\|\mathbf{x}^*\| \leq \alpha^{-1/2}$ so the above results apply (see Fact 27 in the Appendix). In fact when $\mu$ has mean 0, one will always have $\epsilon_d \geq \Omega(1)$ for $c_d = 1$ (as noted within the proof of Proposition 6). Proposition 6 represents the typical case in that $\mu$ can be "mildly off-centered", for example by centering around a point $\mathbf{x}$ drawn from $\mu$ itself (see again Fact 27).

In Appendix B, we show tightness of the dependency of Equation (2) on the parameters of Assumption 3. More formally, we prove three lower bounds, one corresponding to each of the parameters $c_d, c_v, \epsilon_d$.

In doing so, we show that there exist instances of the problem such that no algorithm can achieve 1-spectral exploration with fewer than $\Omega(1/c_d)$ samples, $\Omega(1/c_v)$ samples, and $\Omega(\log(1/\epsilon_d))$ samples. Therefore, we can conclude that Theorem 4 is tight in terms of these three parameters up to polynomial factors. Furthermore, for the distributions discussed in Propositions 5 and 6, these results imply a lower bound on the number of samples that is superlinear in $d$. Whether or not tight lower bounds exist that exactly match the sample complexity in Equation (2) remains an open question.

**On the Smoothness of the Action Space in Online Optimization and Learning**  Our positive results are in contrast with [Sellke, 2023, Proposition 3.9], which shows that $e^{\Omega(d)}$ time can be necessary for BIC exploration for $r$-regular $\mu$. We believe that our results are not specific to the unit ball, but that the fundamental distinction between these two examples is the *smoothness* of the action set. In [Sellke, 2023, Proposition 3.9], the action set is a non-smooth polytope with corners, and the optimal action under the prior distribution is one of the corners. This means that given a small amount of new information, the posterior-optimal action will not change at all. By contrast our main algorithm crucially relies on expansiveness of the function

$$\boldsymbol{\ell}^* \mapsto \arg\max_{\mathbf{x} \in \mathcal{A}} \langle \boldsymbol{\ell}^*, \mathbf{x} \rangle.$$

In our setting $\mathcal{A}$ is the unit ball and so this function is simply $\boldsymbol{\ell}^*/\|\boldsymbol{\ell}^*\|$. However we expect our methods to generalize to other smooth bodies, and plan to pursue this in future work.

It is worth mentioning that the geometry of the action set has long been understood to play an important role in high-dimensional learning and optimization. This was exemplified by the interplay between self-concordant barrier functions Nesterov and Nemirovskii [1994] and linear bandits via online stochastic mirror descent Abernethy et al. [2008], Bubeck et al. [2012a,b, 2018], Bubeck and Eldan [2019], Kerdreux et al. [2021b]. Geometric properties of the action space are also known to yield acceleration for full-information online learning and offline optimization Garber and Hazan [2015], Huang et al. [2017], Levy and Krause [2019], Kerdreux et al. [2021a], Mhammedi [2022], Molinaro [2023], Tsuchiya and Ito [2024].

## 1.2  Additional Notation

We will use the following notation throughout the paper. Unless otherwise specified, $\|\cdot\|$ will refer to the $\ell_2$ norm. We will use $\mathbf{e}_i$ to refer to the $i$th vector of the standard basis. For a random variable $X$, we say that $X$ is $K$-sub-gaussian if $\mathbb{P}(X \geq t) \leq 2e^{-t^2/K^2}$. We use the standard $f(d) = O(g(d))$ (and corresponding $\Omega$) to mean that there exists some constant $C > 0$ such that $f(d) \leq C \cdot g(d)$ for all sufficiently large $d$. We often write $x^{\otimes 2}$ for $xx^\top$. We also use $\mathcal{P}_S(\mathbf{u})$ to represent the projection of the vector $\mathbf{u}$ onto the space $S$.

## 1.3  Outline

The rest of the paper will be organized as follows. In Section 2, we present sketches of our main algorithms and discuss the three key technical ideas behind the algorithm and analysis. In Section 3, we give the detailed main algorithm, present the two key technical propositions, and formally state our main theorem bounding the sample complexity of the algorithm.

## 2  Algorithm and Technical Overview

In this section, we present pseudocode for the main algorithm and give informal intuition for the key technical ideas used to show that this algorithm is BIC and satisfies the sample complexity bound of Theorem 4.

### 2.1  Algorithm Sketch

Before presenting the algorithm, we first state Lemma 7, which is the key observation used by the algorithm to select BIC actions. Informally, this lemma says that for any $\psi$ that is a function of historical actions and rewards and possibly external randomness (but not on future information), the action $\mathbf{A}^{(t)}$ in the direction of $\mathbb{E}[\boldsymbol{\ell}^* \mid \psi]$ is BIC (or any action is BIC if $\mathbb{E}[\boldsymbol{\ell}^* \mid \psi] = \mathbf{0}$). Thus to prove an action $\mathbf{A}^{(t)}$ is BIC, it suffices to find such a $\psi$ so that $\mathbf{A}^{(t)}$ is in the direction of $\mathbb{E}[\boldsymbol{\ell}^* \mid \psi]$ whenever $\mathbb{E}[\boldsymbol{\ell}^* \mid \psi] \neq \mathbf{0}$.

**Lemma 7** (Proof in Appendix D). *Suppose that $\psi$ is a function of the history before time $t$ and potentially some external independent randomness $\xi$, i.e. $\psi = \psi(\mathbf{A}^{(1)}, r^{(t-1)}..., \mathbf{A}^{(t-1)}, r^{(t-1)}, \xi)$.*

*Let $\mathbf{v}$ be any vector in $\mathbb{R}^d$. Define*

$$\text{Exploit}(\psi, \mathbf{v}) := \begin{cases} \frac{\mathbb{E}[\boldsymbol{\ell}^* | \psi]}{\|\mathbb{E}[\boldsymbol{\ell}^* | \psi]\|} & \textit{if } \|\mathbb{E}[\boldsymbol{\ell}^* \mid \psi]\| \neq \boldsymbol{0} \\ \mathbf{v} & \textit{otherwise.} \end{cases}$$

*Then $A^{(t)} = \text{Exploit}(\psi, \mathbf{v})$ is BIC at time $t$.*

Algorithm 1, together with Algorithms 2–3, presents a high-level sketch of the procedure used to prove Theorem 4. The algorithm first uses the prior-based BIC action ($\mathbf{e}_1$) for $\text{poly}(d)$ steps. In order to explore the rest of the action space, Algorithm 1 runs a single loop that repeatedly checks if $\lambda$-spectral exploration has been achieved. If $\lambda$-spectral exploration has not yet been achieved, then Algorithm 1 uses the subroutine InitialExploration to find a BIC action $\mathbf{a}$ that has magnitude at least $\Omega(\epsilon_d)$ when projected onto the space of not-yet-sufficiently explored actions. Algorithm 1 then gives $\mathbf{a}$ as input to the subroutine ExponentialGrowth. ExponentialGrowth returns another BIC action that has at least twice as large of a magnitude when projected onto the not-yet-sufficiently explored space of actions. Algorithm 1 repeatedly passes the action $\mathbf{a}$ through ExponentialGrowth until the BIC action $\mathbf{a}$ has magnitude of at least $\sqrt{\lambda}$ when projected onto the space of not-yet-sufficiently explored actions. Algorithm 1 then uses this action for $\text{poly}(d)$ steps. If $\lambda$-spectral exploration has still not been achieved, then Algorithm 1 explores a new direction by repeating the process of calling InitialExploration followed by repeated calls to ExponentialGrowth. In Sections 2.2 and 2.3, we discuss the main intuition of the subroutines InitialExploration and ExponentialGrowth respectively.

---

**Algorithm 1** BIC Exploration Pseudocode

---

1: Set $\mathbf{A}^{(t)} = \mathbf{e}_1$ for $\text{poly}(d)$ steps
2: **while** $\lambda$-spectral exploration has not yet been achieved **do**
3:   $S \leftarrow$ space of actions that have already been sufficiently explored
4:   $\mathbf{a} \leftarrow \text{InitialExploration}(\cdot)$   ▷ BIC-vector with $\Omega(\epsilon_d)$-magnitude when projected onto $S^\perp$
5:   **while** magnitude of $\mathbf{a}$ when projected onto $S^\perp$ is less than $\sqrt{\lambda}$ **do**
6:     $\mathbf{a} \leftarrow \text{ExponentialGrowth}(\mathbf{a})$   ▷ new BIC-vector with double the magnitude when projected onto $S^\perp$
7:   Set $\mathbf{A}^{(t)} = \mathbf{a}$ for $\text{poly}(d)$ steps

---

---

**Algorithm 2** InitialExploration Pseudocode

---

1: $\mathbf{M} \leftarrow \sum_{i=1}^{t-1} \left(\mathbf{A}^{(i)}\right)^{\otimes 2}$
2: $\mathbf{w}_1, ..., \mathbf{w}_{\ell_\lambda} \leftarrow$ orthonormal eigenvectors of $\mathbf{M}$ with corresponding eigenvalues greater than $\lambda$
3: $S \leftarrow \text{Span}(\mathbf{w}_1, ..., \mathbf{w}_{\ell_\lambda})$   ▷ Space of already-sufficiently explored actions
4: $\hat{y}_i \leftarrow$ empirical estimate of $\langle \boldsymbol{\ell}^*, \mathbf{w}_i \rangle$ for $i \leq \ell_\lambda$
5: $\mathbf{z} \leftarrow \mathbb{E}[\boldsymbol{\ell}^* \mid \hat{\mathbf{y}}]$
6: $f \leftarrow$ function of $\mathbf{z}$ such that $\mathbb{E}[\mathbf{z} f(\mathbf{z})] = 0$ and $f(\mathbf{z}) \in [\Omega(\epsilon_d), 1]$.
7: $E \leftarrow \{\text{Bernoulli}(f(\mathbf{z})) = 1\}$
8: Set $\mathbf{A}^{(t)} = \text{Exploit}(1_E, \mathbf{v})$ for $\mathbf{v} \in S^\perp$   ▷ $\mathbf{A}^{(t)}$ explores a new direction with probability $f(\mathbf{z})$
9: $r \leftarrow r^{(t)}$ if we explored and otherwise $r \leftarrow N(0, 1)$
10: **return** $\text{Exploit}(\text{sign}(r), \mathbf{v})$   ▷ where $\mathbf{v}$ is in $S^\perp$

---

---

**Algorithm 3** ExponentialGrowth Pseudocode

---

1: $S \leftarrow$ space of actions that have already been sufficiently explored
2: Set $\mathbf{A}^{(t)} = \mathbf{a}$ for $\text{poly}(d)$ steps to observe a set of rewards $\{r^{(t)}\}$.
3: $R \leftarrow$ average of rewards from the component of $\mathbf{a}$ projected onto the space of not-yet-sufficiently explored actions ($S^\perp$)
4: **return** $\text{Exploit}(\text{sign}(R), \mathbf{v})$   ▷ where $\mathbf{v}$ is in $S^\perp$

---

## 2.2 Initial Exploration

The goal of the InitialExploration routine (Algorithm 2) is to find a BIC action that has sufficient magnitude when projected onto the not-yet-sufficiently explored space of actions $S^\perp$. To do this, we

first design an event $E$ based on the historical actions and rewards $H_t$ and external randomness such that $\mathbb{P}(E \mid H_t) \geq \Omega(\epsilon_d)$ for all histories $H_t$ and such that conditional on $E$, $\boldsymbol{\ell}^*$ has 0 expectation when projected onto any direction in $S$. We then choose the BIC action $A^{(t)}$ in Line 8 so that $A^{(t)}$ explores $S^\perp$ whenever event $E$ holds. More concretely, the second condition on $E$ implies that the action $A^{(t)} = \text{Exploit}(1_E, \mathbf{v})$ for any $\mathbf{v} \in S^\perp$ will satisfy $A^{(t)} \in S^\perp$ whenever event $E$ holds, and $A^{(t)}$ is BIC by Lemma 7. Using this BIC action on Line 8 therefore explores $S^\perp$ whenever event $E$ holds. We define the signal $r$ as the reward at time $t$ if event $E$ holds and otherwise as independent $N(0,1)$ noise. We next show that, conditional on the sign of $r$, the expectation of $\boldsymbol{\ell}^*$ always has sufficient magnitude when projected onto $S^\perp$ (see Lemma 14). This implies that for any $\mathbf{v} \in S^\perp$, the action $\text{Exploit}(\text{sign}(r), \mathbf{v})$ will have sufficient magnitude when projected onto $S^\perp$ as desired. Therefore, we can return the action $\text{Exploit}(\psi, \mathbf{v})$, which is BIC by Lemma 7.

All that remains is to define the event $E$ from the previous paragraph. Formally, we want an event $E$ that is a function of the historical actions and rewards and independent random variable $\xi$ such that $\mathbb{E}[\langle \boldsymbol{\ell}^*, \mathbf{w}_i \rangle \mid E] = 0$ for all $i \leq \ell_\lambda$ and such that $\mathbb{P}(E \mid H_t) \geq \Omega(\epsilon_d)$ for all histories $H_t$. The key to constructing $E$ is Lemma 8, which implies that for any random variable $\mathbf{x}$ not confined to any half-spaces, there exists a function $f$ such that $\mathbf{x}$ has expectation equal to $\mathbf{0}$ conditional on the event $\{\text{Bernoulli}(f(\mathbf{x})) = 1\}$

**Lemma 8** (Proof in Appendix G). *Let $\mu$ be a probability distribution on $\mathbb{R}^d$ with finite first moment and suppose for $0 < \epsilon \leq 1/2$ we have that*

$$\min_{\|\mathbf{v}\|=1} \mathbb{E}^{\mathbf{x} \sim \mu}[\langle \mathbf{v}, \mathbf{x} \rangle_+] \geq \epsilon.$$

*Then there exists a Borel measurable function $f : \mathbb{R}^d \to \left[ \frac{\epsilon}{4 \max(\|\mathbb{E}[\mathbf{x}]\|, 1)}, 1 \right]$ with $\mathbb{E}[\mathbf{x} f(\mathbf{x})] = \mathbf{0}$.*

If we knew the exact values of the vector $\mathbf{x} := (\langle \boldsymbol{\ell}^*, \mathbf{w}_1 \rangle, ..., \langle \boldsymbol{\ell}^*, \mathbf{w}_{\ell_\lambda} \rangle)$, then we could directly apply Lemma 8 to $\mathbf{x}$ and use the resulting function $f$ to define the event $E = \{\text{Bernoulli}(f(\mathbf{x})) = 1\}$. This event $E$ would satisfy the desired property that $\mathbb{E}[\langle \boldsymbol{\ell}^*, \mathbf{w}_i \rangle \mid E] = 0$ for $i \leq \ell_\lambda$ and that $\Pr(E \mid H_t) = \Omega(\epsilon_d)$ for all $H_t$. However, we do not know the exact values of $\langle \boldsymbol{\ell}^*, \mathbf{w}_1 \rangle, ..., \langle \boldsymbol{\ell}^*, \mathbf{w}_{\ell_\lambda} \rangle$ because we do not know $\boldsymbol{\ell}^*$, and therefore this event $E$ is not a function of historical actions and rewards. Instead, we can estimate $\langle \boldsymbol{\ell}^*, \mathbf{w}_i \rangle$ as $\hat{y}_i$ using the historical actions and returns. These estimates will be relatively accurate because these directions are already well-explored. Defining $\mathbf{z} = \mathbb{E}[\boldsymbol{\ell}^* \mid \hat{\mathbf{y}}]$, we show that $\mathbf{z}$ also satisfies the assumption of Lemma 8 (Lemma 17). Therefore, applying Lemma 8 to $\mathbf{z}$ gives a function $f$ such that $f(\mathbf{z}) \geq \Omega(\epsilon_d)$ for all $\mathbf{z}$. $\mathbf{z}$ is a function of historical actions and rewards and external randomness as desired. Defining the event $E = \{\text{Bernoulli}(f(\mathbf{z})) = 1\}$, we have as desired that $\mathbb{E}[\langle \boldsymbol{\ell}^*, \mathbf{w}_i \rangle \mid E] = 0$ for all $i \leq \ell_\lambda$ and that $\mathbb{P}(E \mid H_t) = \Omega(\epsilon_d)$ for all $H_t$.

### 2.3 Exponential Growth

The goal of the ExponentialGrowth routine (Algorithm 3) is to take a BIC action $\mathbf{a}$ and return a new BIC action that has twice as large of a magnitude when projected onto the not-yet-sufficiently explored space of actions $S^\perp$. Using Lemma 7, we will find a signal $R$ such that for any $\mathbf{v} \in S^\perp$, the action $\text{Exploit}(R, \mathbf{v})$ will have twice as large magnitude when projected onto $S^\perp$ as $\mathbf{a}$ has.

The key intuition is that because the action space is curved, conditioning on noisy information about the sign of $\boldsymbol{\ell}^*$ in a specific direction will increase the magnitude of the expectation of $\boldsymbol{\ell}^*$ projected in that direction. Consider the following simplified example. Suppose that $d = 2$. Also suppose we already know $\ell^*_1$, and the goal is to explore $\ell^*_2$. Furthermore, assume that the initial BIC action is $\mathbf{A}^{(t)}$ shown in the left-most diagram of Figure 1 that has $\epsilon$ as the $y$-coordinate. Using this action gives a rewards $r^{(t)}$. Because we know the value of $\ell^*_1$, we can remove this component of the reward $r^{(t)}$, and we are left with a signal $r = \epsilon \ell^*_2 + N(0, \sigma^2)$ for some $\sigma^2 > 0$. If we take $\mathbf{A}^{(t+1)}$ to be the unit vector in the direction of $\mathbb{E}[\boldsymbol{\ell}^* \mid r]$, then we will have that the new $y$-coordinate is $2\epsilon$. By Lemma 7, this choice of $\mathbf{A}^{(t+1)}$ is BIC, which is an important consequence of the curved action space. We then observe $r^{(t+1)}$, and we can repeat this process again to find a BIC action $\mathbf{A}^{(t+2)}$ that has y-coordinate of $4\epsilon$. Therefore, in this simplified example we are able to exponentially grow the y-coordinate for BIC actions, which is a consequence of the curvature of the action space. This process is demonstrated in Figure 1.

Equipped with the intuition of the previous paragraph, we now analyze Algorithm 3. Recall that $S$ is the space of actions that have already been sufficiently explored. Algorithm 3 first uses the action $\mathbf{A}^{(t)} = \mathbf{a}$ for poly$(d)$ steps. Taking an average of the rewards $\{r^{(t)}\}$ from these actions gives

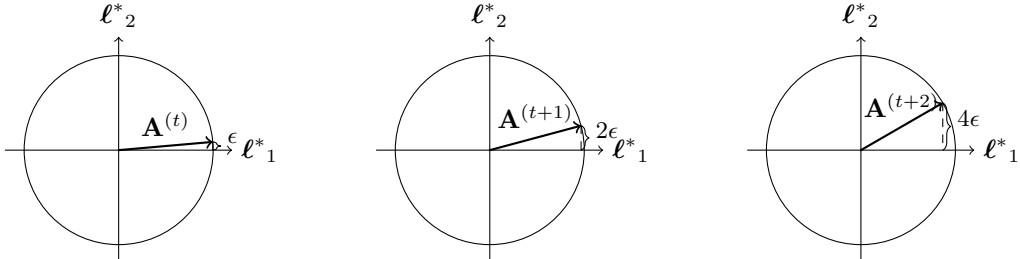

Figure 1: Diagram illustrating exponentially growing exploration. First, we have a BIC action $\mathbf{A}^{(t)}$ with y-coordinate $\epsilon$. Using $r^{(t)}$, we design a signal conditional on which the expectation of $\boldsymbol{\ell}^*{}_2$ doubles. Exploit$(\cdot)$ then gives a BIC action $A^{(t+1)}$ with $y$-coordinate $2\epsilon$. Using $r^{(t+1)}$, we again double the conditional expectation of $\boldsymbol{\ell}^*{}_2$ and get BIC action $A^{(t+2)}$ with y-coordinate $4\epsilon$. Increasing the conditional expectation of $\boldsymbol{\ell}^*{}_2$ gives a BIC action with a larger $y$-coordinate, and this action's feedback gives a stronger signal that more rapidly increases the conditional expectation of $\boldsymbol{\ell}^*{}_2$. This allows us to "bootstrap" an exponentially weak starting signal all the way to constant signal strength without suffering exponential sample complexity.

a close estimate of $\langle \mathbf{a}, \boldsymbol{\ell}^* \rangle$. Using the previous observed actions and rewards, we can remove the component of $\langle \mathbf{a}, \boldsymbol{\ell}^* \rangle$ that comes from $\mathbf{a}$ projected onto $S$. This leaves a signal $R$ which is the average of rewards from the component of $\mathbf{a}$ projected onto just $S^\perp$. Using concentration laws for conditional probabilities (Lemmas 15 and 16), we formalize the intuition from the previous paragraph to show that for any $\mathbf{v} \in S^\perp$, the projection of Exploit$(R, \mathbf{v})$ onto $S^\perp$ will always have magnitude at least $2\mathcal{P}_{S^\perp}(\mathbf{a})$. The action Exploit$(R, \mathbf{v})$ is BIC by Lemma 7. Therefore, we have found a new BIC action Exploit$(R, \mathbf{v})$ that has twice as large magnitude when projected onto $S^\perp$ as $\mathbf{a}$.

### 2.4 Pushing eigenvalues upwards

An important step of the proof of Algorithm 1 is showing that the while loop will terminate in polynomial time. To show this, we show that the action $\mathbf{a}$ used for poly$(d)$ steps at the end of each round of the while loop sufficiently increases the eigenvalues of $\mathbf{M}^{(t)} := \sum_{i=1}^{t} \left(\mathbf{A}^{(i)}\right)^{\otimes 2}$. Note that at each time $t$, the matrix $\mathbf{M}^{(t)}$ increases by a rank-1 update, i.e. $\mathbf{M}^{(t+1)} = M^{(t)} + \left(\mathbf{A}^{(t+1)}\right)^{\otimes 2}$. In order to show that we will eventually achieve $\lambda$-spectral exploration, we must show that the small eigenvalues of $\mathbf{M}^{(t+1)}$ increase relative to the small eigenvalues of $\mathbf{M}^{(t)}$ after each round of the while loop. Although we do not follow this route, we mention that Golub [1973] provides exact descriptions for the eigenvalues of rank-one updates as above, giving a rational function $\omega(x)$ (with coefficients depending on $\mathbf{M}^{(t)}$ and $\mathbf{A}^{(t+1)}$) which has roots equal to the eigenvalues of $\mathbf{M}^{(t+1)}$. However, it is not clear how helpful this is for the quantitative estimates we require.

At first glance, we might hope that we can sufficiently increase all of the eigenvalues of $\mathbf{M}^{(t)}$ in just $d$ rounds if in each round we partially explore a new not-yet-explored direction. However, this unfortunately is not always the case. For example, suppose in the first round we use BIC action $\mathbf{A}^{(1)} = (1, 0, 0, ..., 0)$. Writing $\varphi = 1/\sqrt{5}$, we then in the next $d-1$ actions use the BIC actions $\mathbf{A}^{(2)} = (2\varphi, -\varphi, 0, 0, ..., 0)$, $\mathbf{A}^{(3)} = (0, 2\varphi, -\varphi, 0, 0, ..., 0)$, and so on until $\mathbf{A}^{(d)} = (0, 0, ..., 0, 2\varphi, -\varphi)$. Then each $\mathbf{A}^{(i)}$ has distance $\varphi = \Omega(1)$ from the span of the preceding actions, so we might hope that this already yields $\Omega(1)$-spectral exploration. However, note that the vector $\mathbf{x} = (1, 2, 4, \ldots, 2^{d-1})$ satisfies $\langle \mathbf{x}, \mathbf{A}^{(i)} \rangle = 1_{i=1} \leq 2^{-(d-1)}\|\mathbf{x}\|$ for all $1 \leq i \leq d$. It follows that the matrix $\mathbf{M}^{(d)} = \sum_{i=1}^{d} \left(\mathbf{A}^{(i)}\right)^{\otimes 2}$ has smallest eigenvalue which is *exponentially small* in $d$ (since $\langle \mathbf{x}, \mathbf{M}^{(d)}\mathbf{x} \rangle / \|\mathbf{x}\|^2$ is exponentially small).

We show that our algorithm terminates in poly$(d)$ steps using the linear algebraic Lemma 9 below, which may be of independent interest. Informally, this lemma says that if $\mathbf{u}$ has non-negligible projection onto the space orthogonal to the large-or-medium eigenvalues of $\mathbf{M}$, then the rank-1 update of $\mathbf{M} + \mathbf{u}^{\otimes 2}$ increases the total sum of the small-or-medium eigenvalues non-negligibly.

**Lemma 9** (Proof in Appendix E)**.** *Let* $\mathbf{v}_1, \mathbf{v}_2, ...\mathbf{v}_j \in \mathbb{R}^d$ *such that* $\|\mathbf{v}_i\| = 1$. *Define* $\mathbf{M} = \sum_{i=1}^{j} \mathbf{v}_i^{\otimes 2}$. *Suppose* $\mathbf{w}_1, ..., \mathbf{w}_d$ *are orthonormal eigenvectors of* $\mathbf{M}$ *with corresponding eigenvalues*

$\lambda_1 \geq ... \geq \lambda_d \geq 0$. Define $\ell_\epsilon$ as the largest index such that $\lambda_j \geq \epsilon$ for some $0 < \epsilon < 1$, and define $S = \mathrm{Span}(\mathbf{w}_1, ..., \mathbf{w}_{\ell_\epsilon})$. Suppose $\mathbf{u}$ is a vector such that $\|P_{S^\perp}(\mathbf{u})\|_2^2 \geq \epsilon$, and define $\mathbf{M}' = \mathbf{M} + \mathbf{u}^{\otimes 2}$. Let $\mathbf{w}_1', ... \mathbf{w}_d'$ be the orthonormal eigenvectors of $\mathbf{M}'$ with corresponding eigenvalues $\lambda_1' \geq ... \geq \lambda_d'$. Finally, let $\ell$ be the largest index such that $\lambda_\ell \geq \frac{200 d^3}{\epsilon^2}$. Then

$$\sum_{i=\ell+1}^{d} \lambda_i' \geq \epsilon/2 + \sum_{i=\ell+1}^{d} \lambda_i.$$

## 3 Main Results

In this section, we will formally state our main algorithms and theorems. For presentation purposes, for the formal algorithms and theorems we will define

$$\lambda := \min\left(1, \min(\delta_{L15}, \delta_{L16}, 1/c_{L16})^2 \frac{(c_v/\sqrt{8\pi})^2}{4d(K\sqrt{\pi}+1)^2}\right) = \Omega(c_v/d),$$

where $\delta_{L15}$ is a constant from Lemma 15 and $\delta_{L16}, c_{L16}$ are constants from Lemma 16. Note that if we can do $\lambda$-spectral exploration in $T$ rounds with this value of $\lambda$, then for any $\bar{\lambda}$ we can do $\bar{\lambda}$-spectral exploration in $\lceil \frac{\bar{\lambda}}{\lambda} \rceil \cdot T$ rounds by simply repeating the algorithm $\lceil \frac{\bar{\lambda}}{\lambda} \rceil$ times. Therefore, if we can ensure $\lambda$-spectral exploration for the $\lambda$ value described above in $\mathrm{poly}(d)$ rounds, then we can also ensure $\bar{\lambda}$-spectral exploration in $\bar{\lambda} \cdot \mathrm{poly}(d)$ rounds for any $\bar{\lambda} \geq \lambda$.

### 3.1 Algorithms

Our main algorithm is presented in Algorithm 4, with subroutines in Algorithms 5 and 6. Note that these three algorithms directly correspond to the pseudocode presented in Algorithms 1–3.

---
**Algorithm 4** BIC Exploration

---
**Input:** $\lambda$
1: $\kappa \leftarrow \max\left(\frac{1}{\lambda c_{L17}}, \frac{4d(K\sqrt{\pi}+1)(1+\frac{1}{\lambda})}{c_v^2/(8\pi)}\right)$
2: $\mathbf{v}_1 \leftarrow \mathbf{e}_1$
3: **for** $t \in [0 : \kappa)$ **do**
4:      Set $\mathbf{A}^{(t)} = \mathbf{v}_1$
5:      $q_1^t \leftarrow r^{(t)}$
6: $t \leftarrow \kappa$
7: $j \leftarrow 1$
8: **while** minimum eigenvalue of $\mathbf{M} = \sum_{i=1}^{j} \mathbf{v}_i^{\otimes 2}$ is smaller than $\lambda$ **do**
9:      $\mathbf{w}_1, ..., \mathbf{w}_d \leftarrow$ orthonormal eigenvectors of $\mathbf{M}$ with corresponding eigenvalues $\lambda_1 \geq ... \geq \lambda_d$.
10:      $\ell_\lambda \leftarrow \max\{i : \lambda_i \geq \lambda\}$
11:      $S \leftarrow \mathrm{Span}(\mathbf{w}_1, ..., \mathbf{w}_{\ell_\lambda})$
12:      $\mathbf{a}, t \leftarrow \mathrm{InitialExploration}(\{\mathbf{w}_i\}_{i=1}^{d}, \{\lambda_i\}_{i=1}^{\ell_\lambda}, \{\mathbf{v}_i\}_{i=1}^{j}, \{\{q_i^{t'}\}_{t'=0}^{\kappa-1}\}_{i=1}^{j}, t)$
13:      **while** $\|\mathcal{P}_{S^\perp}(\mathbf{a})\| \leq \sqrt{\lambda}$ **do**
14:          $\mathbf{a}, t \leftarrow \mathrm{ExponentialGrowth}(\mathbf{a}, \{\mathbf{w}_i\}_{i=1}^{\ell_\lambda}, \{\lambda_i\}_{i=1}^{\ell_\lambda}, \{\mathbf{v}_i\}_{i=1}^{j}, \{\{q_i^{t'}\}_{t'=0}^{\kappa-1}\}_{i=1}^{j}, t)$
15:      $\mathbf{v}_{j+1} \leftarrow \mathbf{a}$
16:      **for** $t' \in [0 : \kappa)$ **do**
17:          Set $\mathbf{A}^{(t+t')} = \mathbf{a}$
18:          $q_{j+1}^{t'} \leftarrow r^{(t+t')}$
19:      $t \leftarrow t + \kappa$
20:      $j \leftarrow j + 1$

---

Lemma 10 shows that the application of Lemma 8 (in the form of Lemma 17) in Algorithm 5 is valid.
**Lemma 10** (Proof in App C)**.** *Every time Algorithm 5 (Line 2) calls Algorithm 4 to define $\hat{y}_\ell$, we have $\hat{y}_\ell \overset{d}{=} x_\ell^* + N(0, c_{L17} \frac{\lambda}{\lambda_\ell})$. Thus, $\mathbf{z}(\hat{\mathbf{y}})$ (Line 4) satisfies the assumptions of Lemma 8 with $\epsilon = \frac{\epsilon_d c_d}{4}$.*

---

**Algorithm 5** InitialExploration

**Input:** $\{\mathbf{w}_i\}_{i=1}^d, \{\lambda_i\}_{i=1}^{\ell_\lambda}, \{\mathbf{v}_i\}_{i=1}^j, \{\{q_i^{t'}\}_{t'=0}^{\kappa-1}\}_{i=1}^j, t$

1: Define $\mathbf{x}^* \in \mathbb{R}^{\ell_\lambda}$ as $x_\ell^* = \langle \boldsymbol{\ell}^*, \mathbf{w}_\ell \rangle$.

2: Define $\hat{\mathbf{y}} \in \mathbb{R}^{\ell_\lambda}$ as $\hat{y}_\ell = \lambda c_{L17} \sum_{t'=0}^{\frac{1}{\lambda c_{L17}}-1} \sum_{k=1}^{j} \frac{\langle \mathbf{v}_k, \mathbf{w}_\ell \rangle}{\lambda_\ell} q_k^{t'}$      $\triangleright \hat{y}_\ell \sim x_\ell^* + N(0, c_{L17}\frac{\lambda}{\lambda_\ell})$ by Lemma 18 via Lemma 10

3: $\mathbf{z}(\mathbf{y}) \leftarrow \mathbb{E}[\mathbf{x}^* \mid \hat{\mathbf{y}} = \mathbf{y}]$

4: $f \leftarrow$ function from Lemma 8 for $\mathbf{z}(\hat{\mathbf{y}})$ with $\epsilon = \frac{\epsilon_d c_d}{4}$.      $\triangleright f$ exists by Lemma 17 via Lemma 10

5: $\Psi \leftarrow \text{Bernoulli}(f(\mathbf{z}(\hat{\mathbf{y}})))$

6: Set $\mathbf{A}^{(t)} = \text{Exploit}(\Psi, \mathbf{w}_{\ell_\lambda+1})$

7: $R \leftarrow \begin{cases} r^{(t)} & \text{w.p. } \frac{\epsilon_d c_d}{16(K\sqrt{\pi}+1)f(\mathbf{z}(\hat{\mathbf{y}}))} \text{ if } \Psi = 1 \\ N(0,1) & \text{otherwise} \end{cases}$      $\triangleright$ The above equation involves valid

    probabilities by Lemma 8 and because $\max(\|\mathbb{E}[\mathbf{z}(\hat{\mathbf{y}})]\|, 1) \le \|\mathbb{E}[\mathbf{x}^*]\| + 1 \le K\sqrt{\pi} + 1$

8: $\mathbf{a} \leftarrow \text{Exploit}(\mathbb{1}_{R>0}, \mathbf{w}_{\ell_\lambda+1})$.

9: **return** $\mathbf{a}, t+1$

---

**Algorithm 6** ExponentialGrowth

**Input:** $\mathbf{a}, \{\mathbf{w}_i\}_{i=1}^{\ell_\lambda}, \{\lambda_i\}_{i=1}^{\ell_\lambda}, \{\mathbf{v}_i\}_{i=1}^j, \{\{q_i^{t'}\}_{t'=0}^{\kappa-1}\}_{i=1}^j, t$

1: $S \leftarrow \text{Span}(\mathbf{w}_1, ..., \mathbf{w}_{\ell_\lambda})$

2: $\mathbf{x} \leftarrow \frac{\mathcal{P}_{S^\perp}(\mathbf{a})}{\|\mathcal{P}_{S^\perp}(\mathbf{a})\|}$

3: $c_k \leftarrow \sum_{i=1}^{\ell_\lambda} \frac{\langle \mathcal{P}_S(\mathbf{a}), \mathbf{w}_i \rangle \langle \mathbf{v}_k, \mathbf{w}_i \rangle}{\lambda_i}$ for $k \in [1:j]$      $\triangleright \mathcal{P}_S(\mathbf{a}) = \sum_{k=1}^j c_k \mathbf{v}_k$ by Lemma 18

4: $L \leftarrow \frac{4d(\mathbb{E}[\boldsymbol{\ell}^*_1]+1)^2(1+\sum_{k=1}^j c_k^2)}{c_{L15}^2}$

5: For $t' \in [t, t+L)$, set $\mathbf{A}^{(t')} = \mathbf{a}$

6: $t \leftarrow t + L$

7: $R \leftarrow \sum_{t'=t}^{t+L-1} \left(r^{(t')} - \sum_{k=1}^j c_k q_k^{t'}\right)$

8: $\mathbf{b} \leftarrow \text{Exploit}(\mathbb{1}_{R>0}, \mathbf{w}_{\ell_\lambda+1})$

9: **return** $\mathbf{b}, t$

---

### 3.2 Propositions and Theorem

As discussed in Section 2.2, the main purpose of Algorithm 5 is to find a BIC action $\mathbf{a}$ that has sufficiently high magnitude when projected onto the space of not-yet-sufficiently explored actions. This is formalized in Proposition 11. As discussed in Section 2.3, the main purpose of Algorithm 6 is to double the magnitude of $\mathbf{a}$ when projected onto the space of not-yet-sufficiently explored actions. This is formalized in Proposition 12.

**Proposition 11** (Proof in Appendix I). *The action $\mathbf{a}$ returned by Algorithm 5 satisfies*

$$\|\mathcal{P}_{S^\perp}(\mathbf{a})\| \ge \frac{c_{L14} \, c_v^{2.5} \, \epsilon_d c_d}{16(K\sqrt{\pi}+1)} := c_{P11} \, c_v^{2.5} \epsilon_d c_d.$$

**Proposition 12** (Proof in Appendix J). *The action returned by Algorithm 6 satisfies*

$$\|\mathcal{P}_{S^\perp}(\mathbf{b})\| \ge 2 \|\mathcal{P}_{S^\perp}(\mathbf{a})\|.$$

We can now state our main theorem bounding the sample complexity of Algorithm 4.

**Theorem 13** (Proof in Appendix K). *Algorithm 4 is BIC and has sample complexity*

$$O\left(\log\left(\frac{1}{c_v \, \epsilon_d c_d}\right)\left(\frac{d^5}{\lambda^4 \, c_v^2} + \frac{d^4}{c_d^2 \lambda^4}\right)\right).$$

As discussed above, for any $\bar{\lambda}$, we can repeat Algorithm 4 for $\lceil \bar{\lambda}/\lambda \rceil$ times to get $\bar{\lambda}$-spectral exploration. This is because trace is additive, and therefore if running Algorithm 4 once gives $\lambda$-spectral exploration, then running it $\lceil \bar{\lambda}/\lambda \rceil$ times will give $\bar{\lambda}$-spectral exploration. Multiplying the bound from Theorem 13 by $\lceil \bar{\lambda}/\lambda \rceil$ gives that $\bar{\lambda}$-spectral exploration is achievable in $\bar{\lambda} \left(\frac{d}{c_v + c_d}\right)^{O(1)} \log(1/\epsilon_d)$ rounds, matching the desired result of Theorem 4.

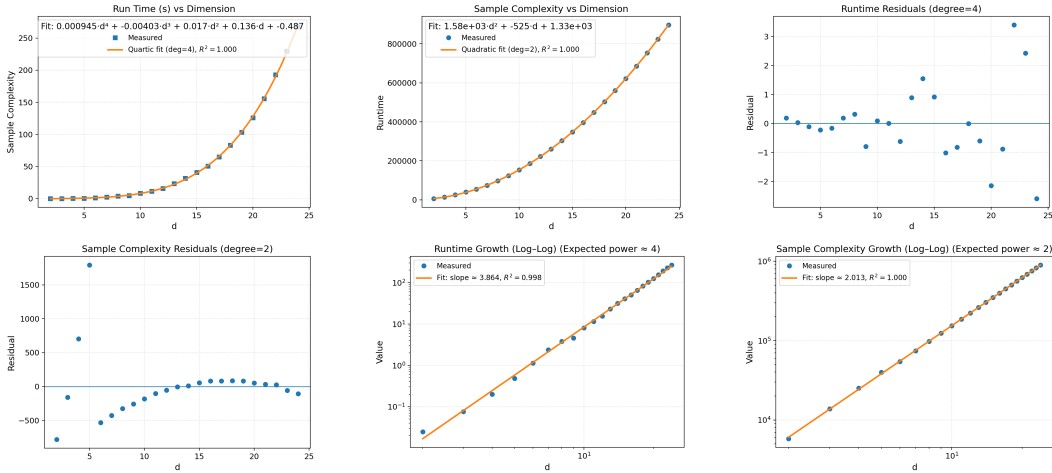

Figure 2: Summary of computational scaling. (a)–(b) show polynomial fits supporting quartic runtime and quadratic sample complexity; (c)–(d) residuals indicate good model adequacy; (e)–(f) log–log plots corroborate slopes near 4 and 2, respectively.

## 4 Experimental Results

Finally, we conclude with a simple experiment validating the practicality of the proposed algorithm. We implemented Algorithm 4 and tested on synthetic data (Figure 2). Our experiments focus on the setting where the prior distribution is a $d$-dimensional Gaussian that is independent across all dimensions and has mean $0.1$ in the first dimension and mean $0$ in all other dimensions. Note that our algorithm can be applied to arbitrary prior distributions, however we ran experiments for the independent case as this simplifies the code significantly. For this setting, Assumption 3 holds for $\epsilon_d = 0.1$, $c_d = 1$, $K = 1$, and $c_v = 1$. The constants in our algorithm are certainly not optimal for this specific instance, yet these constants are what allows our theoretical results to hold for any prior distribution. We ran our algorithm for values of $d$ ranging from $d = 2$ to $d = 24$ and tracked the number of samples necessary to achieve $\lambda$-spectral exploration. The results show a quadratic dependence of the sample complexity on dimension, and quartic scaling for running time. Therefore, in practice the number of steps grows polynomially in $d$ at a much better rate than the worst-case bound in our theoretical results. One reason for this is that a factor of $d^4$ in our bound comes from Lemma 9, which is a worst-case bound on how much exploration we gain in each step due to subtleties of high-dimensional geometry. Experiments were run on an XPS 13 with an Intel Core i7.

## 5 Discussion

We conclude with a more detailed discussion on how our results can be combined with the results of Sellke [2023] to achieve end-to-end guarantees for incentivized exploration. Here we focus on $r$-regular $\mu$ as assumed in that work, which is encapsulated by Proposition 5. Recall [Sellke, 2023, Theorem 3.5] shows that for $\epsilon > 0$, if an algorithm has already achieved $\tilde{O}(d^4/r^2\epsilon^2)$-spectral exploration at time $t$, then running Thompson sampling from time $t$ onward will be $\epsilon$-BIC (where $\epsilon$-BIC relaxes Definition 1 by subtracting an $\epsilon$ term from the right-hand side). Theorem 4 efficiently achieves the necessary spectral exploration, with at most poly$(d, 1/r, 1/\epsilon)$ sample complexity (and thus additional regret). Note that our algorithm actually gives a stronger guarantee than in Sellke [2023] (BIC rather than $\epsilon$-BIC). If we only need to guarantee the initial exploration is $\epsilon$-BIC, then we no longer need the InitialExploration phase of the algorithm, and therefore can drop Assumption 1. Combining our result with Sellke [2023] and the analysis of Thompson sampling in Dong and Van Roy [2018] or Agrawal and Goyal [2013], we therefore obtain an end-to-end $\epsilon$-BIC algorithm with respectively Bayesian regret poly$(d, 1/r, 1/\epsilon) + \tilde{O}(d\sqrt{T})$ or frequentist regret poly$(d, 1/r, 1/\epsilon) + \tilde{O}(d^{3/2}\sqrt{T})$. Namely, one first runs our algorithm for poly$(d, 1/r, 1/\epsilon)$ steps to guarantee the required spectral exploration, and then uses Thompson Sampling for all remaining steps. Since the regret from the initial exploration phase is constant relative to $T$ (and polynomial in $d$), this combined algorithm will asymptotically obey the state-of-the-art regret bounds for Thompson Sampling.

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

# A  Technical Lemmas

In this appendix, we introduce a series of technical lemmas that form the basis of our results. Our proofs rely on carefully analyzing how the conditional expectation of $\ell^*$ changes when conditioning on different signals $\psi$. Lemmas 14-17 are the main tools we use to analyze the behavior of these conditional expectations.

Lemma 14 says that for random variable $X$, if we have some signal $R$ that is equal to $X$ plus noise with some small probability and is just pure noise otherwise, then the conditional expectation of $X$ given the sign of $R$ has magnitude that is $\Omega(\epsilon)$. In the "initial exploration" phase of our algorithm we explore a new (not previously explored) direction with very small probability. Lemma 14 implies that this exploration will lead to the conditional expectation of $\ell^*$ in the newly-explored direction having magnitude proportional to the probability of exploration.

**Lemma 14.** *Suppose $X$ is a real-valued $K$-sub-gaussian random variable such that $\mathbb{E}[X] = 0$, $\mathbb{E}[X^2] = \sigma_X^2$. Let $R \sim X + N(0,1)$ with probability $\epsilon$ and $R \sim N(0,1)$ with probability $1 - \epsilon$. Then there exists $c_{L14}$ independent of $X$ such that*

$$|\mathbb{E}[X \mid 1_{R>0}]| \geq c_{L14}\sigma_X^5\epsilon.$$

The proof of Lemma 14 can be found in Appendix H.1.

Lemmas 15 and 16 are the main technical tools that allow for us to exponentially grow the amount of exploration in any new direction. Informally, Lemma 15 says that even if $X$ forms only an $\epsilon$ fraction of the signal $r$, conditioning on the sign of $r$ will increase the conditional expectation of $X$ by a multiplicative factor. This lemma will be applied to the expectation of $\ell^*$ in the new direction we are trying to explore. Lemma 16 says that any random variable conditioned on the sign of $r$ cannot have conditional expectation increase by more than $O(\epsilon)$. This will be applied to the expectation of $\ell^*$ in all of the directions that we have already explored. These two lemmas combined allow our algorithm to multiplicatively increase the magnitude of the expectation of $\ell^*$ in an unexplored direction *relative* to the already explored directions.

**Lemma 15.** *Let $X$ be a $K$-sub-gaussian random variable satisfying $\mathbb{E}[X] = 0$ and $\mathbb{E}[X^2] = \sigma_X^2 \geq c_v$. For $Z \sim N(0, \sigma^2)$ such that $Z \perp\!\!\!\perp X$ and $\epsilon > 0$, define $r = \epsilon X + Z$. Then there exists a constant $\delta_{L15}$ such that if $\epsilon/\sigma \leq \delta_{L15}$,*

$$|\mathbb{E}[X \mid 1_{r>0}]| \geq \frac{\epsilon\sigma_X^2}{2\sigma\sqrt{2\pi}} \geq \frac{c_v\,\epsilon}{\sqrt{8\pi}\sigma} := \frac{c_{L15}\epsilon}{\sigma}.$$

The proof of Lemma 15 can be found in Appendix G.1.

**Lemma 16.** *For $K$-sub-gaussian random variables $X, Y$ such that $\mathbb{E}[X] = \mathbb{E}[Y] = 0$ and for $Z \sim N(0, \sigma^2)$ independent of $X$ and $Y$, let $r \sim \epsilon X + Z$. Suppose $\frac{\epsilon}{\sigma} \leq \delta_{L16} := \min\left(1, \frac{1}{2K\sqrt{\log(2)}}\right)$. Then there exists a constant $c_{L16} > 0$ such that*

$$|\mathbb{E}[Y \mid 1_{r>0}]| \leq c_{L16}\epsilon/\sigma.$$

The proof of Lemma 16 can be found in Appendix G.2.

Lemma 17 is a more technical lemma that allows us to better understand the distribution of $\ell^*$ when we condition on averages based on previous rewards. More specifically, this allows us to apply Lemma 8 to the random variable $\mathbf{z}$ as described in Section 2.2. The proof of Lemma 17 can be found in Appendix H.

**Lemma 17.** *For random variable $\mathbf{X}$ in $\mathbb{R}^d$, define $\mathbf{Y} = \mathbf{X} + \mathbf{W}$ where $\mathbf{W} \sim N(\mathbf{0}, \mathrm{Diag}(\mathbf{s}))$ and $\mathbf{W}$ is independent of $\mathbf{X}$. Define $\mathbf{Z}(\mathbf{Y}) = \mathbb{E}[\mathbf{X} \mid \mathbf{Y}]$. If $\min_{\|\mathbf{v}\|=1} \Pr\left(\langle \mathbf{X}, \mathbf{v}\rangle \geq c_d\right) \geq \epsilon_d$ and for all $i \in [1:d]$, $s_i \leq c_{L17} := \frac{c_d^2/32}{\log(4/\epsilon_d)}$ then*

$$\min_{\|\mathbf{v}\|=1} \mathbb{E}[(\langle \mathbf{Z}(\mathbf{Y}), \mathbf{v}\rangle)^+] \geq \frac{\epsilon_d c_d}{4}.$$

The final lemma for this section says that any vector in the span of the top eigenvectors of a positive semi-definite matrix can be represented as a linear combination of these top eigenvectors with coefficients that are not too large. The proof of Lemma 18 can be found in Appendix F.

**Lemma 18.** *Let $\mathbf{v}_1, \mathbf{v}_2, ...\mathbf{v}_j \in \mathbb{R}^d$ such that $\|\mathbf{v}_i\| = 1$. Define $\mathbf{M} = \sum_{i=1}^{j} \mathbf{v}_i^{\otimes 2}$. Suppose $\mathbf{w}_1, ..., \mathbf{w}_d$ are orthonormal eigenvectors of $\mathbf{M}$ with corresponding eigenvalues $\lambda_1 \geq ... \geq \lambda_d \geq 0$. Suppose $\lambda_\ell \geq \epsilon$. Then for any $\mathbf{u} \in \text{Span}(\mathbf{w}_1, ..., \mathbf{w}_\ell)$ such that $\|\mathbf{u}\| \leq 1$, we have $\mathbf{u} = \sum_{i=1}^{j} c_i \mathbf{v}_i$, where $c_i := \sum_{k=1}^{\ell} \left( \frac{\langle \mathbf{u}, \mathbf{w}_k \rangle \langle \mathbf{v}_i, \mathbf{w}_k \rangle}{\lambda_k} \right)$. Furthermore, $\sum_{i=1}^{j} c_i^2 \leq \frac{1}{\epsilon}$.*

We also note that any sub-gaussian random variable $X$ satisfying $\mathbb{P}(|X| > t) \leq 2e^{-t^2/K^2}$ for all $t \geq 0$ (as in Condition 3) satisfies the following bounds on the moments of $X$:

$$\mathbb{E}[X] \leq \mathbb{E}[|X|] = \int_0^\infty \mathbb{P}(|X| > t)\, dt \leq \int_0^\infty 2e^{-t^2/K^2}\, dt = K\sqrt{\pi}, \tag{2}$$

$$\mathbb{E}[X^2] = \int_0^\infty \mathbb{P}(X^2 > t)\, dt \leq \int_0^\infty 2e^{-t/K^2}\, dt = 2K^2. \tag{3}$$

# B Discussion on Assumptions

Here we illustrate the importance of Assumption 3 by presenting a series of propositions lower bounding the number of samples needed in the worst-case for 1-spectral exploration in terms of the different parameters of Assumption 3.

**Lemma 19.** *There exist instances that require $\Omega(1/c_v)$ samples for 1-spectral exploration.*

*proof.* Consider the following example with $d = 2$ and where the coordinates of $\ell^*$ are independent (and assume that $c_d < 1, \epsilon_d < 1, c_v < 1$):

$$\ell_1^* = \begin{cases} -c_d & \text{w.p. } \epsilon_d \\ 1 & \text{w.p. } 1 - \epsilon_d \end{cases}, \qquad \ell_2^* = \begin{cases} -c_v & \text{w.p. } 1/2 \\ c_v & \text{w.p. } 1/2 \end{cases}$$

Note that if $\mathbb{E}[\ell_1^* \mid \psi] = 1 - O(\epsilon_d) \geq 0.5$ for sufficiently small $\epsilon_d$, then no optimal action will ever put weight more than $O(c_v)$ on the second coordinate because the magnitude of $\ell_2^*$ is bounded by $c_v$. This implies that we must need $1/c_v$ steps in order to guarantee 1-spectral exploration. $\square$

**Proposition 20.** *There exist instances that require $\Omega(c_d)$ samples for 1-spectral exploration.*

*proof.* Consider the following example

$$\ell_1^* = \begin{cases} -c_d & \text{w.p. } \epsilon_d \\ 2c_d & \text{w.p. } 2\epsilon_d \\ 1 & \text{w.p. } 1 - 3\epsilon_d \end{cases}, \qquad \ell_2^* = \begin{cases} -c_v & \text{w.p. } 1/2 \\ c_v & \text{w.p. } 1/2 \end{cases}$$

Once again, the first action must be $e_1$. Furthermore, in this case we need $O(poly(1/c_d))$ actions of $e_1$ in order to decrease the conditional expectation of $\ell_1^*$ to be 0, as we need this many samples to be able to effectively distinguish between $\ell_1^* = -c_d$ and $\ell_1^* = 2c_d$. This implies that we require $O(poly(1/c_d))$ samples to explore the second dimension in this example. $\square$

**Lemma 21.** *Let $L = \pm c$ be uniformly random for some $|c| \leq 1$. Suppose we receive noisy observations $r_i = s_i L + Z_i$ for a sequence $r_1, ...$ that is adapted to the filtration $\mathcal{F}_t$ generated by $(s_1, r_1, s_2, r_2, ..., s_t, r_t)$. (I.e. the signal strengths $s_i$ may depend on the past.) Let $T_t = \sum_{i=1}^{t} s_i^2$. Then the expected information gain on $L$ is at most $O(\mathbb{E}[T_t])$.*

*proof.* This is a special case of observing Brownian motion with drift $L$ up to the random stopping time $T_t$. (Since observing $r_i = s_i L + Z_i$ is equivalent to observing Brownian motion with drift $L$ for time $s_i^2$, up to rescaling.) Let $Q_\pm(T_t)$ be the laws of Brownian motion with drifts $\pm c$ up to time $T_t$. By symmetry the expected information gain can be computed assuming $L = c$, and is bounded by

$$\mathbb{E}[KL(Q_+(T_t), [Q_+(T_t) + Q_-(T_t)]/2)]$$

$$\overset{\text{Convexity of KL}}{\leq} \underbrace{\mathbb{E}[KL(Q_+(T_t), Q_+(T_t))]}_{0} + \mathbb{E}[KL(Q_+(T_t), Q_-(T_t))]$$

$$\leq \mathbb{E}[T_t].$$

Here we used $\mathbb{E}[KL(Q_+(T_t)|Q_-(T_t))] = c^2\,\mathbb{E}[T_t] \leq \mathbb{E}[T_t]$ by Girsanov's theorem.

$\square$

**Corollary 21.1.** *In the setting of Lemma 21, let $C_t = \mathbb{E}[L|\mathcal{F}_t] = \mathbb{E}[L|(s_1, ..., s_t, r_1, ..., r_t)]$. Then $\mathbb{E}[C_t^2] \leq O(\mathbb{E}[T_t])$.*

*proof.* This follows from the previous lemma: the expected relative entropy between the prior and posterior distributions of $L$ is precisely the information gain. In turn, the relative entropy is a strictly convex and even function of $C_t$ which is $\Omega(C_t^2)$. $\square$

**Proposition 22.** *There exist instances that require $\Omega(\log(1/\epsilon_d))$ samples for $1$-spectral exploration.*

*Proof of Proposition 22.* This proof is more subtle and requires inductive control of the information gain on the 2nd coordinate.

We now return to the first example prior from the $c_v$ lower bound, and apply Corollary 21.1 to the observations of the 2nd coordinate. To make the application direct, whenever an action $(a_{1,t}, a_{2,t})$ is played, we replace the noisy reward observation with separate observations for both $(a_{1,t}, 0)$ and $(0, a_{2,t})$ (with half the noise level, which only affects constant factors in the argument). This gives strictly more information since the two separate observations can be added to recover the original observation. We let $C_t, \mathcal{F}_t$ from above correspond to observations in the second coordinate.

At each time $t$, by Jensen's inequality on the convex function $f(x) = (1/3 - x)_+$, we see that for any signal $\psi$:

$$P[\mathbb{E}[\ell_1|\psi] \leq 1/2] \leq O(\mathbb{E}[f(\mathbb{E}[\ell_1|\psi])]) \leq O(\mathbb{E}[f(\ell_1)]) \leq O(\epsilon_d).$$

On the main high-probability event that $\mathbb{E}[\ell_1|\psi] \geq 1/2$, the 2nd coordinate of $Exploit(\psi)$ has absolute value at most $O(|C_t|)$. Via the Lemma and Corollary above, it follows that

$$\mathbb{E}[T_{t+1}] - \mathbb{E}[T_t] \leq O(\mathbb{E}[T_t] + \epsilon_d).$$

Namely the case $\{\mathbb{E}[\ell_1|\psi] \leq 1/2\}$ contributes $O(\epsilon_d)$ while the remaining case contributes $O(\mathbb{E}[C_t^2]) \leq O(\mathbb{E}[T_t])$.

Since $T_0 = 0$, this implies by induction that

$$\mathbb{E}[T_t] \leq e^{O(t)}\epsilon_d.$$

Finally note that we need $T_t \geq 1$ for $1$-spectral exploration. Indeed, $1$-spectral exploration requires that the actions $a_1, \ldots, a_t$ satisfy

$$\sum_{i=1}^{t} \langle a_t^\top, Ma_t \rangle \geq Tr(M)$$

for all positive semi-definite matrices $M$, and taking $M = \begin{pmatrix} 0 & 0 \\ 0 & 1 \end{pmatrix}$ recovers the claim. In all this gives the desired $\log(1/\epsilon_d)$ lower bound.

$\square$

## C  Proof of Lemma 10

*Proof of Lemma 10.* Recall that the input parameters from Algorithm 5 come from their use in Algorithm 4. We compute as follows, with $\stackrel{d}{=}$ indicating equality in distribution.

$$\hat{y}_\ell = \lambda c_{L17} \sum_{t'=0}^{\frac{1}{\lambda c_{L17}}-1} \sum_{k=1}^{j} \frac{\langle \mathbf{v}_k, \mathbf{w}_\ell \rangle}{\lambda_\ell} q_k^{t'}$$

$$\stackrel{d}{=} \lambda c_{L17} \sum_{t'=0}^{\frac{1}{\lambda c_{L17}}-1} \sum_{k=1}^{j} \frac{\langle \mathbf{v}_k, \mathbf{w}_\ell \rangle}{\lambda_\ell} \langle \mathbf{v}_k, \boldsymbol{\ell}^* \rangle + \lambda c_{L17} \sum_{t'=0}^{\frac{1}{\lambda c_{L17}}-1} \sum_{k=1}^{j} \frac{\langle \mathbf{v}_k, \mathbf{w}_\ell \rangle}{\lambda_\ell} N(0,1)$$

$$\stackrel{d}{=} \left\langle \boldsymbol{\ell}^*, \left( \lambda c_{L17} \sum_{t'=0}^{\frac{1}{\lambda c_{L17}}-1} \sum_{k=1}^{j} \frac{\langle \mathbf{v}_k, \mathbf{w}_\ell \rangle}{\lambda_\ell} \mathbf{v}_k \right) \right\rangle + N\left( 0, \lambda c_{L17} \sum_{k=1}^{j} \frac{\langle \mathbf{v}_k, \mathbf{w}_\ell \rangle^2}{\lambda_\ell^2} \right)$$

$$\stackrel{d}{=} \left\langle \boldsymbol{\ell}^*, \left( \lambda c_{L17} \sum_{t'=0}^{\frac{1}{\lambda c_{L17}}-1} \mathbf{w}_\ell \right) \right\rangle + N\left( 0, \frac{\lambda c_{L17}}{\lambda_\ell^2} \mathbf{w}_\ell^\top \mathbf{M} \mathbf{w}_\ell \right) \qquad \text{[Lemma 18 } (\mathbf{u} = \mathbf{w}_\ell)]$$

$$\stackrel{d}{=} x_\ell^* + N\left( 0, c_{L17} \frac{\lambda}{\lambda_\ell} \right).$$

We will apply Lemma 17 with $\mathbf{X} = \mathbf{x}^*$, $\mathbf{Y} = \hat{\mathbf{y}}$, and $\mathbf{Z}(\mathbf{Y}) = \mathbf{z}(\hat{\mathbf{y}})$. As shown above, (and using that $\frac{\lambda}{\lambda_\ell} \leq 1$ for all $\ell \leq \ell_\lambda$) we have that $\mathbf{Y} - \mathbf{X}$ has the appropriate distribution. The last thing we need to show is that

$$\min_{\|\mathbf{q}\|=1} \Pr\left( \langle \mathbf{X}, \mathbf{q} \rangle \geq c_d \right) = \min_{\|\mathbf{q}\|=1} \Pr\left( \sum_{\ell=1}^{\ell_\lambda} \langle \boldsymbol{\ell}^*, \mathbf{w}_\ell \rangle q_\ell \geq c_d \right)$$

$$= \min_{\|\mathbf{q}\|=1} \Pr\left( \left\langle \boldsymbol{\ell}^*, \left( \sum_{\ell=1}^{\ell_\lambda} q_\ell \mathbf{w}_\ell \right) \right\rangle \geq c_d \right)$$

$$\geq \min_{\|\mathbf{v}\|=1} \Pr\left( \langle \boldsymbol{\ell}^*, \mathbf{v} \rangle \geq c_d \right)$$

$$\geq \epsilon_d. \qquad \text{[Assumption 1]}.$$

This means we can apply Lemma 17 to get that

$$\min_{\|\mathbf{v}\|=1} \mathbb{E}[\langle \mathbf{z}(\hat{\mathbf{y}}), \mathbf{v} \rangle^+] \geq \frac{\epsilon_d c_d}{4}.$$

We have therefore shown that $\mathbf{z}(\hat{\mathbf{y}})$ satisfies the assumptions of Lemma 8 with $\epsilon = \frac{\epsilon_d c_d}{4}$. $\qquad \square$

## D  Proof of Lemma 7

*Proof of Lemma 7.* The BIC optimal action given $\psi$ is

$$\mathbf{A}^* = \arg\max_{\mathbf{A} \in S^{d-1}} \mathbb{E}[\langle \mathbf{A}, \boldsymbol{\ell}^* \rangle \mid \psi]$$

$$= \arg\max_{\mathbf{A} \in S^{d-1}} \langle \mathbf{A}, \mathbb{E}[\boldsymbol{\ell}^* \mid \psi] \rangle.$$

Therefore, the BIC action is $\mathbf{A}^* = \frac{\mathbb{E}[\boldsymbol{\ell}^* | \psi]}{\|\mathbb{E}[\boldsymbol{\ell}^* | \psi]\|}$ if $\|\mathbb{E}[\boldsymbol{\ell}^* \mid \psi]\| \neq 0$. If $\|\mathbb{E}[\boldsymbol{\ell}^* \mid \psi]\| = 0$, then any action is BIC including $\mathbf{v}$. $\qquad \square$

## E  Proof of Lemma 9

We will prove the following equivalent lemma.

**Lemma 23.** *In the setting of Lemma 9,*

$$\sum_{i=1}^{\ell} \lambda_i' \leq (1 - \epsilon/2) + \sum_{i=1}^{\ell} \lambda_i$$

We first observe that Lemma 23 implies the desired Lemma 9.

*Proof of Lemma 9.* By linearity of trace,

$$\sum_{i=1}^{d} \lambda_i' = 1 + \sum_{i=1}^{d} \lambda_i.$$

Therefore, Lemma 23 implies the desired result that

$$\sum_{i=\ell+1}^{d} \lambda_i' \geq \epsilon/2 + \sum_{i=\ell+1}^{d} \lambda_i. \qquad \square$$

*Proof of Lemma 23.* First, note the following, where the max is over $\mathbf{x}_1, ..., \mathbf{x}_\ell$ that are orthonormal.

$$\sum_{i=1}^{\ell} \lambda_i' = \max_{\mathbf{x}_1, ..., \mathbf{x}_\ell} \sum_{i=1}^{\ell} \mathbf{x}_i^T \mathbf{M}' \mathbf{x}_i = \max_{\mathbf{x}_1, ..., \mathbf{x}_\ell} \left( \sum_{i=1}^{\ell} \mathbf{x}_i^\top \mathbf{M} \mathbf{x}_i + \sum_{i=1}^{\ell} \langle \mathbf{x}_i, \mathbf{u} \rangle^2 \right).$$

Define

$$\mathbf{x}_1^*, ..., \mathbf{x}_\ell^* = \arg\max_{\mathbf{x}_1, ..., \mathbf{x}_\ell} \sum_{i=1}^{\ell} \mathbf{x}_i^\top \mathbf{M}' \mathbf{x}_i. \tag{5}$$

We will now prove (by contradiction) that for all $i \leq \ell$, $\|\mathcal{P}_{S^\perp}(\mathbf{x}_i^*)\|_2^2 \leq \frac{\epsilon^2}{100d^2}$. Suppose that there exists some $i' \in 1, ..., \ell$ such that $\|\mathcal{P}_{S^\perp}(\mathbf{x}_{i'}^*)\|_2^2 > \frac{\epsilon^2}{100d^2}$. Then we find

$$\sum_{i=1}^{\ell} (\mathbf{x}_i^*)^\top \mathbf{M}' \mathbf{x}_i^* = \sum_{i=1}^{\ell} (\mathbf{x}_i^*)^\top \mathbf{M} \mathbf{x}_i^* + \sum_{i=1}^{\ell} \langle \mathbf{x}_i^*, \mathbf{u} \rangle^2$$

$$\leq 1 + \sum_{i=1}^{\ell} (\mathbf{x}_i^*)^\top \mathbf{M} \mathbf{x}_i^* \qquad \left[ \mathbf{x}_i^* \text{ orthonormal so } \sum_{i=1}^{\ell} \langle \mathbf{x}_i^*, \mathbf{u} \rangle^2 \leq \|\mathbf{u}\|^2 \leq 1 \right]$$

$$= 1 + \sum_{i=1}^{\ell} \left( \mathcal{P}_{S^\perp}(\mathbf{x}_i^*)^\top \mathbf{M} \mathcal{P}_{S^\perp}(\mathbf{x}_i^*) + \mathcal{P}_S(\mathbf{x}_i^*)^\top \mathbf{M} \mathcal{P}_S(\mathbf{x}_i^*) \right)$$

$$= 1 + \sum_{i=1}^{\ell} \mathcal{P}_{S^\perp}(\mathbf{x}_i^*)^\top \mathbf{M} \mathcal{P}_{S^\perp}(\mathbf{x}_i^*) + \sum_{i=1}^{\ell} \mathcal{P}_S(\mathbf{x}_i^*)^\top \mathbf{M} \mathcal{P}_S(\mathbf{x}_i^*)$$

$$= 1 + d\epsilon + \sum_{i=1}^{\ell} \mathcal{P}_S(\mathbf{x}_i^*)^\top \mathbf{M} \mathcal{P}_S(\mathbf{x}_i^*) \qquad [S^\perp = \text{span of evectors with evalues} \leq \epsilon]$$

$$= 1 + d\epsilon + \sum_{i=1}^{\ell} \left( \sum_{k=1}^{\ell_\epsilon} \langle \mathbf{x}_i^*, \mathbf{w}_k \rangle \mathbf{w}_k \right)^\top \mathbf{M} \left( \sum_{k=1}^{\ell_\epsilon} \langle \mathbf{x}_i^*, \mathbf{w}_k \rangle \mathbf{w}_k \right)$$

$$= 1 + d\epsilon + \sum_{i=1}^{\ell} \sum_{k=1}^{\ell_\epsilon} \langle \mathbf{x}_i^*, \mathbf{w}_k \rangle^2 \mathbf{w}_k^\top \mathbf{M} \mathbf{w}_k$$

$$= 1 + d\epsilon + \sum_{i=1}^{\ell} \sum_{k=1}^{\ell_\epsilon} \langle \mathbf{x}_i^*, \mathbf{w}_k \rangle^2 \lambda_k. \tag{6}$$

Because $\mathbf{x}_i^*$ are orthonormal, we know that $\sum_{i=1}^{\ell}\langle \mathbf{x}_i^*, \mathbf{w}_k\rangle^2 \leq \|\mathbf{w}_k\|_2^2 = 1$. Because we assumed that $\|\mathcal{P}_{S^\perp}(\mathbf{x}_{i'}^*)\|_2^2 > \frac{\epsilon^2}{100d^2}$, we know that $\sum_{i=1}^{\ell}\sum_{k=1}^{\ell_\epsilon}\langle \mathbf{x}_i^*, \mathbf{w}_k\rangle^2 = \sum_{i=1}^{\ell}\|\mathcal{P}_S(\mathbf{x}_i^*)\|^2 \leq \ell - \frac{\epsilon^2}{100d^2}$. Combining these two statements with the fact that $\lambda_k$ is decreasing in $k$,

$$\sum_{i=1}^{\ell}\sum_{k=1}^{\ell_\epsilon}\langle \mathbf{x}_i^*, \mathbf{w}_k\rangle^2 \lambda_k = \sum_{k=1}^{\ell_\epsilon}\sum_{i=1}^{\ell}\langle \mathbf{x}_i^*, \mathbf{w}_k\rangle^2 \lambda_k \leq \left(1 - \frac{\epsilon^2}{100d^2}\right)\lambda_\ell + \sum_{i=1}^{\ell-1}\lambda_i.$$

Continuing where we left off with Equation (6), we have that

$$= 1 + d\epsilon + \sum_{i=1}^{\ell}\sum_{k=1}^{\ell_\epsilon}\langle \mathbf{x}_i^*, \mathbf{w}_k\rangle^2 \lambda_k \leq 1 + d\epsilon + \left(1 - \frac{\epsilon^2}{100d^2}\right)\lambda_\ell + \sum_{i=1}^{\ell-1}\lambda_i$$

$$\leq 1 + d\epsilon - \frac{\epsilon^2}{100d^2}\lambda_\ell + \sum_{i=1}^{\ell}\lambda_i$$

$$\leq 1 + d\epsilon - 2d + \sum_{i=1}^{\ell}\lambda_i \qquad\qquad [\lambda_\ell \geq \frac{200d^3}{\epsilon^2}]$$

$$< \sum_{i=1}^{\ell}\lambda_i \qquad\qquad [\epsilon < 1]$$

$$= \sum_{i=1}^{\ell}\mathbf{w}_i^\top \mathbf{M}\mathbf{w}_i \leq \sum_{i=1}^{\ell}\mathbf{w}_i^\top \mathbf{M}'\mathbf{w}_i.$$

Therefore, we have a contradiction, as $\mathbf{x}_1^*, ..., \mathbf{x}_\ell^*$ cannot be a solution to Equation (5) because these vectors are strictly beaten by $\mathbf{w}_1, ..., \mathbf{w}_\ell$.

Therefore, we have shown that $\|\mathcal{P}_{S^\perp}(\mathbf{x}_i^*)\|_2^2 \leq \frac{\epsilon^2}{100d^2}$ for all $i$.

In the following equation, we define $P_S$ as the projection matrix for projecting a vector onto $S$. Now, we have that

$$\sum_{i=1}^{\ell}\lambda_i' = \sum_{i=1}^{\ell}(\mathbf{x}_i^*)^\top \mathbf{M}\mathbf{x}_i^* + \sum_{i=1}^{\ell}\langle \mathbf{x}_i^*, \mathbf{u}\rangle^2$$

$$\leq \sum_{i=1}^{\ell}\lambda_i + \sum_{i=1}^{\ell}\langle \mathbf{x}_i^*, \mathbf{u}\rangle^2$$

$$\leq \sum_{i=1}^{\ell}\lambda_i + \sum_{i=1}^{\ell}\left(\langle \mathcal{P}_S(\mathbf{x}_i^*), \mathcal{P}_S(\mathbf{u})\rangle + \langle \mathcal{P}_{S^\perp}(\mathbf{x}_i^*), \mathcal{P}_{S^\perp}(\mathbf{u})\rangle\right)^2$$

$$\leq \sum_{i=1}^{\ell}\lambda_i + \sum_{i=1}^{\ell}(|\langle \mathcal{P}_S(\mathbf{x}_i^*), \mathcal{P}_S(\mathbf{u})\rangle| + \frac{\epsilon}{10d})^2 \qquad [\|\mathcal{P}_{S^\perp}(\mathbf{x}_i^*)\|_2^2 \leq \frac{\epsilon^2}{100d^2}]$$

$$= \sum_{i=1}^{\ell}\lambda_i + \sum_{i=1}^{\ell}\left(\langle \mathcal{P}_S(\mathbf{x}_i^*), \mathcal{P}_S(\mathbf{u})\rangle^2 + \frac{\epsilon}{5d}|\langle \mathcal{P}_S(\mathbf{x}_i^*), \mathcal{P}_S(\mathbf{u})\rangle| + \frac{\epsilon^2}{100d^2}\right)$$

$$\leq \sum_{i=1}^{\ell}\lambda_i + \sum_{i=1}^{\ell}\left(\langle \mathcal{P}_S(\mathbf{x}_i^*), \mathcal{P}_S(\mathbf{u})\rangle^2 + \frac{\epsilon}{5d} + \frac{\epsilon^2}{100d^2}\right)$$

$$\leq \sum_{i=1}^{\ell}\lambda_i + \frac{\epsilon}{5} + \frac{\epsilon^2}{100d} + \sum_{i=1}^{\ell}\langle \mathcal{P}_S(\mathbf{x}_i^*), \mathcal{P}_S(\mathbf{u})\rangle^2$$

$$= \sum_{i=1}^{\ell}\lambda_i + \frac{\epsilon}{5} + \frac{\epsilon^2}{100d} + \sum_{i=1}^{\ell}\left((\mathbf{x}_i^*)^\top P_S^\top P_S \mathbf{u}\right)^2$$

$$= \sum_{i=1}^{\ell}\lambda_i + \frac{\epsilon}{5} + \frac{\epsilon^2}{100d} + \sum_{i=1}^{\ell}\left((\mathbf{x}_i^*)^\top P_S \mathbf{u}\right)^2$$

$$= \sum_{i=1}^{\ell} \lambda_i + \frac{\epsilon}{5} + \frac{\epsilon^2}{100d} + \sum_{i=1}^{\ell} \langle \mathbf{x}_i^*, \mathcal{P}_S(\mathbf{u}) \rangle^2$$

$$\leq \sum_{i=1}^{\ell} \lambda_i + \frac{\epsilon}{5} + \frac{\epsilon^2}{100d} + \|\mathcal{P}_S(\mathbf{u})\|_2^2$$

$$\leq \sum_{i=1}^{\ell} \lambda_i + \frac{\epsilon}{5} + \frac{\epsilon^2}{100d} + 1 - \epsilon$$

$$\leq \sum_{i=1}^{\ell} \lambda_i + (1 - \epsilon/2).$$

This completes the proof of Lemma 23. $\qquad\square$

## F  Proof of Lemma 18

*Proof of Lemma 18.* Define $\mathbf{A} \in \mathbb{R}^{j \times d}$ with rows corresponding to $\mathbf{v}_1, ..., \mathbf{v}_j$. Then $\mathbf{M} = \mathbf{A}^\top \mathbf{A}$. We want to write $\mathbf{u} = \mathbf{A}^\top \mathbf{c}$ for $\mathbf{c} \in \mathbb{R}^j$.

Because $\mathbf{w}_1, ..., \mathbf{w}_\ell$ are orthonormal and $\mathbf{u} \in \mathrm{Span}(\mathbf{w}_1, ..., \mathbf{w}_\ell)$, we have that

$$\mathbf{u} = \sum_{i=1}^{\ell} \langle \mathbf{u}, \mathbf{w}_i \rangle \mathbf{w}_i$$

Because $\mathbf{w}_i$ is an eigenvector of $\mathbf{M}$ with eigenvalue $\lambda_i$, we know that for any $i \leq \ell$

$$\lambda_i \mathbf{w}_i = \mathbf{M} \mathbf{w}_i = \mathbf{A}^\top \mathbf{A} \mathbf{w}_i$$

Rearranging terms and multiplying both sides by $\langle \mathbf{u}, \mathbf{w}_i \rangle$, we have that for any $i \leq \ell$

$$\langle \mathbf{u}, \mathbf{w}_i \rangle \mathbf{w}_i = \mathbf{A}^\top \left( \frac{\langle \mathbf{u}, \mathbf{w}_i \rangle \mathbf{A} \mathbf{w}_i}{\lambda_i} \right).$$

Therefore, we have that

$$\mathbf{u} = \sum_{i=1}^{\ell} \langle \mathbf{u}, \mathbf{w}_i \rangle \mathbf{w}_i = \sum_{i=1}^{\ell} \mathbf{A}^\top \left( \frac{\langle \mathbf{u}, \mathbf{w}_i \rangle \mathbf{A} \mathbf{w}_i}{\lambda_i} \right) = \mathbf{A}^\top \sum_{i=1}^{\ell} \left( \frac{\langle \mathbf{u}, \mathbf{w}_i \rangle \mathbf{A} \mathbf{w}_i}{\lambda_i} \right).$$

Now, we can define

$$\mathbf{c} = \sum_{i=1}^{\ell} \left( \frac{\langle \mathbf{u}, \mathbf{w}_i \rangle \mathbf{A} \mathbf{w}_i}{\lambda_i} \right) = \mathbf{A} \sum_{i=1}^{\ell} \frac{\langle \mathbf{u}, \mathbf{w}_i \rangle \mathbf{w}_i}{\lambda_i}.$$

This implies that

$$\begin{aligned}
\|\mathbf{c}\|_2^2 &= \left\| \mathbf{A} \sum_{i=1}^{\ell} \frac{\langle \mathbf{u}, \mathbf{w}_i \rangle \mathbf{w}_i}{\lambda_i} \right\|_2^2 \\
&= \left( \sum_{i=1}^{\ell} \frac{\langle \mathbf{u}, \mathbf{w}_i \rangle \mathbf{w}_i}{\lambda_i} \right)^\top \mathbf{M} \left( \sum_{i=1}^{\ell} \frac{\langle \mathbf{u}, \mathbf{w}_i \rangle \mathbf{w}_i}{\lambda_i} \right) \\
&= \sum_{i=1}^{\ell} \lambda_i \left( \frac{\langle \mathbf{u}, \mathbf{w}_i \rangle}{\lambda_i} \right)^2 && [\mathbf{w}_i^\top \mathbf{M} \mathbf{w}_i = \lambda_i] \\
&= \sum_{i=1}^{\ell} \frac{\langle \mathbf{u}, \mathbf{w}_i \rangle^2}{\lambda_i} \\
&\leq \frac{\|\mathbf{u}\|_2^2}{\epsilon} \\
&\leq \frac{1}{\epsilon}.
\end{aligned}$$

This vector $\mathbf{c}$ therefore satisfies the desired properties. $\qquad\square$

# G  Proof of Lemma 8

We begin by proving the following lemma.

**Lemma 24.** *Let $\mu$ be a probability distribution on $\mathbb{R}^d$ with finite first moment and suppose*

$$\min_{\|\mathbf{v}\|=1} \mathbb{E}^{\mathbf{x}\sim\mu}[\langle \mathbf{v}, \mathbf{x}\rangle_+] \geq \epsilon.$$

*Then for any $\mathbf{w}$ with $\|\mathbf{w}\| < \epsilon$ there is a $[0,1]$-valued measurable function $f$ such that $\mathbb{E}[\mathbf{x}f(\mathbf{x})] = \mathbf{w}$.*

*Proof of Lemma 24.* Let $K$ be the set of possible vectors $\mathbb{E}[\mathbf{x}f(\mathbf{x})]$ where $f$ is a a $[0,1]$-valued measurable function. Then for any $\mathbf{a}, \mathbf{b} \in K$, there exist corresponding $[0,1]$-valued functions $f^a$ and $f^b$ such that $\mathbb{E}[\mathbf{x}f^a(\mathbf{x})] = \mathbf{a}$ and $\mathbb{E}[\mathbf{x}f^b(\mathbf{x})] = \mathbf{b}$. Therefore for any $t \in [0,1]$, $t\mathbf{a} + (1-t)\mathbf{b} \in K$ because $\mathbb{E}[\mathbf{x}f^t(\mathbf{x})] = t\mathbf{a} + (1-t)\mathbf{b}$ for $f^t(\mathbf{x}) = tf^a(\mathbf{x}) + (1-t)f^b(\mathbf{x})$. This implies that $K$ is convex.

We will now prove the desired result by contradiction. Suppose $\mathbf{w} \notin K$. Because $K$ is convex, if $\mathbf{w} \notin K$ then there is a "separating hyperplane" unit vector $\mathbf{v}$ such that

$$\sup_{\mathbf{u}\in K} \langle \mathbf{v}, \mathbf{u}\rangle \leq \langle \mathbf{v}, \mathbf{w}\rangle.$$

(Note that we do not argue here that $K$ is closed.) Because by assumption we have that $\|\mathbf{w}\| < \epsilon$, this implies that

$$\sup_{\mathbf{u}\in K} \langle \mathbf{v}, \mathbf{u}\rangle < \epsilon.$$

For any $\mathbf{v}$, by definition of $K$ and linearity of expectation we have that

$$\begin{aligned}
\sup_{\mathbf{u}\in K} \langle \mathbf{v}, \mathbf{u}\rangle &= \sup_{f:\mathbb{R}^d\to[0,1]} \langle \mathbf{v}, \mathbb{E}^{x\sim\mu}[\mathbf{x}f(\mathbf{x})]\rangle \\
&= \sup_{f:\mathbb{R}^d\to[0,1]} \mathbb{E}^{\mathbf{x}\sim\mu}[f(\mathbf{x})\langle \mathbf{v}, \mathbf{x}\rangle] \\
&= \mathbb{E}^{x\sim\mu}[\langle \mathbf{v}, \mathbf{x}\rangle_+] \\
&\geq \epsilon.
\end{aligned}$$

where the last line followed from the assumption of the lemma. This gives a contradiction, and therefore $\mathbf{w} \in K$ must be true. $\qquad\square$

*Proof of Lemma 8.* Applying the above lemma with $\mathbf{w} = \mathbf{0}$, there exists $f^0 : \mathbb{R}^d \to [0,1]$ such that $\mathbb{E}[\mathbf{x}f^0(\mathbf{x})] = \mathbf{0}$. Define the function $f'$ as $f'(\mathbf{x}) = \frac{f^0(\mathbf{x})+2\epsilon}{4\max(\|\mathbb{E}[\mathbf{x}]\|,1)}$. By this construction,

$$\|\mathbb{E}[\mathbf{x}f'(\mathbf{x})]\| = \left\| \frac{\mathbb{E}[\mathbf{x}f^0(\mathbf{x})] + 2\epsilon\mathbb{E}[\mathbf{x}]}{4\max(\|\mathbb{E}[\mathbf{x}]\|,1)} \right\| = \frac{2\epsilon\|\mathbb{E}[\mathbf{x}]\|}{4\max(\|\mathbb{E}[\mathbf{x}]\|,1)} \leq \frac{\epsilon}{2}.$$

Therefore, $\mathbf{w} := -\mathbb{E}[\mathbf{x}f'(\mathbf{x})]$ satisfies $\|\mathbf{w}\| < \epsilon$. Applying the above lemma again, there exists $f^w : \mathbb{R}^d \to [0,1]$ such that $\mathbb{E}[\mathbf{x}f^w(\mathbf{x})] = -\mathbb{E}[\mathbf{x}f'(\mathbf{x})]$. Now define $f(\mathbf{x}) = \frac{f'(\mathbf{x})+f^w(\mathbf{x})}{2}$. By construction, we have that $1 \geq f(\mathbf{x}) \geq \frac{f'(\mathbf{x})}{2} \geq \frac{\epsilon}{4\max(\|\mathbb{E}[\mathbf{x}]\|,1)}$. Furthermore, by linearity of expectation and the construction of $f^w$ we have that

$$\mathbb{E}[\mathbf{x}f(\mathbf{x})] = \mathbb{E}\left[\mathbf{x}\frac{f'(\mathbf{x})+f^w(\mathbf{x})}{2}\right] = \frac{1}{2}\left(\mathbb{E}[\mathbf{x}f'(\mathbf{x})] + \mathbb{E}[\mathbf{x}f^w(\mathbf{x})]\right) = \frac{1}{2}\left(\mathbb{E}[\mathbf{x}f'(\mathbf{x})] - \mathbb{E}[\mathbf{x}f'(\mathbf{x})]\right) = 0.$$

Therefore, $f(\mathbf{x})$ is a $\left[\frac{\epsilon}{4\max(\|\mathbb{E}[\mathbf{x}]\|,1)}, 1\right]$-valued function that satisfies $\mathbb{E}[\mathbf{x}f(\mathbf{x})] = 0$ as desired. $\quad\square$

## G.1  Proof of Lemma 15

**Lemma 25.** *Let $\Phi^C(x) = \mathbb{P}(Z > x)$ for $Z \sim N(0,1)$. Then for $|x| \leq 1$,*

$$\left| \Phi^C(x) - \left(\frac{1}{2} - \frac{1}{\sqrt{2\pi}}x\right) \right| \leq |x^3|/15.$$

*Proof of Lemma 25.* A third order Taylor expansion shows the error is at most

$$|x^3| \cdot \frac{\sup_{|y| \leq 1} |(\Phi^C)'''(y)|}{6}.$$

For $|y| \leq 1$ we easily compute

$$|(\Phi^C)'''(y)| = \frac{|y^2 - 1| \, e^{-y^2/2}}{\sqrt{2\pi}} \leq 1/\sqrt{2\pi} \leq 2/5. \qquad \square$$

**Lemma 26.** *If $X$ is $K$-sub-gaussian, then for any event $E$ and any $\mathbb{P}(E) \geq a > 0$, we have*

$$\mathbb{E}[X^2 1_E] \leq K^2 \Pr(E) \log(2/a) + K^2 a,$$
$$\mathbb{E}[X 1_E] \leq O(\Pr(E) \log(1/a)). \tag{7}$$

*Proof of Lemma 26.* We prove both estimates using the tail-sum formula. For the truncated second moment,

$$\begin{aligned}
\mathbb{E}[X^2 1_E] &= \int_0^\infty \Pr(|X^2 1_E| \geq t) dt \\
&\leq \int_0^\infty \min(\Pr(E), \Pr(X^2 \geq t)) dt \\
&\leq \int_0^\infty \min(\Pr(E), 2e^{-t/K^2}) dt \\
&\leq \Pr(E) \log(2/a) K^2 + \int_{\log(2/a)K^2}^\infty 2e^{-t/K^2} dt \\
&= K^2 \Pr(E) \log(2/a) + K^2 a.
\end{aligned}$$

Similarly for the truncated first moment,

$$\begin{aligned}
\mathbb{E}[X 1_E] &= \int_0^\infty \Pr(|X 1_E| \geq t) dt \\
&\leq \int_0^\infty \min(\Pr(E), \Pr(X \geq t)) dt \\
&\leq \int_0^\infty \min(\Pr(E), 2e^{-t^2/K^2}) dt \\
&\leq \Pr(E) \sqrt{\log(3/a)} K + \int_{\sqrt{\log(3/a)}K}^\infty 2e^{-t^2/K^2} dt \\
&= O(\Pr(E) \log(1/a)). \qquad \square
\end{aligned}$$

*Proof of Lemma 15.* Let $d\mu_X$ be the law of $X$ and $d\mu_{X|r>0}$ be the conditional law of $X$ given the event $r > 0$. Then by Bayes rule,

$$\begin{aligned}
d\mu_{X|r>0}(x) &= \frac{\Pr(r > 0 \mid X = x) d\mu_X(x)}{\Pr(r > 0)} \\
&= \frac{\Pr\left(N(\epsilon x, \sigma^2) > 0\right) d\mu_X(x)}{\Pr(r > 0)} \\
&= \frac{\Phi^C(-\epsilon x/\sigma) d\mu_X(x)}{\Pr(r > 0)}.
\end{aligned}$$

Note that

$$\mathbb{E}[X \mid r > 0] = \int_{-\infty}^\infty x \, d\mu_{X|r>0}(x) = \frac{1}{\Pr(r > 0)} \int_{-\infty}^\infty x \Phi^C(-\epsilon x/\sigma) d\mu_X(x).$$

Since $\Pr(r > 0) \le 1$ it suffices to lower-bound the latter integral by a suitable positive value. As long as $\epsilon/\sigma \le \delta_{L15} \le \sqrt{\frac{1}{4\sigma_X^2\sqrt{2\pi}}}$, we have

$$\int_{-\infty}^{\infty} x\Phi^C(-\epsilon x/\sigma)d\mu_X(x)$$

$$= \int_{-\infty}^{\infty} x\left(\frac{1}{2} + \frac{1}{\sqrt{2\pi}}\frac{\epsilon x}{\sigma}\right)d\mu_X(x)$$

$$\qquad + \int_{-\infty}^{\infty} x\left(\Phi^C(\frac{-\epsilon x}{\sigma}) - \frac{1}{2} - \frac{1}{\sqrt{2\pi}}\frac{\epsilon x}{\sigma}\right)d\mu_X(x)$$

$$= \left(\frac{\epsilon\,\mathbb{E}[X^2]}{\sigma\sqrt{2\pi}} + \int_{-\infty}^{\infty} x\left(\Phi^C(\frac{-\epsilon x}{\sigma}) - \frac{1}{2} - \frac{1}{\sqrt{2\pi}}\frac{\epsilon x}{\sigma}\right)d\mu_X(x)\right) \qquad [\mathbb{E}[X] = 0]$$

$$\ge \left(\frac{\epsilon\sigma_X^2}{\sigma\sqrt{2\pi}} - \frac{2\,\mathbb{E}[X^4]\epsilon^3}{\sigma^3}\right) \qquad\qquad\qquad\qquad\qquad \text{[Inequality (9) Below]}$$

$$\ge \frac{\epsilon\sigma_X^2}{2\sigma\sqrt{2\pi}}. \qquad\qquad\qquad\qquad\qquad\qquad\qquad [\frac{\epsilon^2}{\sigma^2} \le \frac{\sigma_X^2}{4\,\mathbb{E}[X^4]\sqrt{2\pi}}]$$

Above we used the following estimate (9). For sufficiently small $\delta_{L15}$,

$$\int_{-\infty}^{\infty} x\left(\Phi^C(\frac{-\epsilon x}{\sigma}) - \frac{1}{2} - \frac{1}{\sqrt{2\pi}}\frac{\epsilon x}{\sigma}\right)d\mu_X(x)$$

$$= \int_{|x|\le\frac{\sigma}{10\epsilon}} x\left(\Phi^C(\frac{-\epsilon x}{\sigma}) - \frac{1}{2} - \frac{1}{\sqrt{2\pi}}\frac{\epsilon x}{\sigma}\right)d\mu_X(x) + \int_{|x|>\frac{\sigma}{10\epsilon}} x\left(\Phi^C(\frac{-\epsilon x}{\sigma}) - \frac{1}{2} - \frac{1}{\sqrt{2\pi}}\frac{\epsilon x}{\sigma}\right)d\mu_X(x)$$

$$\ge \int_{|x|\le\frac{\sigma}{10\epsilon}} x\left(\Phi^C(\frac{-\epsilon x}{\sigma}) - \frac{1}{2} - \frac{1}{\sqrt{2\pi}}\frac{\epsilon x}{\sigma}\right)d\mu_X(x) - \int_{|x|>\frac{\sigma}{10\epsilon}}\left(\frac{|x|}{2} + \frac{\epsilon x^2}{\sigma\sqrt{2\pi}}\right)d\mu_X(x)$$

$$\ge -\int_{|x|\le\frac{\sigma}{10\epsilon}} |x|\left|\frac{\epsilon x}{\sigma}\right|^3 d\mu_X(x) - \int_{|x|>\frac{\sigma}{10\epsilon}}\left(\frac{|x|}{2} + \frac{\epsilon x^2}{\sigma\sqrt{2\pi}}\right)d\mu_X(x) \qquad\qquad \text{[Lemma 25]}$$

$$= -\frac{\mathbb{E}[X^4]\epsilon^3}{\sigma^3} - \int_{|x|>\frac{\sigma}{10\epsilon}}\left(\frac{|x|}{2} + \frac{\epsilon x^2}{\sigma\sqrt{2\pi}}\right)d\mu_X(x)$$

$$\ge -\frac{\mathbb{E}[X^4]\epsilon^3}{\sigma^3} - \int_{|x|>\frac{\sigma}{10\epsilon}} x^2 d\mu_X(x) \qquad\qquad\qquad\qquad [\text{if } \epsilon/\sigma \le \delta_{L15} \le 1/10]$$

$$\ge -\frac{\mathbb{E}[X^4]\epsilon^3}{\sigma^3} - \left(K^2\Pr(E)\log(2/a) + K^2 a\right) \qquad [\text{Lemma 26, } E := \{|x| > \frac{\sigma}{10\epsilon}\}, a = 2e^{-\frac{\sigma^2}{100\epsilon^2 K^2}}]$$

$$\ge -\frac{\mathbb{E}[X^4]\epsilon^3}{\sigma^3} - 2K^2 e^{-\frac{\sigma^2}{100K^2\epsilon^2}}\left(\frac{\sigma^2}{100\epsilon^2 K^2} + 1\right) \qquad [\Pr(E) \le 2e^{-\sigma^2/(100\epsilon^2 K^2)}]$$

$$\ge -\frac{2\,\mathbb{E}[X^4]\epsilon^3}{\sigma^3} \qquad [2K^2 e^{-\frac{\sigma^2}{100K^2\epsilon^2}}\left(\frac{\sigma^2}{100\epsilon^2 K^2} + 1\right) \le \frac{(\sigma_X^2)^2\epsilon^3}{\sigma^3} \le \frac{\mathbb{E}[X^4]\epsilon^3}{\sigma^3}]$$

$$\tag{9}$$

where the second to last line holds for sufficiently small $\epsilon/\sigma$. $\qquad\qquad\qquad\qquad\qquad\qquad\square$

### G.2 Proof of Lemma 16

*Proof of Lemma 16.* First, we observe that $\mathbb{E}[Y \mid r > 0] = \frac{\mathbb{E}[Y\mathbf{1}\{r>0\}]}{\mathbb{P}(r>0)}$. If $\frac{\epsilon}{\sigma}K\sqrt{\log(4)} \le 1/2$, we also have that

$$\Pr(r > 0)$$

$$\ge \Pr(r > 0 \mid X \ge -K\sqrt{\log(4)})\Pr(X \ge -K\sqrt{\log(4)})$$

$$\ge \Phi^C(\frac{\epsilon}{\sigma}K\sqrt{\log(4)})\Pr(X \ge -K\sqrt{\log(4)})$$

$$\ge \Phi^C(\frac{\epsilon}{\sigma}K\sqrt{\log(4)})\left(1 - 2e^{-K^2\log(4)/K^2}\right)$$

$$\geq \Phi^C(1/2)1/2 \qquad\qquad\qquad\qquad\qquad [\frac{\epsilon}{\sigma}K\sqrt{\log(4)} \leq 1/2]$$

$$\geq 1/8.$$

Therefore, it is sufficient to upper bound $|\mathbb{E}[Y\mathbf{1}\{r > 0\}]|$. By law of total expectation,

$$\mathbb{E}[Y\mathbf{1}\{r > 0\}] = \mathbb{E}[\mathbb{E}[Y\mathbf{1}\{r > 0\} \mid X]] = \mathbb{E}\left[\mathbb{E}[Y \mid X]\mathbb{P}(Z > -\frac{\epsilon}{\sigma}X)\right].$$

Define $Q(X) = \mathbb{E}[Y \mid X]$. Because $Y$ is sub-gaussian, the random variable $Q(X)$ must also be sub-gaussian with parameter $\sqrt{18}K$ (see [Van Handel, 2014, Exercise 3.1]). Furthermore, $\mathbb{E}[Q(X)] = \mathbb{E}[\mathbb{E}[Y \mid X]] = \mathbb{E}[Y] = 0$. Now, we have the following

$$|\mathbb{E}[Y\mathbf{1}\{r > 0\}]|$$

$$= \left|\mathbb{E}\left[\mathbb{E}[Y \mid X]\mathbb{P}(Z > -\frac{\epsilon}{\sigma}X)\right]\right|$$

$$= \left|\int_{-\infty}^{\infty} Q(x)\Phi^C(-\frac{\epsilon x}{\sigma})d\mu_X(x)\right|$$

$$= \left|\int_{-\infty}^{\infty} \frac{1}{2}Q(x)d\mu_X(x) + \int_{-\infty}^{\infty} Q(x)\left(\Phi^C(-\frac{\epsilon x}{\sigma}) - \frac{1}{2}\right)d\mu_X(x)\right|$$

$$= \left|\frac{1}{2}\mathbb{E}[Q(X)] + \int_{-\infty}^{\infty} Q(x)\left(\Phi^C(-\frac{\epsilon x}{\sigma}) - \frac{1}{2}\right)d\mu_X(x)\right|$$

$$= \left|\int_{-\infty}^{\infty} Q(x)\left(\Phi^C(-\frac{\epsilon x}{\sigma}) - \frac{1}{2}\right)d\mu_X(x)\right|$$

$$\leq \int_{|x|\leq\frac{\sigma}{10\epsilon}} |Q(x)|\left(\frac{|\epsilon x|}{\sigma\sqrt{2\pi}} + \left|\frac{\epsilon x}{\sigma}\right|^3\right)d\mu_X(x)$$

$$\qquad + \int_{|x|>\frac{\sigma}{10\epsilon}} Q(x)\left(\Phi^C(-\frac{\epsilon x}{\sigma}) - \frac{1}{2}\right)d\mu_X(x) \qquad\qquad \text{[Lemma 25]}$$

$$\leq \frac{\epsilon}{\sigma}\int_{-\infty}^{\infty} |Q(x)|\left(\frac{|x|}{\sqrt{2\pi}} + |x^3|\right)d\mu_X(x) + \frac{1}{2}\int_{|x|>\frac{\sigma}{10\epsilon}} |Q(x)|d\mu_X(x) \quad [\epsilon/\sigma \leq 1]$$

$$\leq \frac{\epsilon}{\sigma}\sqrt{\mathbb{E}[Q(X)^2]\,\mathbb{E}\left[\left(\frac{|x|}{\sqrt{2\pi}} + |x^3|\right)^2\right]} + \frac{1}{2}O\left(\frac{\sigma}{10\epsilon K}(2e^{-\frac{\sigma^2}{100\epsilon^2 K^2}})\right) \quad \text{[Lemma 26 Equation (7)]}$$

$$\leq O(\epsilon/\sigma).$$

where in the second to last line we used that $\Pr(|X| > \frac{\sigma}{10\epsilon}) \leq 2e^{-\frac{\sigma^2}{100\epsilon^2 K^2}}$ and that $Q(X)$ is $K\sqrt{18}$-sub-gaussian. The last line again uses that both $X$ and $Q(X)$ are sub-gaussian.

Therefore, we have shown that

$$|\mathbb{E}[Y \mid r > 0]| = \left|\frac{\mathbb{E}[Y\mathbf{1}\{r > 0\}]}{\mathbb{P}(r > 0)}\right| \leq 8\,|\mathbb{E}[Y\mathbf{1}\{r > 0\}]| = O(\epsilon/\sigma).$$

By symmetric arguments to the ones above, we have the same bound on $|\mathbb{E}[Y \mid r \leq 0]|$.

$\square$

## H  Proof of Lemma 17

*Proof of Lemma 17.* Fix any $\mathbf{v}$ such that $\|\mathbf{v}\| = 1$. First, we will show that

$$\Pr(|\mathbf{v} \cdot (\mathbf{Z}(\mathbf{Y}) - \mathbf{X})| \geq c_d/2) \leq \epsilon_d/2.$$

To do this, we will show that $\langle \mathbf{v}, (\mathbf{Z}(\mathbf{Y}) - \mathbf{X})\rangle$ is a sub-gaussian random variable. Let $\hat{\mathbf{X}}$ be a draw from the distribution of $\mathbf{X} \mid \mathbf{Y}$. Then we have that

$$\langle \mathbf{v}, \hat{\mathbf{X}} - \mathbf{X}\rangle = \langle \mathbf{v}, \hat{\mathbf{X}} - \mathbf{Y}\rangle + \langle \mathbf{v}, \mathbf{Y} - \mathbf{X}\rangle.$$

By standard properties of posterior samples, $\langle \mathbf{v}, \hat{\mathbf{X}} - \mathbf{Y} \rangle$ and $\langle \mathbf{v}, \mathbf{Y} - \mathbf{X} \rangle$ are identically distributed with distribution $N(0, \sigma^2)$ for $\sigma^2 = \sum_{i=1}^{d} \mathbf{v}_i^2 s_i$ (here one averages over all randomness). Therefore, we have that

$$
\begin{aligned}
&\mathbb{E}\left[\exp\left(t\langle \mathbf{v}, \mathbf{Z}(\mathbf{Y}) - \mathbf{X} \rangle\right)\right] \\
&= \mathbb{E}\left[\exp\left(t\langle \mathbf{v}, \mathbb{E}[\hat{\mathbf{X}}] - \mathbf{X} \rangle\right)\right] \\
&\leq \mathbb{E}\left[\exp\left(t\langle \mathbf{v}, \hat{\mathbf{X}} - \mathbf{X} \rangle\right)\right] && \text{[Jensen]} \\
&= \mathbb{E}\left[\exp\left(t\langle \mathbf{v}, \hat{\mathbf{X}} - \mathbf{Y} + \mathbf{Y} - \mathbf{X} \rangle\right)\right] \\
&\leq \sqrt{\mathbb{E}\left[\exp\left(2t\langle \mathbf{v}, \hat{\mathbf{X}} - \mathbf{Y} \rangle\right)\right]\mathbb{E}\left[\exp\left(2t\langle \mathbf{v}, \mathbf{Y} - \mathbf{X} \rangle\right)\right]} && \text{[Cauchy-Schwarz]} \\
&\leq e^{2t^2\sigma^2}.
\end{aligned}
$$

Therefore, $\langle \mathbf{v}, (\mathbf{Z}(\mathbf{Y}) - \mathbf{X}) \rangle$ is sub-gaussian and satisfies the tail bound

$$
\Pr(|\langle \mathbf{v}, (\mathbf{Z}(\mathbf{Y}) - \mathbf{X}) \rangle| > t) \leq 2e^{-t^2/(8\sigma^2)}.
$$

Taking $t = c_d/2$, because $\sigma^2 = \sum_{i=0}^{d} \mathbf{v}_i^2 s_i \leq \max_i s_i \leq \frac{c_d^2/32}{\log(4/\epsilon_d)}$, we have that

$$
\Pr(|\langle \mathbf{v}, (\mathbf{Z}(\mathbf{Y}) - \mathbf{X}) \rangle| > \frac{c_d}{2}) \leq 2e^{-c_d^2/(32\sigma^2)} = \epsilon_d/2. \tag{10}
$$

Now, we can prove the desired result that

$$
\begin{aligned}
&\mathbb{E}[(\langle \mathbf{Z}(\mathbf{Y}), \mathbf{v} \rangle)^+] \\
&\geq \mathbb{E}\left[(\langle \mathbf{Z}(\mathbf{Y}), \mathbf{v} \rangle)^+ \Big| |\langle \mathbf{v}, (\mathbf{Z}(\mathbf{Y}) - \mathbf{X}) \rangle| \leq \frac{c_d}{2}, \langle \mathbf{X}, \mathbf{v} \rangle \geq c_d\right] \Pr\left(|\langle \mathbf{v}, (\mathbf{Z}(\mathbf{Y}) - \mathbf{X}) \rangle| \leq \frac{c_d}{2}, \langle \mathbf{X}, \mathbf{v} \rangle \geq c_d\right) \\
&\geq \mathbb{E}\left[\frac{c_d}{2} \Big| |\langle \mathbf{v}, (\mathbf{Z}(\mathbf{Y}) - \mathbf{X}) \rangle| \leq \frac{c_d}{2}, \langle \mathbf{X}, \mathbf{v} \rangle \geq c_d\right] \Pr\left(|\langle \mathbf{v}, (\mathbf{Z}(\mathbf{Y}) - \mathbf{X}) \rangle| \leq \frac{c_d}{2}, \langle \mathbf{X}, \mathbf{v} \rangle \geq c_d\right) \\
&= \frac{c_d}{2} \Pr\left(|\langle \mathbf{v}, (\mathbf{Z}(\mathbf{Y}) - \mathbf{X}) \rangle| \leq \frac{c_d}{2}, \langle \mathbf{X}, \mathbf{v} \rangle \geq c_d\right) \\
&\geq \frac{c_d}{2}\left(\Pr(\langle \mathbf{X}, \mathbf{v} \rangle \geq c_d) - \Pr\left(|\langle \mathbf{v}, (\mathbf{Z}(\mathbf{Y}) - \mathbf{X}) \rangle| > \frac{c_d}{2}\right)\right) \\
&\geq \frac{c_d}{2}\left(\epsilon_d - \frac{\epsilon_d}{2}\right) && \text{[Eq (10) and lemma assum]} \\
&= \frac{c_d\epsilon_d}{4}. && \square
\end{aligned}
$$

### H.1 Proof of Lemma 14

*Proof of Lemma 14.* Let $B$ be a Bernoulli random variable such that $\Pr(B = 1) = \epsilon$ and let $Z \sim N(0, 1)$ be independent of $B$ and $X$. Then we can write $R \sim X \cdot B + Z$.

Then we have that

$$
\begin{aligned}
&\mathbb{E}[X \mid R > 0] \\
&= \mathbb{E}[X \mid R > 0, B = 1]\Pr(B = 1 \mid R > 0) + \mathbb{E}[X \mid R > 0, B = 0]\Pr(B = 0 \mid R > 0) \\
&= \mathbb{E}[X \mid X + Z > 0]\Pr(B = 1 \mid R > 0) \\
&= \mathbb{E}[X \mid X + Z > 0]\frac{\Pr(R > 0 \mid B = 1)\Pr(B = 1)}{\Pr(R > 0)} \\
&\geq \mathbb{E}[X \mid X + Z > 0]\Pr(R > 0 \mid B = 1)\Pr(B = 1) \\
&= \mathbb{E}[X \mid X + Z > 0]\Pr(X + Z > 0)\epsilon. \tag{11}
\end{aligned}
$$

Next, we need to lower bound $\mathbb{E}[X \mid X + Z > 0] \Pr(X + Z > 0)$. Applying Baye's rule gives

$$\mathbb{E}[X \mid X + Z > 0] \Pr(X + Z > 0)$$

$$= \Pr(X + Z > 0) \int_{-\infty}^{\infty} x \, d\mu_{X \mid X+Z>0}(x)$$

$$= \int_{-\infty}^{\infty} x \left(\Phi^C(-x)\right) d\mu_X(x)$$

$$= \int_{-\infty}^{\infty} x \left(\Phi^C(-x) - 1/2\right) d\mu_X(x) \qquad\qquad [\mathbb{E}[X] = 0]$$

Note that $\left(\Phi^C(-x) - 1/2\right)$ has the same sign as $x$ and has magnitude increasing in $|x|$. Therefore,

$$\geq \int_{x \geq \frac{\sigma_X}{\sqrt{10}}} \frac{\sigma_X}{\sqrt{10}} \mathbb{P}\left(0 \leq Z \leq \frac{\sigma_X}{\sqrt{10}}\right) d\mu_X(x) + \int_{x \leq -\frac{\sigma_X}{\sqrt{10}}} \frac{\sigma_X}{\sqrt{10}} \mathbb{P}\left(0 \leq Z \leq \frac{\sigma_X}{\sqrt{10}}\right) d\mu_X(x)$$

$$= \frac{\sigma_X \mathbb{P}(0 \leq Z \leq \frac{\sigma_X}{\sqrt{10}})}{\sqrt{10}} \mathbb{P}(|X| > \frac{\sigma_X}{\sqrt{10}})$$

$$\geq \frac{\sigma_X \mathbb{P}(0 \leq Z \leq \frac{\sigma_X}{\sqrt{10}})}{\sqrt{10}} \left(\frac{4\sigma_X^2}{5K^2 \log\left(\frac{20K^2}{\sigma_X^2}\right)}\right). \qquad\qquad [\text{Equation (13) below}]$$

$$\geq \Omega(\sigma_X^5). \qquad\qquad\qquad\qquad\qquad\qquad\qquad\qquad\qquad\qquad (12)$$

Combining Equations (11) and (12) gives the desired result of the lemma.

It remains to show the lower bound on $\mathbb{P}(|X| > \frac{\sigma_X}{\sqrt{10}})$ used in the penultimate line above.

Define $a = K^2 \log\left(\frac{20K^2}{\sigma_X^2}\right) \geq \sigma_X^2 \log(10)/2 > \sigma_X^2/10$ (using Equation (3) ). Next, we observe that

$$\mathbb{E}[X^2]$$

$$= \int_0^{\infty} \mathbb{P}(X^2 > t) dt$$

$$= \int_0^{\infty} \mathbb{P}(X > \sqrt{t}) dt$$

$$= \int_0^{\sigma_X^2/10} \mathbb{P}(X > \sqrt{t}) dt + \int_{\sigma_X^2/10}^{a} \mathbb{P}(X > \sqrt{t}) dt + \int_a^{\infty} \mathbb{P}(X > \sqrt{t}) dt$$

$$\leq \sigma_X^2/10 + \int_{\sigma_X^2/10}^{a} \mathbb{P}(X > \sqrt{t}) dt + \int_a^{\infty} 2e^{-t/K^2} dt$$

$$= \sigma_X^2/10 + \int_{\sigma_X^2/10}^{a} \mathbb{P}(X > \sqrt{t}) dt + 2K^2 e^{-a/K^2}$$

$$= \sigma_X^2/5 + \int_{\sigma_X^2/10}^{a} \mathbb{P}(X > \sqrt{t}) dt \qquad\qquad [\text{Def of } a]$$

$$\leq \sigma_X^2/5 + \left(a - \frac{\sigma_X^2}{10}\right) \mathbb{P}\left(X > \frac{\sigma_X}{\sqrt{10}}\right). \qquad\qquad [\mathbb{P}(X > \sqrt{t}) \text{ monotone decr.}]$$

Since $\mathbb{E}[X^2] = \sigma_X^2$, this implies that

$$\left(a - \frac{\sigma_X^2}{10}\right) \mathbb{P}\left(X > \frac{\sigma_X}{\sqrt{10}}\right) \geq \frac{4\sigma_X^2}{5}.$$

Therefore, we can conclude that

$$\mathbb{P}\left(X > \frac{\sigma_X}{\sqrt{10}}\right) \geq \frac{4\sigma_X^2/5}{a - \sigma_X^2/10} \geq \frac{4\sigma_X^2/5}{a} = \frac{4\sigma_X^2}{5K^2 \log\left(\frac{20K^2}{\sigma_X^2}\right)}. \tag{13}$$

By symmetry, identical logic as above gives the desired upper bound on $\mathbb{E}[X \mid R \leq 0]$. $\qquad\square$

# I  Proof of Proposition 11

*Proof of Proposition 11.* We first show that $\mathbf{A}^{(t)}$ on Line 6 satisfies $\mathbf{A}^{(t)} \in S^\perp$ when $\Psi = 1$. Recall $\mathbf{x}^*$ defined as $x_\ell^* = \langle \boldsymbol{\ell}^*, \mathbf{w}_\ell \rangle$ and recall that $\mathbf{z}(\mathbf{y}) = \mathbb{E}[\mathbf{x}^* \mid \hat{\mathbf{y}} = \mathbf{y}]$. By construction,

$$\mathbb{E}[\mathbf{x}^* \mid \Psi = 1]$$
$$= \int \mathbb{E}[\mathbf{x}^* \mid \Psi = 1, \hat{\mathbf{y}} = \mathbf{y}] d\mu_{\hat{\mathbf{y}}|\Psi=1}(\mathbf{y})$$
$$= \int \mathbb{E}[\mathbf{x}^* \mid \hat{\mathbf{y}} = \mathbf{y}] \frac{\Pr(\Psi = 1 \mid \hat{\mathbf{y}} = \mathbf{y})}{\Pr(\Psi = 1)} d\mu_{\hat{\mathbf{y}}}(\mathbf{y}) \qquad [\Psi = 1 \text{ is a function of } \hat{\mathbf{y}}]$$
$$= \frac{1}{\Pr(\Psi = 1)} \int \mathbb{E}[\mathbf{x}^* \mid \hat{\mathbf{y}} = \mathbf{y}] f\left(\mathbf{z}(\mathbf{y})\right) d\mu_{\hat{\mathbf{y}}}(\mathbf{y})$$
$$= \frac{1}{\Pr(\Psi = 1)} \int \mathbf{z}(\mathbf{y}) f(\mathbf{z}(\mathbf{y})) d\mu_{\hat{\mathbf{y}}}(\mathbf{y})$$
$$= \frac{1}{\Pr(\Psi = 1)} \mathbb{E}\left[\mathbf{z}(\hat{\mathbf{y}}) f(\mathbf{z}(\hat{\mathbf{y}}))\right]$$
$$= 0. \qquad\qquad\qquad\qquad [\text{Definition of } f]$$

Because $x_\ell^* = \langle \boldsymbol{\ell}^*, \mathbf{w}_\ell \rangle$ for $\ell \leq \ell_\lambda$, this implies that $\mathbb{E}[\langle \boldsymbol{\ell}^*, \mathbf{w}_\ell \rangle \mid \Psi = 1] = 0$ for $\ell \leq \ell_\lambda$. Therefore, we must have that $\mathbb{E}[\boldsymbol{\ell}^* \mid \Psi = 1] \in S^\perp$. By construction of $\mathbf{A}^{(t)}$ in Line 6, this implies that $\mathbf{A}^{(t)} \in S^\perp$ when $\Psi = 1$.

Define

$$\mathbf{A} = \begin{cases} \frac{\mathbb{E}[\boldsymbol{\ell}^*|\Psi=1]}{\|\mathbb{E}[\boldsymbol{\ell}^*|\Psi=1]\|_2} & \text{if } \mathbb{E}[\boldsymbol{\ell}^* \mid \Psi = 1] \neq 0 \\ \mathbf{w}_{\ell_\lambda+1} & \text{otherwise,} \end{cases}$$

in other words $\mathbf{A}$ is equal to $\mathbf{A}^{(t)}$ when $\Psi = 1$.

By the choice of $f$, we have that $\mathbb{P}(\Psi = 1) \geq \frac{\epsilon_d c_d}{16 \max(\|\mathbb{E}[\mathbf{z}(\hat{\mathbf{y}})]\|, 1)} \geq \frac{\epsilon_d c_d}{16\left(K\sqrt{\pi}+1\right)}$, where in the last line we used that Equation (A) implies

$$\max(\|\mathbb{E}[\mathbf{z}(\hat{\mathbf{y}})]\|, 1) = \max(\|\mathbb{E}[\mathbf{x}^*]\|, 1) \leq \max\left(\|\mathbb{E}[\boldsymbol{\ell}^*]\|, 1\right) \leq \max\left(K\sqrt{\pi}, 1\right) \leq K\sqrt{\pi} + 1.$$

By construction, we therefore have that for any realization of $\mathbf{z}(\hat{\mathbf{y}})$, the probability that $R = r^{(t)} = \langle \boldsymbol{\ell}^*, \mathbf{A}^{(t)} \rangle + w_t = \langle \boldsymbol{\ell}^*, \mathbf{A} \rangle + w_t$ is exactly $\frac{\epsilon_d c_d}{16\left(K\sqrt{\pi}+1\right)}$ and otherwise $R \sim N(0,1)$.

We can now apply Lemma 14 with $X = \langle \boldsymbol{\ell}^*, \mathbf{A} \rangle$, and $\epsilon = \frac{\epsilon_d c_d}{16\left(K\sqrt{\pi}+1\right)}$ to get that either $\mathbf{a} = \mathbf{w}_{\ell_\lambda+1}$ or

$$|\langle \mathbf{A}, \mathbf{a} \rangle| = |\langle \mathbf{A}, \mathbb{E}[\boldsymbol{\ell}^* \mid 1_{R>0}] \rangle| = |\mathbb{E}[\langle \boldsymbol{\ell}^*, \mathbf{A} \rangle \mid 1_{R>0}]| \geq \frac{c_{L14}\epsilon_d c_d \operatorname{Var}(\langle \boldsymbol{\ell}^*, \mathbf{A} \rangle)^{2.5}}{16\left(K\sqrt{\pi}+1\right)} \geq \frac{c_{L14}\epsilon_d c_d \, c_v^{2.5}}{16\left(K\sqrt{\pi}+1\right)},$$

where in the last inequality we used Assumption 2.

Because $\mathbf{A} \in S^\perp$ and $\|\mathbf{A}\| = 1$, the previous equation implies the desired result that $\|\mathcal{P}_{S^\perp}(\mathbf{a})\| \geq \frac{c_{L14}\epsilon_d c_d \, c_v^{2.5}}{16\left(K\sqrt{\pi}+1\right)}$. $\qquad\square$

# J  Proof of Proposition 12

*Proof of Proposition 12.* The first step is to rewrite $R$ from Algorithm 6 Line 7 so that we can apply Lemmas 15 and 16.

Define

$$W := \sum_{t'=t}^{t+L-1} \left( w_{t'} - \sum_{k=1}^{j} (c_k q_k^{t'} - c_k \langle \mathbf{v}_k, \boldsymbol{\ell}^* \rangle) \right) = \sum_{t'=t}^{t+L-1} \left( w_{t'} - \sum_{k=1}^{j} c_k q_k^{t'} \right) + L \sum_{k=1}^{j} c_k \langle \mathbf{v}_k, \boldsymbol{\ell}^* \rangle.$$

Note that $W$ is normally distributed with mean $0$ and variance $\sigma_W^2 := L(1 + \sum_{k=1}^{j} c_k^2)$. By construction, we can rewrite $R$ as

$$
\begin{aligned}
R &= \sum_{t'=t}^{t+L-1} \left( r^{(t')} - \sum_{k=1}^{j} c_k q_k^{t'} \right) \\
&= \sum_{t'=t}^{t+L-1} \left( (\langle \mathbf{a}, \boldsymbol{\ell}^* \rangle + w_{t'}) - \sum_{k=1}^{j} c_k q_k^{t'} \right) \\
&= \sum_{t'=t}^{t+L-1} \langle \mathbf{a}, \boldsymbol{\ell}^* \rangle - L \sum_{k=1}^{j} c_k \langle \mathbf{v}_k, \boldsymbol{\ell}^* \rangle + W \\
&= L \langle \mathbf{a}, \boldsymbol{\ell}^* \rangle - L \langle \mathcal{P}_S(\mathbf{a}), \boldsymbol{\ell}^* \rangle + W \qquad \text{[Lemma 18 implies } \mathcal{P}_S(\mathbf{a}) = \sum_{k=1}^{j} c_k \mathbf{v}_k] \\
&= L \langle (\mathbf{a} - \mathcal{P}_S(\mathbf{a})), \boldsymbol{\ell}^* \rangle + W \\
&= L \langle \mathcal{P}_{S^\perp}(\mathbf{a}), \boldsymbol{\ell}^* \rangle + W \\
&= L \| \mathcal{P}_{S^\perp}(\mathbf{a}) \| \langle \mathbf{x}, \boldsymbol{\ell}^* \rangle + W. \qquad\qquad (14)
\end{aligned}
$$

Therefore, $R$ is exactly in the form necessary to apply Lemmas 15 and 16. In order to apply these lemmas, we need that $\frac{L \| \mathcal{P}_{S^\perp}(\mathbf{a}) \|}{\sigma_W} \leq \delta_{L15}$ and $\frac{L \| \mathcal{P}_{S^\perp}(\mathbf{a}) \|}{\sigma_W} \leq \delta_{L16}$ respectively.

To see this, note that $\mathcal{P}_{S^\perp}(\mathbf{a}) \leq \sqrt{\lambda}$ (as otherwise ExponentialGrowth would not have been called), and therefore

$$
\begin{aligned}
\frac{L \| \mathcal{P}_{S^\perp}(\mathbf{a}) \|}{\sigma_W} &\leq \frac{L \sqrt{\lambda}}{\sqrt{L(1 + \sum_{k=1}^{j} c_k^2)}} \\
&= \sqrt{\frac{4 \lambda d (\mathbb{E}[\boldsymbol{\ell}^*_1] + 1)^2}{c_{L15}^2}} \\
&\leq \sqrt{\frac{4 \lambda d (K \sqrt{\pi} + 1)^2}{(c_{\mathrm{v}} / \sqrt{8\pi})^2}} \qquad\qquad \text{[Equation (A), Assum 2]} \\
&\leq \min(\delta_{L15}, \delta_{L16}, 1/c_{L16}), \qquad\qquad (15)
\end{aligned}
$$

where in the last line we used $\lambda \leq \min(\delta_{L15}, \delta_{L16}, 1/c_{L16})^2 \frac{(c_{\mathrm{v}} / \sqrt{8\pi})^2}{4d(K \sqrt{\pi} + 1)^2}$ by our assumption on $\lambda$.

Applying Lemmas 15 and 16 gives the following two bounds. Define $\mathbf{y} = \mathbb{E}[\boldsymbol{\ell}^* \mid 1_{R>0}]$. The first is a lower bound on $|\langle \mathbf{x}, \mathbf{y} \rangle|$. Importantly, we can apply Lemma 15 for $X = \langle \mathbf{x}, \boldsymbol{\ell}^* \rangle$ because of Equations (14) and (15).

$$
\begin{aligned}
|\langle \mathbf{x}, \mathbf{y} \rangle| &= |\langle \mathbf{x}, \mathbb{E}[\boldsymbol{\ell}^* \mid 1_{R>0}] \rangle| \\
&= |\mathbb{E}[\langle \mathbf{x}, \boldsymbol{\ell}^* \rangle \mid 1_{R>0}]| \\
&\geq \frac{c_{L15} L \| \mathcal{P}_{S^\perp}(\mathbf{a}) \|}{\sigma_W} \qquad\qquad \text{[Lemma 15]} \\
&= \frac{c_{L15} \sqrt{L} \| \mathcal{P}_{S^\perp}(\mathbf{a}) \|}{\sqrt{1 + \sum_{k=1}^{j} c_k^2}} \\
&= c_{L15} \sqrt{\frac{4d(\mathbb{E}[\boldsymbol{\ell}^*_1] + 1)^2}{c_{L15}^2}} \| \mathcal{P}_{S^\perp}(\mathbf{a}) \|
\end{aligned}
$$

$$= \sqrt{4d(\mathbb{E}[\boldsymbol{\ell}^*_1] + 1)^2} \, \|\mathcal{P}_{S^\perp}(\mathbf{a})\|$$
$$= 2\sqrt{d}(\mathbb{E}[\boldsymbol{\ell}^*_1] + 1) \, \|\mathcal{P}_{S^\perp}(\mathbf{a})\| \,. \tag{16}$$

The next equation is an upper bound on $\|\mathbf{y}_i\|$ for all $i \in [d]$:

$$|\mathbf{y}_i| = \mathbb{E}[\boldsymbol{\ell}^*_i \mid 1_{R>0}]$$
$$\leq \mathbb{E}[\boldsymbol{\ell}^*_i] + c_{L16} L \, \|\mathcal{P}_{S^\perp}(\mathbf{a})\| / \sigma_W \qquad \text{[Lemma 16]}$$
$$\leq \mathbb{E}[\boldsymbol{\ell}^*_i] + 1. \qquad\qquad\qquad \text{[Equation (15)]}$$

Using the above equation, we can bound $\|\mathbf{y}\|_2$ as follows. Because $\mathbb{E}[\boldsymbol{\ell}^*_1] \geq \mathbb{E}[\boldsymbol{\ell}^*_i]$ for all $i$,

$$\|\mathbf{y}\|_2 \leq \sqrt{\sum_{i=1}^{d} (\mathbb{E}[\boldsymbol{\ell}^*_i] + 1)^2} \leq \sqrt{d}(\mathbb{E}[\boldsymbol{\ell}^*_1] + 1).$$

Equation (16) implies that $\mathbf{y} \neq \mathbf{0}$. This implies by construction that $\mathbf{b} = \text{Exploit}(1_{R>0}, \mathbf{w}_{\ell_\lambda+1}) = \frac{\mathbf{y}}{\|\mathbf{y}\|}$. Putting everything together, we have that

$$|\langle \mathbf{x}, \mathbf{b} \rangle| = \frac{|\langle \mathbf{x}, \mathbf{y} \rangle|}{\|\mathbf{y}\|_2} \geq \frac{2\sqrt{d}(\mathbb{E}[\boldsymbol{\ell}^*_1] + 1)) \, \|\mathcal{P}_{S^\perp}(\mathbf{a})\|}{\sqrt{d}(\mathbb{E}[\boldsymbol{\ell}^*_1] + 1)} \geq 2 \, \|\mathcal{P}_{S^\perp}(\mathbf{a})\| \,.$$

Finally, because $\mathbf{x} \in S^\perp$, this implies the desired result that

$$\|\mathcal{P}_{S^\perp}(\mathbf{b})\| \geq |\langle \mathbf{x}, \mathbf{b} \rangle| \geq 2 \, \|\mathcal{P}_{S^\perp}(\mathbf{a})\| \,. \qquad \square$$

# K   Proof of Theorem 13

*Proof of Theorem 13.* We begin by proving that Algorithm 4 is BIC. There are four places where we set $\mathbf{A}^{(t)}$. The first is in the Line 4 of Algorithm 4, where we set $\mathbf{A}^{(t)} = \mathbf{e}_1$. This is BIC because we assumed (without loss of generality) that $\mathbb{E}[\boldsymbol{\ell}^*_i] = 0$ for all $i > 1$ and $\mathbb{E}[\boldsymbol{\ell}^*_1] \geq 0$.

The second place we set $\mathbf{A}^{(t)}$ is in Line 6 of Algorithm 5. This choice of $\mathbf{A}^{(t)}$ is BIC with the signal $\Psi$ by construction and Lemma 7.

The third place we set $\mathbf{A}^{(t)}$ is in Line 5 of Algorithm 6. In order for this to be BIC, we must show that every input $\mathbf{a}$ to Algorithm 3 is BIC. The first time Algorithm 6 is used for any fixed value of $j$, the input action $\mathbf{a}$ is the action returned by Algorithm 5. This is BIC for signal $R$ defined on Line 7 of Algorithm 5 by construction. Each subsequent call to Algorithm 6 for a fixed value of $j$ uses an action $\mathbf{a}$ that is returned by the previous call to Algorithm 6. This is BIC for signal $R$ defined on Line 7 of Algorithm 6.

The final time we set an action is on Line 17 of Algorithm 4 This action is again an action returned by the last call to Algorithm 6, which as argued above is BIC for signal $R$.

The rest of the proof will focus on bounding the sample complexity of Algorithm 4.

First, we will bound the number of times the inner while loop (Line 13) calls Algorithm 6 for each value of $j$. By Proposition 11, the action returned by Algorithm 5 satisfies $\|\mathcal{P}_{S^\perp}(\mathbf{a})\| \geq c_{P11} c_{\mathrm{v}}^{2.5} \epsilon_d c_d$. Furthermore, by Proposition 12, $\|\mathcal{P}_{S^\perp}(\mathbf{a})\|$ doubles with each call to Algorithm 6. Therefore, $\|\mathcal{P}_{S^\perp}(\mathbf{a})\| \geq \sqrt{\lambda}$ will be satisfied after at most $\log_2 \left( \frac{\sqrt{\lambda}}{c_{P11} c_{\mathrm{v}}^{2.5} \epsilon_d c_d} \right) = O\left( \log(\frac{1}{c_{\mathrm{v}} \epsilon_d c_d}) \right)$ calls to Algorithm 6.

Next we will bound the number of steps in each call to Algorithm 6, which is equivalent to bounding the $L$ defined on Line 4 of Algorithm 6. To do this, we note that the $c_i$ in Algorithm 6 are the same as the $c_i$ in Lemma 18 with $\epsilon = \lambda$, $\ell = \ell_\lambda$, $\mathbf{u} = \mathcal{P}_{S^\perp}(\mathbf{a})$, and $\mathbf{v}_1, ..., \mathbf{v}_j$. This implies that

$$\sum_{k=1}^{j} c_k^2 \leq \frac{1}{\lambda}. \qquad\qquad \text{[Lemma 18]}$$

Therefore, we can bound $L$ as follows:

$$L = \frac{4d(\mathbb{E}[\boldsymbol{\ell}^*_1]+1)(1+\sum_{k=1}^j c_k^2)}{c_{L15}^2} \le \frac{4d(\mathbb{E}[\boldsymbol{\ell}^*_1]+1)(1+\frac{1}{\lambda})}{c_{L15}^2}$$

$$\le \frac{4d(K\sqrt{\pi}+1)(1+\frac{1}{\lambda})}{c_v^2/(8\pi)} \qquad \text{[Assum 2, Eq (A)]}$$

$$= O\left(\frac{d}{\lambda c_v^2}\right).$$

For each loop of the while loop on Line 8, we also have $\kappa = O(\frac{\log(1/\epsilon_d)}{\lambda c_d^2} + \frac{d}{\lambda c_v^2})$ steps in the loop on Line 16. All together, this gives that each iteration of the loop on Line 8 takes at most

$$O\left(\frac{d\log(\frac{1}{c_v \epsilon_d c_d})}{\lambda c_v^2} + \frac{\log(1/\epsilon_d)}{\lambda c_d^2}\right) = O\left(\log\left(\frac{1}{c_v \epsilon_d c_d}\right)\left(\frac{d}{\lambda c_v^2} + \frac{1}{\lambda c_d^2}\right)\right)$$

steps. Next, we will bound the number of iterations of the while loop on Line 8.

For each $j$, we will apply Lemma 9 with $\epsilon = \lambda$, $\mathbf{u} = \mathbf{v}_{j+1}$, and the vectors $\mathbf{v}_1, ..., \mathbf{v}_j$. By construction of the algorithm, $S^\perp$ is non-empty because the algorithm has not yet terminated, and $\|\mathcal{P}_{S^\perp}(\mathbf{v}_{j+1})\|^2 \ge \lambda$ by the termination condition of the while loop on Line 13 of Algorithm 4. Therefore, this satisfies the assumption of Lemma 9. Define $\lambda_1^j, ..., \lambda_d^j$ as the eigenvalues of $\mathbf{M}^j := \sum_{i=1}^j \mathbf{v}_i^{\otimes 2}$ and define $\ell^j$ as the largest index such that $\lambda_{\ell^j}^j \ge 200d^3/\lambda^2$ (and $\ell^j = 0$ if all eigenvalues of $\mathbf{M}^j$ are less than $200d^3/\lambda^2$). Now define

$$\Delta^j = \sum_{i=\ell^j+1}^d \left(\frac{200d^3}{\lambda^2} - \lambda_i\right).$$

Note that for any fixed $i$, the $i$th eigenvalue does not decrease between $\mathbf{M}^j$ and $\mathbf{M}^{j+1}$. Because of this monotonicity, Lemma 9 implies that for every round $j$, either

$$\ell^{j+1} \ge \ell^j + 1 \quad \text{or} \quad \Delta^{j+1} \le \Delta^j - \frac{\lambda}{2}.$$

Because $\ell^1 \ge 0$ and $\Delta^1 \le \frac{200d^3}{\lambda^2} \cdot d = \frac{200d^4}{\lambda^2}$, this implies that after $\frac{\frac{200d^4}{\lambda^2}}{\lambda/2} + d$ applications of Lemma 9, either $\ell^j = d$ or $\Delta^j = 0$. This means that after $\frac{400d^4}{\lambda^2} + d$ applications of Lemma 9, the smallest eigenvalue of $\mathbf{M}^j$ must be at least $200d^3/\lambda^2 \ge \lambda$. However, this means that the algorithm must terminate before round $400d^4/\lambda^3 + d$. Therefore, the number of iterations of the while loop on Line 8 is less than $O(d^4/\lambda^3)$. Putting everything together, the total number of steps needed for $\lambda$-exploration is upper bounded by

$$O\left(\log\left(\frac{1}{c_v \epsilon_d c_d}\right)\left(\frac{d}{\lambda c_v^2} + \frac{1}{\lambda c_d^2}\right)\right) \cdot O\left(\frac{d^4}{\lambda^3}\right) = O\left(\log\left(\frac{1}{c_v \epsilon_d c_d}\right)\left(\frac{d^5}{\lambda^4 c_v^2} + \frac{d^4}{c_d^2\lambda^4}\right)\right). \quad \square$$

## L   Proof of Proposition 5

*Proof of Proposition 5.* First, for any unit vector $\mathbf{v}$, we have $B_{r/3}(2r\mathbf{v}/3) \subseteq \mathcal{K} \subseteq B_1(0)$. Therefore

$$\mu(B_{r/3}(2r\mathbf{v}/3)) = \text{Vol}(B_{r/3}(2r\mathbf{v}/3))/\text{Vol}(\mathcal{K}) \ge \text{Vol}(B_{r/3}(2r\mathbf{v}/3))/\text{Vol}(B_1(0)) = (r/3)^d.$$

Since $\langle \mathbf{x}, \mathbf{v}\rangle \ge r/3$ for all $\mathbf{x} \in B_{r/3}(2r\mathbf{v}/3)$, this confirms the values $(c_d, \epsilon_d) = (r/3, (r/3)^d)$.

The bound on $c_v$ follows by [Sellke, 2023, Lemma 3.2] and Jensen's inequality since $\mathcal{K}$ has width at least $2r$ in any direction. The bound on $K$ is trivial since $2e^{-(t/1.25)^2} \ge 1$ for $|t| \le 1$. $\qquad\square$

## M   Proof of Proposition 6

We first recall several useful facts on log-concave distributions. Throughout we take $\mu$ to be $\alpha$-log-concave and $\beta$-log-smooth with mode $\mathbf{x}^*$ and mean $\bar{\mathbf{x}}$, possibly in dimension 1. (The proof will use 1-dimensional projections of the original measure $\mu$.) We will write $\mathbf{x} \sim \mu$ instead of $\boldsymbol{\ell}^* \sim \mu$.

**Fact 27** ([Dwivedi et al., 2019, Lemma 5], [Durmus and Moulines, 2019, Theorem 1]). *For $x \sim \mu$, we have $\mathbb{E}[\|x - x^*\|^2] \leq 1/\alpha$ and with probability $1 - \delta$:*

$$\|x - x^*\|_2 \leq 2\alpha^{-1/2}\left(1 + \sqrt{\frac{\log(1/\delta)}{d}} + \sqrt[4]{\frac{\log(1/\delta)}{d}}\right).$$

**Fact 28** ([Chewi and Pooladian, 2023, Lemma 2]). *We have the covariance bounds*

$$\frac{I_d}{\alpha d} \succeq Cov(\mu) \succeq \frac{I_d}{\beta d}. \tag{18}$$

**Fact 29.** *Any $1$-dimensional projection of $\mu$ is also $\alpha d$-log-concave and $\beta d$-log-smooth.*

*Proof.* Preservation of strong log-concavity under projection is well known, see e.g. [Saumard and Wellner, 2014, Theorem 3.8]. For log-smoothness, supposing for convenience that the projection is onto the first coordinate axis, the claim is proved by the following standard computation. With $e^{-f(\mathbf{x})}$ the density of $\mu$ and $e^{-g(x)}$ the density of the projection of $\mu$ to the first coordinate axis, one may compute as in [Saumard and Wellner, 2014, Proof of Proposition 7.1] that

$$g''(x) = \mathbb{E}^\mu[\partial_{1,1}f(\mathbf{x})|x_1 = x] - \operatorname{Var}^\mu[\partial_1 f(\mathbf{x})|x_1 = x] \leq \mathbb{E}^\mu[\partial_{1,1}f(\mathbf{x})|x_1 = x] \leq \beta d.$$

This completes the proof. $\qquad\square$

*Proof of Proposition 6.* We have $c_v \geq \frac{1}{\beta d}$ directly from (18).

For $\epsilon_d$, let $\bar{\mathbf{x}}$ be the mean under $\mu$ and note that from (18), we find

$$
\begin{aligned}
\|\bar{\mathbf{x}} - \mathbf{x}^*\| &= \sup_{\|\mathbf{w}\|=1} \langle \bar{\mathbf{x}} - \mathbf{x}^*, \mathbf{w} \rangle \\
&= \sup_{\|\mathbf{w}\|=1} \mathbb{E}^{\mathbf{x}\sim\mu}[\langle \mathbf{x} - \mathbf{x}^*, \mathbf{w} \rangle] \\
&\leq \sqrt{\sup_{\|\mathbf{w}\|=1} \mathbb{E}^{\mathbf{x}\sim\mu}[\langle \mathbf{x} - \mathbf{x}^*, \mathbf{w} \rangle^2]} \\
&\leq \sqrt{\langle Cov(\mu), \mathbf{w}^{\otimes 2} \rangle} \\
&\leq 1/\sqrt{\alpha d}.
\end{aligned}
$$

Fixing a unit vector $\mathbf{v}$ as in Assumption 3, we consider the projection $P$ onto the $1$-dimensional subspace spanned by $\mathbf{v}$, and let $P(\mu)$ be the pushforward of $\mu$ under the projection (to which Fact 29 applies). Identifying $P(\mathbb{R}^d)$ isometrically with $\mathbb{R}$, let $\hat{x}$ be the mode of $P(\mu)$. Then the same argument as above applies to $P(\mu)$ shows $\|P(\bar{\mathbf{x}}) - \hat{x}\| \leq 1/\sqrt{\alpha d}$, and so

$$\|\hat{x}\| \leq \|\mathbf{x}^*\| + \frac{2}{\sqrt{\alpha d}} \leq \gamma + \frac{2}{\sqrt{\alpha d}}.$$

(Note that if $\bar{\mathbf{x}} = 0$ then this shows $\|\hat{x}\| \leq 1/\sqrt{\alpha d}$, which following the arguments below leads to $\epsilon_d \geq \Omega(1)$ as mentioned below Proposition 6.)

Write $f : \mathbb{R} \to \mathbb{R}_+$ for the density of $P(\mu)$, and $g(x) = \log f(x)$. We have $f'(\hat{x}) = 0$ and so $g'(\hat{x}) = 0$ also. By Fact 29, we have $g''(x) \in [-\beta d, -\alpha d]$ for all $x$, so for $x \geq \hat{x}$ we have:

$$g'(x) = g'(x) - g'(\hat{x}) = \int_{\hat{x}}^x g''(y)dy \in [-\beta d(x - \hat{x}), -\alpha d(x - \hat{x})].$$

Integrating again, we find

$$g(x) - g(\hat{x}) = \int_{\hat{x}}^x g'(y)dy \in [-\beta d(x - \hat{x})^2/2, -\alpha d(x - \hat{x})^2/2].$$

Identical reasoning gives the same conclusion for $x \leq \hat{x}$. Translating back to $f = e^g$, we conclude that for each $x \in \mathbb{R}$:

$$e^{-\beta d|x - \hat{x}|^2/2} \leq \frac{f(x)}{f(\hat{x})} \leq e^{-\alpha d|x - \hat{x}|^2/2}.$$

It follows that for $\mathbf{x} = \ell^* \sim \mu$ and $x \sim P(\mu)$:

$$\Pr[\langle \mathbf{v}, \mathbf{x} \rangle \geq c_d] \geq \Pr[x \geq \gamma + \frac{2}{\sqrt{\alpha d}} + c_d].$$

Letting $J = \gamma + \frac{2}{\sqrt{\alpha d}} + c_d$, the latter probability is at least

$$\frac{\int_J^\infty e^{-\beta d z^2/2} dz}{\int_\mathbb{R} e^{-\alpha d z^2/2} dz} = \sqrt{\alpha/\beta} \cdot \Phi^C(J\sqrt{\beta d}) \geq \frac{J e^{-J^2 \beta d/2} \sqrt{\alpha d}}{(1 + J^2 \beta d)\sqrt{2\pi}}.$$

The last inequality follows from the classical bound $\Phi^C(\kappa) \geq \frac{\varphi(\kappa)\kappa}{1+\kappa^2}$ where $\varphi$ is the standard Gaussian density Gordon [1941]. This confirms the value of $\epsilon_d$.

For $K$ we consider a similar projection, and note that by Fact 29 and the Bakry-Emery theory for strongly log-concave measures (see e.g. [Anderson et al., 2010, Lemma 2.3.3], we have

$$\mathbb{E}[e^{\lambda\langle \mathbf{v}, \mathbf{x} - \bar{\mathbf{x}} \rangle}] \leq e^{\lambda^2 \alpha d/2}, \quad \forall \lambda \in \mathbb{R}.$$

Thus with $J_0 = \gamma + \frac{1}{\sqrt{\alpha d}} \geq \|\langle \bar{\mathbf{x}}, \mathbf{v} \rangle\|$, we have (using $\lambda J_0 \leq \frac{1+\lambda^2 J_0^2}{2}$):

$$\mathbb{E}[e^{\lambda\langle \mathbf{v}, \mathbf{x} \rangle}] \leq e^{(\lambda^2 \alpha d/2) + \lambda J_0} \leq e^{0.5} \cdot e^{\lambda^2(J_0^2 + \alpha d)/2} \leq 2e^{\lambda^2(J_0^2 + \alpha d)/2}.$$

It follows by the usual Markov inequality arguments that

$$\Pr[|\langle \mathbf{v}, \mathbf{x} \rangle| \geq t] \leq 4e^{-\frac{t^2}{2(J_0^2 + \alpha d)}}.$$

Since probabilities are at most 1 and $a \leq \sqrt{a}$ for $a \leq 1$ we find

$$\Pr[|\langle \mathbf{v}, \mathbf{x} \rangle| \geq t] \leq 2e^{-\frac{t^2}{4(J_0^2 + \alpha d)}}$$

which completes the verification of $K$ since $(J_0^2 + \alpha d)^{1/2} \leq J_0 + \sqrt{\alpha d}$.

For the counterexample, we may take $(\alpha, \beta) = (1, 2)$ and let $\nu$ be the distribution on $\mathbb{R}$ with density proportional to $e^{-dx^2 \cdot (1 + 1_{x \geq 0})/2}$. Then let $\mu = \nu^{\otimes d}$, so that $x \sim \mu$ has IID coordinates with law $\nu$. Then the mode $\mathbf{x}^*$ is indeed zero but the mean of $\nu$ is non-zero, so taking $\mathbf{v} = (1, 1, \ldots, 1)/\sqrt{d}$, a Chernoff estimate shows $\Pr^{\mathbf{x} \sim \mu}[\langle \mathbf{x}, \mathbf{v} \rangle \geq 0] \leq e^{-\Omega(d)}$. $\qquad\square$

