# OpenReview forum: "Geometry Meets Incentives: Sample-Efficient Incentivized Exploration with Linear Contexts"
_NeurIPS.cc/2025/Conference — NeurIPS 2025 spotlight_

### Official Review · Reviewer_8Eaz · 2025-06-01

**Clarity:** 2
**Significance:** 2
**Originality:** 3
**Rating:** 4
**Confidence:** 3

**Summary:**

The paper considers how geometric conditions can help the initial exploration phase. They use a Bayesian framework and some not-that-intuitive algorithms to prove a polynomial complexity. It avoids exponential dependency.

**Questions:**

Can you provide more motivation and any potential applications of the problem?

Do you have any results of corresponding lower bounds? Lower bounds can make the paper sound and more complete.

The results heavily rely on the agents being myopic. However, in practice, some agents may consider future revenue and misreport now. In other words, they may sacrifice current revenue to mislead the planner. Could you discuss the importance of this assumption and any potential difficulty extending the existing results?

The authors don't provide any experiments. Even a simulation comparing their algorithms and existing algorithms will improve the paper significantly.

**Ethical Concerns:**

["NO or VERY MINOR ethics concerns only"]

**Final Justification:**

I think the authors address most of my concerns. I hope they can give a rigorous proof of the lower bound in the final version. Based on these, I keep my score.

**Limitations:**

yes

**Quality:**

3

**Strengths And Weaknesses:**

The problem is interesting and important to learning theory. In my opinion, the results are non-trivial and inspiring. However, the paper lacks enough comparison of existing literature; for example, more discussion on "Incentivizing Exploration with Linear Contexts and Combinatorial Actions" is needed. I'm looking forward to a new but small section about the difference (new results under what conditions) between the paper and others.

---

> ### Author Rebuttal · Authors · 2025-07-30
>
> Thank you for your helpful comments and feedback. Below we address your specific questions:
>
> First, we will happily provide more discussion on the related work in the final version of the paper and apologize for the initial confusion.
>
> >*Can you provide more motivation and any potential applications of the problem?*
>
>
> Many modern online platforms recommend products to customers, for example an online retailer recommends certain products for a consumer to buy. The customers can then choose to accept the recommendation and buy the product or ignore the recommendation and buy a different product. The goal of the platform is to learn the optimal product to recommend based on customer feedback. However, if customers ignore the platform’s recommendation based on a prior belief that another option is better, then the platform may be unable to learn efficiently. This motivates our studying of Bayesian Incentive Compatible algorithms, i.e. algorithms that make recommendations in a way that incentivizes the customers to always follow the platform’s recommendation. Other similar instances of this problem are in mobile health applications that make recommendations to patients that patients can choose to follow or ignore. As mentioned at the start of [Sellke 2023], the BIC property also implies that "exploration helps everyone"; this may be especially relevant for reliability in health applications as a codification of the Hippocratic Oath. We will add further discussion of this point in the paper.
>
>
>
> >*Do you have any results of corresponding lower bounds? Lower bounds can make the paper sound and more complete.*
>
> There is of course a trivial $\lambda d$ lower bound on the number of samples needed to do $\lambda$-exploration, so we also know that $poly(d)$ is the right rate. Closing the gap between our upper bound and the linear in $d$ lower bound is a very interesting question for future work!
>
> Furthermore, the examples discussed in Appendix B actually imply lower bounds on the number of samples necessary for exploration in terms of the parameters of Assumption 2. More formally, these examples imply the following three lower bounds. For proof sketches of these three lower bounds, please see the rebuttal to **Reviewer V6tS** (they are omitted here due to space constraints).
>
> **Lower bound $c_v:$** There exist instances (i.e. prior distributions) that require $\Omega(1/c_v)$ samples for $1$-spectral exploration.
>
> **Lower bound $c_d:$** There exist instances that require $\Omega(c_d)$ samples for $1$-spectral exploration.
>
> **Lower bound $\epsilon_d:$** There exist instances that require $\Omega(\log(1/\epsilon_d))$ samples for $1$-spectral exploration.
>
> The three lower bounds described above give a more complete picture of how are algorithm is tight in a polynomial sense for these problem parameters. Furthermore, these results combined with Propositions 4-5 imply a stronger lower bound than just linear in $d$ for the distributions discussed in these propositions. We are very happy to include these more detailed discussions about problem-parameter lower bounds in the final version of the paper.
>
> >*The results heavily rely on the agents being myopic. However, in practice, some agents may consider future revenue and misreport now. In other words, they may sacrifice current revenue to mislead the planner. Could you discuss the importance of this assumption and any potential difficulty extending the existing results?*
>
> When there is a steady arrival of new agents interacting with the platform, we generally expect rational agents to act myopically (without trying to mislead the planner). This is because even if a single agent does interact with the platform multiple times, that agent is still one of many agents that the platform observes, and therefore that agent will have a very small affect on the planner. Intuitively, this suggests it should be hard for that single agent to act strategically in a way that is very advantageous for their own future interactions.
>
>
> We do also agree that the constant arrival of new agents is a strong assumption in works on Bayesian Incentive Compatible algorithms. Indeed there is a substantial literature on more realistic models of incentivized exploration, incorporating e.g. agent heterogeneity ([IMSW 19]) and partial data disclosure ([IMSW 20]) and other references from the introduction. Actually the fact that we consider linear bandits is another step in this direction, as it makes our work more realistic than the finite-armed bandit models originally used to study incentive compatibility. In any case, for theoretical understanding it is also valuable to understand simple models, as they can inform a wide range of more realistic models which each have their own quirks. For example, covariate diversity among agents is generally understood to aid exploration (see e.g. the paper *Mostly Exploration-Free Algorithms for Contextual Bandits*). We thus expect the lack of agent-specific contexts in our setting to only make the problem harder, i.e. our algorithm could be potentially finetuned and/or streamlined to take further advantage of such information. We are happy to add more discussion of these points, including potential caveats and further citations to related works that study various problem settings.
>
>
> >*The authors don't provide any experiments. Even a simulation comparing their algorithms and existing algorithms will improve the paper significantly.*
>
> First, we want to emphasize that the main contribution and algorithm of this paper are theoretical and show that it is possible to explore every dimension in $poly(d)$ time. Furthermore, there **are no** existing algorithms for this problem. That being said, the algorithm itself is relatively simple to implement. Due to the interest from reviewers, we implemented our main algorithm (Algorithm 1) along with the two helper algorithms (Algorithms 2 and 3) in Python (which only takes ~100 lines of code). We then tested the algorithm on synthetic data with varying dimensions when the prior distribution is a $d$-dimensional Gaussian that is independent across dimensions and has mean $1$ in the first dimension and mean $0$ in all other dimensions. Note that our algorithm also can be applied to prior distributions with dependencies between coordinates, however we ran experiments for the independent case as this simplifies the code significantly (and also matches the independent priors in [Sellke-Slivkins 2023]). For this simple setting, we run our algorithm with $\epsilon_d = 0.1$, $c_d = 1$, $K= 1$, and $c_v = 1$.
>
> Of course, the bound in Theorem 3 is a worst-case bound on the sample complexity, and therefore we would expect that on most instances of the problem the number of steps is significantly less than that upper bound. Furthermore, we note that the constants in our algorithm are certainly not optimal for this specific instance of the problem, yet these (relatively large) constants are what allows our theoretical results to hold for any prior distribution. While we are unable to load a graph in this rebuttal, we ran our algorithm for values of $d$ ranging from $d = 1$ to $d = 24$, and sampled $\ell^*$ from the prior distribution 100 times for each value of $d$ and then calculated the number of samples necessary to achieve $\lambda$-spectral exploration. The results show a clear quadratic dependence of the sample complexity on the dimension, and quartic scaling for the running time.
>
> | Dimension | Sample Complexity   | Time (seconds)     | Sample Complexity / $d^2$         | Time/$d^4$         |
> |----|--------|--------|--------|------------|
> | 2  | 5132   | 0.02   | 1283.00| 1.2e-03    |
> | 3  | 11781  | 0.06   | 1309.00| 7.7e-04    |
> | 4  | 21145  | 0.17   | 1321.56| 6.6e-04    |
> | 5  | 32614  | 0.38   | 1304.56| 6.0e-04    |
> | 6  | 47121  | 0.82   | 1308.92| 6.4e-04    |
> | 7  | 64278  | 1.51   | 1311.80| 6.3e-04    |
> | 8  | 84107  | 2.49   | 1314.17| 6.1e-04    |
> | 9  | 106573 | 5.43   | 1315.72| 8.3e-04    |
> | 10 | 131696 | 6.19   | 1316.96| 6.2e-04    |
> | 11 | 159505 | 10.34  | 1318.22| 7.1e-04    |
> | 12 | 189954 | 14.30  | 1319.13| 6.9e-04    |
> | 13 | 223085 | 20.67  | 1320.03| 7.2e-04    |
> | 14 | 258838 | 26.72  | 1320.60| 7.0e-04    |
> | 15 | 297248 | 35.88  | 1321.10| 7.1e-04    |
> | 16 | 338359 | 46.61  | 1321.71| 7.1e-04    |
> | 17 | 382100 | 57.79  | 1322.15| 6.9e-04    |
> | 18 | 428515 | 74.51  | 1322.58| 7.1e-04    |
> | 19 | 477573 | 93.26  | 1322.92| 7.2e-04    |
> | 20 | 529270 | 114.13 | 1323.18| 7.1e-04    |
> | 21 | 583683 | 137.98 | 1323.54| 7.1e-04    |
> | 22 | 640716 | 168.16 | 1323.79| 7.2e-04    |
> | 23 | 700428 | 201.52 | 1324.06| 7.2e-04    |
> | 24 | 762755 | 241.09 | 1324.23| 7.3e-04    |
>
> We do want to emphasize that for a specific instance of our problem (i.e. a specific prior), a user could tune the worst-case constants used in the algorithm to get faster $\lambda$-spectral exploration. However, the above table just uses the default constants of the algorithm with the goal of showing that
> - the algorithm is implementable in relatively few lines of python code
> - the algorithm is efficient to run for reasonably large values of $d$
> - in practice, the number of steps grows polynomially in $d$ at a much better rate than the worst-case bound in our theoretical results (in this experiment we see a roughly quadratic relationship between the dimension and the number of steps). We are  happy to add this result to the final version of our paper with more details about the implementation and with graphs showing the trends.

---

> > ### Comment · Reviewer_8Eaz · 2025-08-06
> >
> > Thank you for the authors' response. A rigorous proof for the lower bound would certainly make the final version more complete. Therefore, I am inclined to keep my score, which is already positive.

---

### Official Review · Reviewer_1V5y · 2025-06-23

**Clarity:** 2
**Significance:** 3
**Originality:** 2
**Rating:** 4
**Confidence:** 2

**Summary:**

The paper "Geometry Meets Incentives: Sample-Efficient Incentivized Exploration with Linear Contexts" proposes to explore the problem of a planner that provides recommendations to selfish and rational agents that aim only to maximize their own expected reward based on its posterior. In order to minimize cumulative regret, the planner needs to incentivize exploration in its recommendations. However, if these recommandations are not in line with agent's own interest, they won't be followed and the process does not collect the expected exploration knowledge. The goal in the paper is to build a BIC planner as a linear bandit, which ensures that recommandations allow to cover the entire action space (in the sense of the \lambda-spectral exploration following [Sellke et al., 2023]. The proposal strongly build on  [Sellke et al., 2023], that proposes to divide the process in two steps of pure exploration followed by a Thompson Sampling procedure. The core idea is that after enough free exploration steps, where the agent has no information at all to decline the recommandation, the parameter posterior is sufficiently concentrated to make sampled recommendations always the most credible from the agent's perspective. Authors show that under some assumptions over the actions space and rewards distribution, the length of the exploration phase can be significantly reduced. More specifically, they propose an algorithm that performs exploration coverage without greatly fewer initial steps of initial data collection.

**Questions:**

see strengths and weaknesses

**Ethical Concerns:**

["NO or VERY MINOR ethics concerns only"]

**Final Justification:**

After discussions with the authors I think the paper can be useful for the community, despite some lack of pedagogy in its current form.
I raised my score to 4, assuming the authors implement the presentation improvements discussed in their final version of the paper.

**Limitations:**

.

**Paper Formatting Concerns:**

.

**Quality:**

2

**Strengths And Weaknesses:**

Strengths :
- Significant theoretical contribution (I did not check all the proofs though)

Weaknesses :

- The paper is only theoretical. It would have been very useful to get some experimental results of the proposed algorithm, both for use cases comprehension and also to capture the benefits of the proposal in various settings, including some of them not fully respecting working assumtpions
- Lack of pedagogy: the setting is not clearly / formally defined. For readers that do not know the incentivized exploration problem, it is very hard to capture what are the objectives of this kind of problem. I had to read [Sellke, 2023] to understand the problem, what is given to agents, what is the goal, etc. In comparison [Sellke, 2023] is far more clear in presenting the problem.
- Beyond the lack of formal presentation of the problem, the paper also uses many notations that are never defined : e.g.,  the symbol used for positive semidefinite partial order on symmetric matrices (I had to find it in Sellke, 2023), the symbol used to define product tensors at order 2, $\Omega$, etc
- While I find the problem theoretically interesting, I am not fully convinced by its real interest for pratical applications, as users can interact multiple times, have their own interests, their own history, the knowledge and understanding of the planner's algorithm, etc. Practical uses should be more discussed in the paper.
- The central assumption of the algorithm is that actions live in a d-dimentionnal unit ball. Isn't it limitative for some applications ? And more importantly, in that setting what would justify a prior that would not be uniform over every vector from that ball ? This can be very limitative for the paper if no reasonnable pratical justification for this  can be found since taking a fully non informative prior is a trivial setting, for which TS is BIC without any initial exploration phase. Or am I wrong on this point ?

---

> ### Author Rebuttal · Authors · 2025-07-30
>
> Thank you for your helpful comments and feedback. Below we address your specific questions:
>
> >The paper is only theoretical. It would have been very useful to get some experimental results of the proposed algorithm [...]
>
> First, we want to emphasize that the main contribution and algorithm of this paper are theoretical and show that it is possible to explore every dimension in $poly(d)$ time. That being said, the algorithm itself is relatively simple to implement. Due to the interest from reviewers, we implemented our main algorithm (Algorithm 1) along with the two helper algorithms (Algorithms 2 and 3) in Python (which only takes ~100 lines of code). We then tested the algorithm on synthetic data with varying dimensions when the prior distribution is a $d$-dimensional Gaussian that is independent across dimensions and has mean $1$ in the first dimension and mean $0$ in all other dimensions. Note that our algorithm also can be applied to prior distributions with dependencies between coordinates, however we ran experiments for the independent case as this simplifies the code significantly (and also matches the independent priors in [Sellke-Slivkins 2023]). For this simple setting, we run our algorithm with $\epsilon_d = 0.1$, $c_d = 1$, $K= 1$, and $c_v = 1$.
>
> Of course, the bound in Theorem 3 is a worst-case bound on the sample complexity, and therefore we would expect that on most instances of the problem the number of steps is significantly less than that upper bound. Furthermore, we note that the constants in our algorithm are certainly not optimal for this specific instance of the problem, yet these (relatively large) constants are what allows our theoretical results to hold for any prior distribution. While we are unable to load a graph in this rebuttal, we ran our algorithm for values of $d$ ranging from $d = 1$ to $d = 24$, and sampled $\ell^*$ from the prior distribution 100 times for each value of $d$ and then calculated the number of samples necessary to achieve $\lambda$-spectral exploration. The results show a clear quadratic dependence of the sample complexity on the dimension, and quartic scaling for the running time.
>
> | Dimension | Sample Complexity   | Time (seconds)     | Sample Complexity / $d^2$         | Time/$d^4$         |
> |----|--------|--------|--------|------------|
> | 2  | 5132   | 0.02   | 1283.00| 1.2e-03    |
> | 3  | 11781  | 0.06   | 1309.00| 7.7e-04    |
> | 4  | 21145  | 0.17   | 1321.56| 6.6e-04    |
> | 5  | 32614  | 0.38   | 1304.56| 6.0e-04    |
> | 6  | 47121  | 0.82   | 1308.92| 6.4e-04    |
> | 7  | 64278  | 1.51   | 1311.80| 6.3e-04    |
> | 8  | 84107  | 2.49   | 1314.17| 6.1e-04    |
> | 9  | 106573 | 5.43   | 1315.72| 8.3e-04    |
> | 10 | 131696 | 6.19   | 1316.96| 6.2e-04    |
> | 11 | 159505 | 10.34  | 1318.22| 7.1e-04    |
> | 12 | 189954 | 14.30  | 1319.13| 6.9e-04    |
> | 13 | 223085 | 20.67  | 1320.03| 7.2e-04    |
> | 14 | 258838 | 26.72  | 1320.60| 7.0e-04    |
> | 15 | 297248 | 35.88  | 1321.10| 7.1e-04    |
> | 16 | 338359 | 46.61  | 1321.71| 7.1e-04    |
> | 17 | 382100 | 57.79  | 1322.15| 6.9e-04    |
> | 18 | 428515 | 74.51  | 1322.58| 7.1e-04    |
> | 19 | 477573 | 93.26  | 1322.92| 7.2e-04    |
> | 20 | 529270 | 114.13 | 1323.18| 7.1e-04    |
> | 21 | 583683 | 137.98 | 1323.54| 7.1e-04    |
> | 22 | 640716 | 168.16 | 1323.79| 7.2e-04    |
> | 23 | 700428 | 201.52 | 1324.06| 7.2e-04    |
> | 24 | 762755 | 241.09 | 1324.23| 7.3e-04    |
>
> We do want to emphasize that for a specific instance of our problem (i.e. a specific prior), a user could tune the worst-case constants used in the algorithm to get faster $\lambda$-spectral exploration. However, the above table just uses the default constants of the algorithm with the goal of showing that
> - the algorithm is implementable in relatively few lines of python code
> - the algorithm is efficient to run for reasonably large values of $d$
> - in practice, the number of steps grows polynomially in $d$ at a much better rate than the worst-case bound in our theoretical results (in this experiment we see a roughly quadratic relationship between the dimension and the number of steps). We are  happy to add this result to the final version of our paper with more details about the implementation and with graphs showing the trends.
>  - Please refer to the rebuttal to **Reviewer Rjs1** for a code snippet (not included here due to char limit).
>
> >Lack of pedagogy: the setting is not clearly / formally defined. For readers that do not know the incentivized exploration problem, it is very hard to capture what are the objectives of this kind of problem. [...] Beyond the lack of formal presentation of the problem, the paper also uses many notations that are never defined [...]
>
> We sincerely apologize for this. Paralleling [Sellke 2023], we will add clearer presentation, motivation and other discussion for the problem formulation, as well as definitions of these notations, to the final version of the paper.
>
> >While I find the problem theoretically interesting, I am not fully convinced by its real interest for pratical applications, as users can interact multiple times, have their own interests, their own history, the knowledge and understanding of the planner's algorithm, etc)
>
> We agree that the constant arrival of new agents is a strong assumption in works on Bayesian Incentive Compatible algorithms. Firstly, there is a substantial literature on more realistic models of incentivized exploration, incorporating e.g. agent heterogeneity ([IMSW 19]) and partial data disclosure ([IMSW 20]) and other references from the introduction. Actually the fact that we consider linear bandits makes our work more realistic than the original finite-armed bandit models used to study incentive compatibility. In any case, for theoretical understanding it is also valuable to understand simple models, as they can inform a wide range of more realistic models which each have their own quirks. For example, covariate diversity among agents is generally understood to aid exploration (see e.g. the paper *Mostly Exploration-Free Algorithms for Contextual Bandits*). Thus we expect the simple nature of our setting to only make the problem harder, meaning that our algorithms could be potentially finetuned and/or streamlined if the model were refined to be more realistic. We are happy to add more discussion of these points, including potential caveats and further citations to related works that study various problem setting.
>
> Regarding applications, many modern online platforms recommend products to customers, for example an online retailer recommends certain products for a consumer to buy. The customers can then choose to accept the recommendation and buy the product or ignore the recommendation and buy a different product. The goal of the platform is to learn the optimal product to recommend based on customer feedback. However, if customers ignore the platform’s recommendation based on a prior belief that another option is better, then the platform may be unable to learn efficiently. This motivates our studying of Bayesian Incentive Compatible algorithms, i.e. algorithms that make recommendations in a way that incentivizes the customers to always follow the platform’s recommendation. Other similar instances of this problem are in mobile health applications that make recommendations to patients that patients can choose to follow or ignore. As mentioned at the start of [Sellke 2023], the BIC property also implies that "exploration helps everyone"; this may be especially relevant for reliability in health applications as a codification of the Hippocratic Oath. We will add further discussion of this point in the paper.
>
> >The central assumption of the algorithm is that actions live in a d-dimentionnal unit ball. Isn't it limitative for some applications ? And more importantly, in that setting what would justify a prior that would not be uniform over every vector from that ball ? This can be very limitative for the paper if no reasonnable pratical justification for this can be found since taking a fully non informative prior is a trivial setting, for which TS is BIC without any initial exploration phase. Or am I wrong on this point ?
>
> First, we want to clarify: it is definitely **not known** that Thompson Sampling is BIC on its own for spherical action sets and uniform priors, even though all arms have the same mean reward under the prior. The **only** bandit setting where this kind of result has been proved is for discrete action sets with independent priors across the arms (in [Sellke-Slivkins 2023]). The proof there applies correlation inequalities to the increments of two Doob martingales, and strongly makes use of this independence. The work [HNSW22] follows up on [Sellke-Slivkins 2023] considers combinatorial semi-bandits (where actions are sets of multiple arms) but does not include a result of this type. [Sellke 2023] considers linear bandits and provides a **counterexample** showing the Thompson sampling may be BIC at time 1 but not time 2. It is plausible that a linear bandit extension of this result is true under some rotational symmetry conditions, but it would be a new, very interesting theorem.
>
> Regarding the prior, note that in our setting the prior encapsulates the knowledge that a fresh agent enters the system with. Thus in modeling, **the prior should take into account all publicly available knowledge**. For example, the publication of a scientific paper or news article related to some of the actions should certainly influence the prior. While it might be challenging to describe exactly the correct prior to use in a particular real-life setting, we believe this illustrates the need for flexibility in handling rather general priors. Note that using a completely wrong prior means that the BIC guarantees we pursue are rendered invalid. This is in contrast with ordinary bandit problems, where the only downside to using a simpler prior is potentially slower learning in the early stages.

---

> > ### Comment · Reviewer_1V5y · 2025-08-05
> >
> > Thanks to the authors for their insightful answers. I will probably increase my score.
> >
> > However, going back to [Selke, 2023], I cannot see anywhere where they say assuming that l*_i is independent from l*_j.
> > Contrarily, they mention on page 2 : "We make no assumptions of independence but instead require natural geometric conditions, ...".
> >
> > Also, could authors summarize precisely what differs from their setting and the one in [Selke,2023] ? Is the better complexity of the exploration phase rather induced by a more convenient setting, or a better algorithm ? Is the considered setting a special case of the more general one studied in  [Selke,2023] ?

---

> > > ### Author Response · Authors · 2025-08-05
> > >
> > > Thank you again for your helpful feedback! We hope that the responses below answer your questions, but let us know if you have any remaining concerns.
> > >
> > >
> > > > However, going back to [Selke, 2023], I cannot see anywhere where they say assuming that l*_i is independent from l*_j. Contrarily, they mention on page 2 : "We make no assumptions of independence but instead require natural geometric conditions, ...".
> > >
> > > You are correct that [Sellke, 2023] does not assume independence for the elements of $\ell^*$ (they do make some strong assumptions which we discuss below). However, many previous works on these types of problems (e.g.  [Sellke and Slivkins 23]) do assume independent priors for the arms, as discussed in our related work section. We are happy to clarify this more explicitly in our paper.
> > >
> > >
> > >
> > > > Also, could authors summarize precisely what differs from their setting and the one in [Selke,2023] ?  Is the considered setting a special case of the more general one studied in [Selke,2023] ?
> > >
> > >
> > >
> > > The main results from [Sellke, 2023] require the "natural geometric condition" that the prior for $\ell^\*$ is uniformly random on a convex body with bounded aspect ratio. Note while this is not the same as assuming independence, this is still a fairly restricted class of prior for $\ell^\*$. In our paper, we make much more general assumptions on the prior distribution of $\ell^*$ that include a broader class of priors including sub-gaussian distributions, more complex dependencies between arms, etc.. So to summarize, the prior distributions we study are more general compared to those of [Sellke, 2023] but the action sets we study are more restricted compared to those in [Sellke, 2023].
> > >
> > >
> > >
> > > We also want to emphasize that our results are not at all a special case of [Sellke, 2023]. Our main result is about how to achieve initial exploration, while the main result of [Sellke, 2023] is that if the initial exploration is already done, then Thompson sampling is BIC. Importantly, [Sellke, 2023] does **not give an algorithm for initial exploration**, so their result cannot be directly compared to ours. This is also why, as we discuss in detail in our conclusion, combining our results with the results of [Sellke, 2023] gives the first end-to-end BIC algorithm for this setting.
> > >
> > >
> > >
> > > >Is the better complexity of the exploration phase rather induced by a more convenient setting, or a better algorithm ?
> > >
> > > [Sellke, 2023] gives a specific example of a prior and an action space where initial exploration takes exponentially many samples. Our results are able to do exploration in polynomially-many samples in part because we assume a smooth action space (thereby excluding the example in [Sellke, 2023])  and in part because we give the first BIC algorithm for achieving initial exploration in this problem. As discussed above, [Sellke, 2023] does not even give an algorithm for doing the initial exploration, and therefore there is no other algorithm to which we can compare our algorithm.

---

> > > > ### Comment · Reviewer_1V5y · 2025-08-08
> > > >
> > > > Thanks for these clarifications !
> > > > I will raise my score to 4, assuming the authors implement the presentation improvements discussed in their final version of the paper.

---

### Official Review · Reviewer_V6tS · 2025-06-29

**Clarity:** 3
**Significance:** 3
**Originality:** 3
**Rating:** 5
**Confidence:** 3

**Summary:**

The paper studies the incentivized exploration problem in linear bandits, where a principal needs to gather information by recommending actions to a stream of myopic self-interested agents while remaining Bayesian Incentive Compatible (BIC).
Earlier work (Sellke ’23) showed a hard instance in which the amount of data needed before one can safely switch to Thompson sampling could be exponentially many rounds. The authors show that this exponential barrier is not intrinsic under mild conditions. More specifically, when the action set is a unit ball, there exists an algorithm whose sample complexity is only polynomial in the dimension.

**Questions:**

- Can authors discuss the tightness of dependence on d?

- What's the computational cost and memory cost of the proposed algorithms?

- If the reward mean is not as a linear function but as a general function, can we get similar results under the same conditions? If not, what's could be the main challenges?

**Ethical Concerns:**

["NO or VERY MINOR ethics concerns only"]

**Final Justification:**

This paper makes solid theoretical contributions and all my concerns are addressed by authors in their rebuttal.

**Limitations:**

yes

**Quality:**

3

**Strengths And Weaknesses:**

Strengths:

- Given the hard result by (Sellke ’23), authors show that when the action set is a unit ball, there exists an algorithm which can break the exponential barrier under mild conditions. Authors present two illustrative examples that satisfy those conditions, including r-regular distributions and a type of log-concave distributions with a certain smoothness. To me, this finding is interesting since it opens some new problems, e.g., can we find a broader class of action sets?

- The algorithm and subroutines are easy to follow. I appreciate the explanations and intuitions behind the algorithm design.

- The paper is generally well-written and well-organized. Prior works are clearly contrasted.

Weaknesses:

- The main results rely on the unit ball assumption. Extending to general smooth convex bodies is hinted at but left open.

- Though the sample complexity is polynomial in parameter d, it suffers large poly(d) factors. It could be helpful to discuss the tightness of dependence on d.

- There is no discussion or analysis on the computational cost and memory cost.

Other minors:

- It could be better to split section 1.1 into two parts: one is problem setting and the other one shows your main results. Mixing them together seems to be a mess to me.

- Adding a link of references and equations is helpful for readers. There is a ref issue in line 218 and 226, it should be Figure 1 instead of Figure 2.3.

---

> ### Author Rebuttal · Authors · 2025-07-30
>
> Thank you for your helpful comments and feedback. Below we address your specific questions:
>
> >*Can authors discuss the tightness of dependence on d?*
>
> There is of course a trivial $\lambda d$ lower bound on the number of samples needed to do $\lambda$-exploration, so we also know that $poly(d)$ is the right rate. Closing the gap between our upper bound and the linear in $d$ lower bound is a very interesting question for future work!
>
> Furthermore, the examples discussed in Appendix B actually imply lower bounds on the number of samples necessary for exploration in terms of the parameters of Assumption 2. More formally, these examples imply the following three lower bounds, and we include a brief proof sketch of each below.
>
>
>
> **Lower bound $c_v:$** There exist instances  (i.e. prior distributions) that require $\Omega(1/c_v)$ samples for $1$-spectral exploration.
>
> **Proof sketch**
> Consider the following example with $d=2$ and where the coordinates of $\ell^\ast$ are independent (and assume that $c_d < 1, \epsilon_d < 1, c_v < 1$):
>
> $$\\ell^\\ast_1 = \\begin{cases}  -c_d & \\text{w.p. $\epsilon_d$} \newline       1 & \\text{w.p. $1-\\epsilon_d$} \\end{cases},\\quad\\quad
> \\ell^\ast_2 = \\begin{cases}  -c_v & \\text{w.p. $1/2$} \newline         c_v & \\text{w.p. $1/2$} \\end{cases}$$
>
> Note that if $E[\ell_1^* \mid \psi] = 1-O(\epsilon_d) \ge  0.5$ for sufficiently small $\epsilon_d$, then no optimal action will ever put weight more than $O(c_v)$ on the second coordinate because the magnitude of $\ell_2^*$ is bounded by $c_v$. This implies that we must need $1/c_v$ steps in order to guarantee $1$-spectral exploration.
>
> $\blacksquare$
>
>
>
>
>
>
>
> **Lower bound $c_d:$** There exist instances that require $\Omega(c_d)$ samples for $1$-spectral exploration.
>
> **Proof Sketch**
>
> Consider the following example
>
> $$\ell^\ast_1 = \begin{cases}  -c_d & \text{w.p. $\epsilon_d$}\newline   2c_d & \text{w.p. $2\epsilon_d$} \newline       1 & \text{w.p. $1-3\epsilon_d$} \end{cases},\quad\quad
> \ell^\ast_2 = \begin{cases}  -c_v & \text{w.p. $1/2$}\newline         c_v & \text{w.p. $1/2$} \end{cases}$$
>
> Once again, the first action must be $e_1$. Furthermore, in this case we need $O(poly(1/c_d))$ actions of $e_1$ in order to decrease the conditional expectation of $\ell^\ast_1$ to be $0$, as we need this many samples to be able to effectively distinguish between $\ell_1^* = -c_d$ and $\ell_1^* = 2c_d$. This implies that we require $O(poly(1/c_d))$ samples to explore the second dimension in this example.
> $\blacksquare$
>
>
>
> **Lower bound $\epsilon_d:$** There exist instances that require $\Omega(\log(1/\epsilon_d))$ samples for $1$-spectral exploration.
>
>
> **Proof sketch:**
>
> This proof is more subtle and requires inductive control of the information gain on the 2nd coordinate. We provide a preparatory lemma and corollary.
>
> **Lemma**: let $L=±c$ be uniformly random for some $|c|≤1$. Suppose we receive noisy observations $r_i = s_i L + Z_i$ for a sequence $r_1,...$ that is adapted to the filtration $\mathcal F_t$ generated by $(s_1,r_1,s_2,r_2,...,s_t,r_t)$. (I.e. the signal strengths $s_i$ may depend on the past.) Let $T_t=\sum_{i=1}^t s_i^2$. Then the expected information gain on $L$ is at most $O(E[T_t])$.
>
>
> Proof: this is a special case of observing Brownian motion with drift $L$ up to the random stopping time $T_t$. (Since observing $r_i = s_i L + Z_i$ is equivalent to observing Brownian motion with drift $L$ for time $s_i^2$, up to rescaling.) Let $Q_±(T_t)$ be the laws of Brownian motion with drifts $±c$ up to time $T_t$. By symmetry the expected information gain can be computed assuming $L=c$, and is bounded by
>
> $$\begin{align}
> &E[KL(Q_+(T_t),[Q_+(T_t)+Q_-(T_t)]/2)]
> \\
> &\stackrel{\text{Convexity of KL}}{\leq}
> E[KL(Q_+(T_t),Q_+(T_t))]
> +
> E[KL(Q_+(T_t),Q_{-}(T_t))]
> \\
> &\leq
> E[T_t].
> \end{align}$$
>
>
> Here we used $E[KL(Q_+(T_t)|Q_-(T_t))]=c^2 E[T_t]≤E[T_t]$ by Girsanov's theorem. $\blacksquare$
>
>
> **Corollary**: in the setting of the previous lemma, let  $C_t = E[L|\mathcal F_t] = E[L|(s_1,...,s_t,r_1,...,r_t)].$ Then $E[C_t^2] \leq O(E[T_t])$.
>
> Proof: This follows from the previous lemma: the expected relative entropy between the prior and posterior distributions of $L$ is precisely the information gain. In turn, the relative entropy is a strictly convex and even function of $C_t$ which is $\Omega(C_t^2)$. $\blacksquare$
>
>
>
> We now return to the first example prior from the $c_v$ lower bound, and apply the Corollary to the observations of the 2nd coordinate. To make the application direct, whenever an action $(a_{1,t},a_{2,t})$ is played, we replace the noisy reward observation with separate observations for both $(a_{1,t},0)$ and $(0,a_{2,t})$ (with half the noise level, which only affects constant factors in the argument). This gives strictly more information since the two separate observations can be added to recover the original observation.
> We let $C_t,\mathcal F_t$ from above correspond to observations in the second coordinate.
>
>
> At each time $t$, by Jensen's inequality on the convex function $f(x)=(1/3 - x)_+$, we see that for any signal $\psi$:
>
>
> $$P[E[\ell_1 | \psi]≤1/2]
> \leq
> O(E[f(E[\ell_1 | \psi])])
> \leq
> O(E[f(\ell_1)])
> \leq
> O(\epsilon_d).$$
>
>
> On the main high-probability event that $E[\ell_1 | \psi]≥1/2$, the 2nd coordinate of $Exploit(\psi)$ has absolute value at most $O(|C_t|)$. Via the Lemma and Corollary above, it follows that
>
>
> $$\mathbb E[T_{t+1}] - \mathbb E[T_t] ≤ O(\mathbb E[T_t] + \epsilon_d).$$
>
> Namely the case $\{E[\ell_1 | \psi]≤1/2\}$ contributes $O(\epsilon_d)$ while the remaining case contributes $O(\mathbb E[C_t^2])\leq O(\mathbb E[T_t])$.
>
>
> Since $T_0=0$, this implies by induction that
>
>
> $$\mathbb E[T_t] ≤ e^{O(t)} \epsilon_d.$$
>
>
> Finally note that we need $T_t ≥ 1$ for $1$-spectral exploration. Indeed, $1$-spectral exploration requires that the actions $a_1,\dots,a_t$ satisfy
> $$
> \sum_{i=1}^t \langle a_t^{\top}, M a_t\rangle
> \geq
> Tr(M)
> $$
> for all positive semi-definite matrices $M$, and taking $M=\begin{pmatrix} 0 & 0 \newline 0 & 1\end{pmatrix}$ recovers the claim. In all this gives the desired $\log(1/\epsilon_d)$ lower bound. $\blacksquare$
>
>
>
> The three lower bounds described above give a more complete picture of how are algorithm is tight in a polynomial sense for these problem parameters. Furthermore, these results combined with Propositions 4-5 imply a stronger lower bound than just linear in $d$ for the distributions discussed in these propositions. We are very happy to include these more detailed discussions about problem-parameter lower bounds in the final version of the paper.
>
> >*What's the computational cost and memory cost of the proposed algorithms?*
>
>
> The memory of the algorithms as-written scales linearly with the number of steps (which is of course polynomial in $d$), as the only information that needs to be stored are the action and reward from each step. The computational cost is also also polynomial in $d$, with the exception potentially of computing the function $f$ used in Line 4 of Algorithm 5. In our paper, we only prove the existance of such a function $f$. However, for many high-dimensional distributions (for example correlated Gaussians) this can be found in polynomial (in $d$) time using a simple linear program. In fact, for independent Gaussians (as in our experimental example), this $f$ can be directly computed in constant time without even needing a linear program.
>
> We mention that this situation is similar to [Sellke-Slivkins 2023] in the $K$-armed setting, where the main algorithm is also shown to be efficient only in special cases. Their examples involve Beta prior distributions (see Appendix G therein) rather than Gaussians.
>
> >*If the reward mean is not as a linear function but as a general function, can we get similar results under the same conditions? If not, what's could be the main challenges?*
>
> This is a great question, and indeed [Sellke 2023] briefly considers generalized linear bandits where a non-linear link function is applied to the inner product. The link function there is required to be strictly increasing with uniformly upper and lower bounded derivative. We do not see any conceptual barriers to extending our results to a similar class of functions, since the main property we rely on is the sensitivity of the optimal action with respect to the posterior mean reward. At the same time it is not straightforward and would require re-thinking many intermediate steps in the algorithm and analysis. (Both of these comments parallel the discussion at the end of page 3 of potentially generalizing to smooth non-spherical action sets.)

---

> > ### Comment · Reviewer_V6tS · 2025-08-05
> >
> > I thank the authors for detailed response and addressing my questions. I believe this is a solid theory paper and will raise my score to 5.

---

### Official Review · Reviewer_Rjs1 · 2025-06-29

**Clarity:** 3
**Significance:** 4
**Originality:** 3
**Rating:** 5
**Confidence:** 4

**Summary:**

This work studies the problem of incentivized exploration in linear bandits. Prior work of Sellke 2023 has shown that Thompson Sampling is BIC with an exponentially (in dimensions) number of initial samples. In this work, the authors show that it is possible to improve on this initial sample complexity when the action set is contained in a $d$-dimensional unit ball. Specifically, the authors provide an initial sampling algorithm with polynomial sample complexity. This is a significant improvement over the current literature on incentivized exploration in bandits.

**Questions:**

Please answer the questions listed above in the weaknesses. Furthermore, do the authors have any insight on whether this initial sample complexity is tight or can it still be improved in the future?

**Ethical Concerns:**

["NO or VERY MINOR ethics concerns only"]

**Final Justification:**

After seeing the responses to my review and other reviewers' comments, I still strongly recommend this paper for acceptance. This paper presents a very strong theoretical contribution to the problem of initial sampling for incentivized bandits exploration, and the authors have included a minimal working experiment as requested in the responses.

**Limitations:**

Yes.

**Paper Formatting Concerns:**

N/A.

**Quality:**

4

**Strengths And Weaknesses:**

Strengths:
- The improvement over prior work is a significant contribution.
- The paper is generally well-written.
- The problem of incentivized exploration in linear bandits is not new, but the initial sampling algorithm is novel.
- The proof sketches are intuitive and easy to follow.

Weaknesses:
- The paper did not include any experiments.
- The paper focuses solely on the initial sampling exploration, and needs to be combined with Sellke 2023's result to have an end-to-end BIC algorithm. Thus, the contribution may seem a bit incremental.
- The paper did not explain the constants $c_v, c_d, \epsilon_d$ in assumption 2. Having more elaborate justification of what each constant means and how they should be chosen in practice would help with understanding these assumptions.

---

> ### Author Rebuttal · Authors · 2025-07-30
>
> >*The paper did not include any experiments.*
>
> First, we want to emphasize that the main contribution and algorithm of this paper are theoretical and show that it is possible to explore every dimension in $poly(d)$ time. That being said, the algorithm itself is relatively simple to implement. Due to the interest from reviewers, we implemented our main algorithm (Algorithm 1) along with the two helper algorithms (Algorithms 2 and 3) in Python (which only takes ~100 lines of code). We then tested the algorithm on synthetic data with varying dimensions when the prior distribution is a $d$-dimensional Gaussian that is independent across dimensions and has mean $1$ in the first dimension and mean $0$ in all other dimensions. Note that our algorithm also can be applied to prior distributions with dependencies between coordinates, however we ran experiments for the independent case as this simplifies the code significantly (and also matches the independent priors in [Sellke-Slivkins 2023]). For this simple setting, we are able to run our algorithm with the constants of $\epsilon_d = 0.1$, $c_d = 1$, $K= 1$, and $c_v = 1$.
>
> Of course, the bound in Theorem 3 is a worst-case bound on the sample complexity, and therefore we would expect that on most instances of the problem the number of steps is significantly less than that upper bound. Furthermore, we note that the constants in our algorithm are certainly not optimal for this specific instance of the problem, yet these (relatively large) constants are what allows our theoretical results to hold for any prior distribution. While we are unable to load a graph in this rebuttal, we ran our algorithm for values of $d$ ranging from $d = 1$ to $d = 24$, and sampled $\ell^*$ from the prior distribution 100 times for each value of $d$ and then calculated the number of samples necessary to achieve $\lambda$-spectral exploration. The results show a clear quadratic dependence of the sample complexity on the dimension, and quartic scaling for the running time.
>
> | Dimension | Sample Complexity   | Time (seconds)     | Sample Complexity / $d^2$         | Time/$d^4$         |
> |----|--------|--------|--------|------------|
> | 2  | 5132   | 0.02   | 1283.00| 1.2e-03    |
> | 3  | 11781  | 0.06   | 1309.00| 7.7e-04    |
> | 4  | 21145  | 0.17   | 1321.56| 6.6e-04    |
> | 5  | 32614  | 0.38   | 1304.56| 6.0e-04    |
> | 6  | 47121  | 0.82   | 1308.92| 6.4e-04    |
> | 7  | 64278  | 1.51   | 1311.80| 6.3e-04    |
> | 8  | 84107  | 2.49   | 1314.17| 6.1e-04    |
> | 9  | 106573 | 5.43   | 1315.72| 8.3e-04    |
> | 10 | 131696 | 6.19   | 1316.96| 6.2e-04    |
> | 11 | 159505 | 10.34  | 1318.22| 7.1e-04    |
> | 12 | 189954 | 14.30  | 1319.13| 6.9e-04    |
> | 13 | 223085 | 20.67  | 1320.03| 7.2e-04    |
> | 14 | 258838 | 26.72  | 1320.60| 7.0e-04    |
> | 15 | 297248 | 35.88  | 1321.10| 7.1e-04    |
> | 16 | 338359 | 46.61  | 1321.71| 7.1e-04    |
> | 17 | 382100 | 57.79  | 1322.15| 6.9e-04    |
> | 18 | 428515 | 74.51  | 1322.58| 7.1e-04    |
> | 19 | 477573 | 93.26  | 1322.92| 7.2e-04    |
> | 20 | 529270 | 114.13 | 1323.18| 7.1e-04    |
> | 21 | 583683 | 137.98 | 1323.54| 7.1e-04    |
> | 22 | 640716 | 168.16 | 1323.79| 7.2e-04    |
> | 23 | 700428 | 201.52 | 1324.06| 7.2e-04    |
> | 24 | 762755 | 241.09 | 1324.23| 7.3e-04    |
>
>
>
> We do want to emphasize that for a specific instance of our problem (i.e. a specific prior), a user could tune the worst-case constants used in the algorithm to get faster $\lambda$-spectral exploration. However, the above table just uses the default constants of the algorithm with the goal of showing that
> - the algorithm is implementable in relatively few lines of python code
> - the algorithm is efficient to run for reasonably large values of $d$
> - in practice, the number of steps grows polynomially in $d$ at a much better rate than the worst-case bound in our theoretical results (in this experiment we see a roughly quadratic relationship between the dimension and the number of steps). We are of course happy to add this result to the final version of our paper with more details about the implementation and with graphs showing the trends.
> - One reason why the sample complexity is so much better here than in our worst-case bound is that a factor of $d^4$ in our worst-case bound comes from applying Lemma 8 in our paper. This lemma is very much a worst-case bound for how much exploration we gain in each step in terms of the eigenvalues, and comes from subtleties of high-dimensional geometry. In most practical settings we would not expect to need as many steps as implied by that lemma.
>
> We don't include all of the code due to space constraints, but the main algorithm corresponds to the algorithm in the paper pretty clearly as follows
> ```
>
> def bic_exploration(d, lambda_val, eps_d, c_d, cLA2, cLA4, K, cv, ell_star, ell0_mean):
>     kappa = max(1 / (lambda_val * cLA4), (4 * d * (K * np.sqrt(np.pi) + 1) * (1 + 1 / lambda_val)) / (cv ** 2 / (8 * np.pi)))
>     v_list = [np.eye(d)[0]]  # Start with e1
>     q_list = [[0]*int(kappa)]
>
>     # Initial exploration with e1
>     for i in range(int(kappa)):
>         q_list[0][i] = np.dot(v_list[0], ell_star)+ np.random.normal()  # reward proxy
>
>     t = int(kappa)
>     j = 0
>     M = lambda V: sum(np.outer(v, v) for v in V)
>     while np.min(np.linalg.eigvals(M(v_list))) \le lambda_val:
>         eigvals, eigvecs = np.linalg.eigh(M(v_list))
>         idx = np.argsort(-eigvals)
>         lambdas = eigvals[idx]
>         w = eigvecs[:, idx].T
>         ell_lambda = np.sum(lambdas >= lambda_val)
>
>         a, t = initial_exploration(w, lambdas[:ell_lambda], v_list, q_list, t, cLA4, lambda_val, eps_d, c_d, K, ell_star, j+1)
>         while np.linalg.norm(a - w[:ell_lambda].T @ (w[:ell_lambda] @ a)) \le np.sqrt(lambda_val):
>             a, t = exponential_growth(a, w, lambdas[:ell_lambda], v_list, q_list, t, ell0_mean, cLA2, lambda_val, ell_star, j+1)
>         v_list.append(a)
>         q_list.append([0]*int(kappa))
>         for i in range(int(kappa)):
>             q_list[-1][i] = np.dot(a, ell_star)+ np.random.normal()
>         t += int(kappa)
>         j += 1
>     return t
> ```
>
>
>
>
> >*The paper focuses solely on the initial sampling exploration, and needs to be combined with Sellke 2023's result to have an end-to-end BIC algorithm. Thus, the contribution may seem a bit incremental.*
>
> We respectfully disagree that the contribution is incremental. The key technical gap left open by Sellke 2023 was that its Thompson‑sampling stage becomes BIC only after collecting an **exponentially large** amount of initial data in the dimension d. Consequently, there was no sample‑efficient end‑to‑end algorithm in high dimensions prior to our work, and it was not clear whether such guarantees should even be possible.
>
> >*The paper did not explain the constants in assumption 2. Having more elaborate justification of what each constant means and how they should be chosen in practice would help with understanding these assumptions.*
>
> Propositions 4 and 5 both give examples of common distributions and associated constants for Assumption 2 that can be used in practice. We also include in Appendix B a more detailed discussion of the necessity for each of these constants and intuition for why we need these assumptions. While we unfortunately had to move this to the appendix for the submission, we are also happy to elaborate on this more in the body for the final version.
>
>
> >*Furthermore, do the authors have any insight on whether this initial sample complexity is tight or can it still be improved in the future?*
>
> There is of course a trivial $\lambda d$ lower bound on the number of samples needed to do $\lambda$-exploration, so we also know that $poly(d)$ is the right rate. Closing the gap between our upper bound and the linear in $d$ lower bound is a very interesting question for future work!
>
> Furthermore, the examples discussed in Appendix B actually imply lower bounds on the number of samples necessary for exploration in terms of the parameters of Assumption 2. More formally, these examples imply the following three lower bounds. For proof sketches of these three lower bounds, please see the rebuttal to **Reviewer V6tS** (they are omitted here due to space constraints).
>
> **Lower bound $c_v:$** There exist instances (i.e. prior distributions) that require $\Omega(1/c_v)$ samples for $1$-spectral exploration.
>
> **Lower bound $c_d:$** There exist instances that require $\Omega(c_d)$ samples for $1$-spectral exploration.
>
> **Lower bound $\epsilon_d:$** There exist instances that require $\Omega(\log(1/\epsilon_d))$ samples for $1$-spectral exploration.
>
> The three lower bounds described above give a more complete picture of how are algorithm is tight in a polynomial sense for these problem parameters. Furthermore, these results combined with Propositions 4-5 imply a stronger lower bound than just linear in $d$ for the distributions discussed in these propositions. We are very happy to include these more detailed discussions about problem-parameter lower bounds in the final version of the paper.

---

> > ### Comment · Reviewer_Rjs1 · 2025-08-04
> >
> > Thank you for the reply. Given the response from the authors, I would maintain my initial score, which is already 'accept'.

---

### Decision · Program_Chairs · 2025-09-17

**Decision:**

Accept (spotlight)

**Comment:**

Previous work, Selke 2023, studied incentivized exploration in linear bandits and proposed Thompson sampling as an exploration method. An unfortunate property of this algorithm is that it requires a number of initial samples that is exponentially large in the dimension before it becomes BIC (Bayesian Incentive Compatible). The current paper addresses this gap, proposing a new method for generating initial samples that becomes BIC after a polynomial number of samples.

Strengths
- The improvement over Selke is significant
- The initial sampling algorithm and its analysis is novel
- The paper is well written, including clear proof sketches, and there is a good discussion of prior work

Weaknesses
- The paper did not include any experiments, though the authors addressed this in their rebuttal
- The reviewers had some questions about the relevance of incentivized exploration to practice and whether the assumptions (e.g., of a linear bandit) restrict the applicability. It is worth observing that incentivized exploration of linear bandits is a well-accepted problem. It and similar problems have been studied by a number of researchers and is generally viewed as important from a theoretical perspective, even if the current level of applicability is not as high as with some other parts of the bandit literature.
- The reviewers identified some opportunities to improve clarity, which were discussed in the rebuttal.

Overall, this is a solid theory paper addressing an important gap in the literature. It is of note that the two reviewers who engaged with the paper most closely and who are most familiar with this part of the literature rated it both 5.  I am recommending acceptance.